# USP14-regulated allostery of the human proteasome by time-resolved cryo-EM

Shuwen Zhang[1,2,6], Shitao Zou[1,2,6], Deyao Yin[1,2], Lihong Zhao[1,2], Daniel Finley[3], Zhaolong Wu[1,2,4] & Youdong Mao[1,2,4,5 ✉]

Proteasomal degradation of ubiquitylated proteins is tightly regulated at multiple levels[1–3]. A primary regulatory checkpoint is the removal of ubiquitin chains from substrates by the deubiquitylating enzyme ubiquitin-specific protease 14 (USP14), which reversibly binds the proteasome and confers the ability to edit and reject substrates. How USP14 is activated and regulates proteasome function remain unknown[4–7]. Here we present high-resolution cryo-electron microscopy structures of human USP14 in complex with the 26S proteasome in 13 distinct conformational states captured during degradation of polyubiquitylated proteins. Time-resolved cryo-electron microscopy analysis of the conformational continuum revealed two parallel pathways of proteasome state transitions induced by USP14, and captured transient conversion of substrate-engaged intermediates into substrate-inhibited intermediates. On the substrate-engaged pathway, ubiquitin-dependent activation of USP14 allosterically reprograms the conformational landscape of the AAA-ATPase motor and stimulates opening of the core particle gate[8–10], enabling observation of a near-complete cycle of asymmetric ATP hydrolysis around the ATPase ring during processive substrate unfolding. Dynamic USP14–ATPase interactions decouple the ATPase activity from RPN11-catalysed deubiquitylation[11–13] and kinetically introduce three regulatory checkpoints on the proteasome, at the steps of ubiquitin recognition, substrate translocation initiation and ubiquitin chain recycling. These findings provide insights into the complete functional cycle of the USP14-regulated proteasome and establish mechanistic foundations for the discovery of USP14-targeted therapies.

The majority of cellular proteins in eukaryotes are targeted to the 26S proteasome for degradation by ubiquitylation pathways, which regulate major aspects of cellular processes[1,10]. The proteasome holoenzyme is assembled from a cylindrical 20S core particle (CP) capped with one or two 19S regulatory particles (RPs), each consisting of the lid and base subcomplexes[1,10]. The ring-like heterohexameric motor of the AAA (ATPase associated with diverse cellular activities) family of adenosine triphosphatase (ATPase) in the base regulates substrate processing in the proteasome via multiple modes of coordinated ATP hydrolysis[10]. The proteasome is dynamically regulated by numerous proteins that reversibly associate with it via mechanisms that remain unknown[3].

USP14 is one of the three proteasome-associated deubiquitinating enzymes[2] (DUBs) and is crucially involved in the regulation of proteostasis, inflammation, neurodegeneration, tumorigenesis and viral infection[2,14,15]. USP14 is a potential therapeutic target for treating cancer, inflammatory and neurodegenerative diseases[6,14,16,17]. In common with its yeast orthologue Ubp6, USP14 has major roles in proteasome regulation and is prominently activated upon reversible association with the proteasome[4–7]. USP14 and Ubp6 stabilize many cellular proteins against proteasomal degradation by ubiquitin chain disassembly as well

as noncatalytically[18–20]. Unlike the stoichiometric subunit RPN11 (also known as PSMD14), the DUB activity of which is coupled to ATP-driven substrate translocation, USP14 catalyses removal of supernumerary ubiquitin chains on a substrate en bloc, independently of ATPase activity, until a single chain remains[7]. Paradoxically, binding of ubiquitylated substrates to USP14 stimulates proteasomal ATPase activity and CP gate opening[20–23]. The molecular mechanisms underlying USP14 activation and its regulation of the proteasome remain unknown.

Previous cryo-electron microscopy (cryo-EM) studies have determined the atomic structures of the USP14-free, substrate-engaged proteasomes at key functional steps, including ubiquitin recognition (states $E_{A1}$ and $E_{A2}$), deubiquitylation (state $E_B$), translocation initiation (states $E_{C1}$ and $E_{C2}$) and processive degradation (states $E_{D1}$ and $E_{D2}$)[10]. Early cryo-EM reconstructions have revealed the approximate locations of USP14 and Ubp6 in the proteasome[23–26]. However, the insufficient resolution and the absence of polyubiquitylated substrates in these studies preclude understanding of USP14-mediated proteasome regulation at the atomic level. In addition, the highly dynamic nature of USP14–proteasome association has hampered structural determination of full-length USP14. Here we report time-resolved cryo-EM studies

[1]State Key Laboratory for Artificial Microstructures and Mesoscopic Physics, School of Physics, Peking University, Beijing, China. [2]Peking-Tsinghua Joint Center for Life Sciences, Peking University, Beijing, China. [3]Department of Cell Biology, Harvard Medical School, Boston, MA, USA. [4]Center for Quantitative Biology, Peking University, Beijing, China. [5]National Biomedical Imaging Center, Peking University, Beijing, China. [6]These authors contributed equally: Shuwen Zhang, Shitao Zou. ✉e-mail: ymao@pku.edu.cn

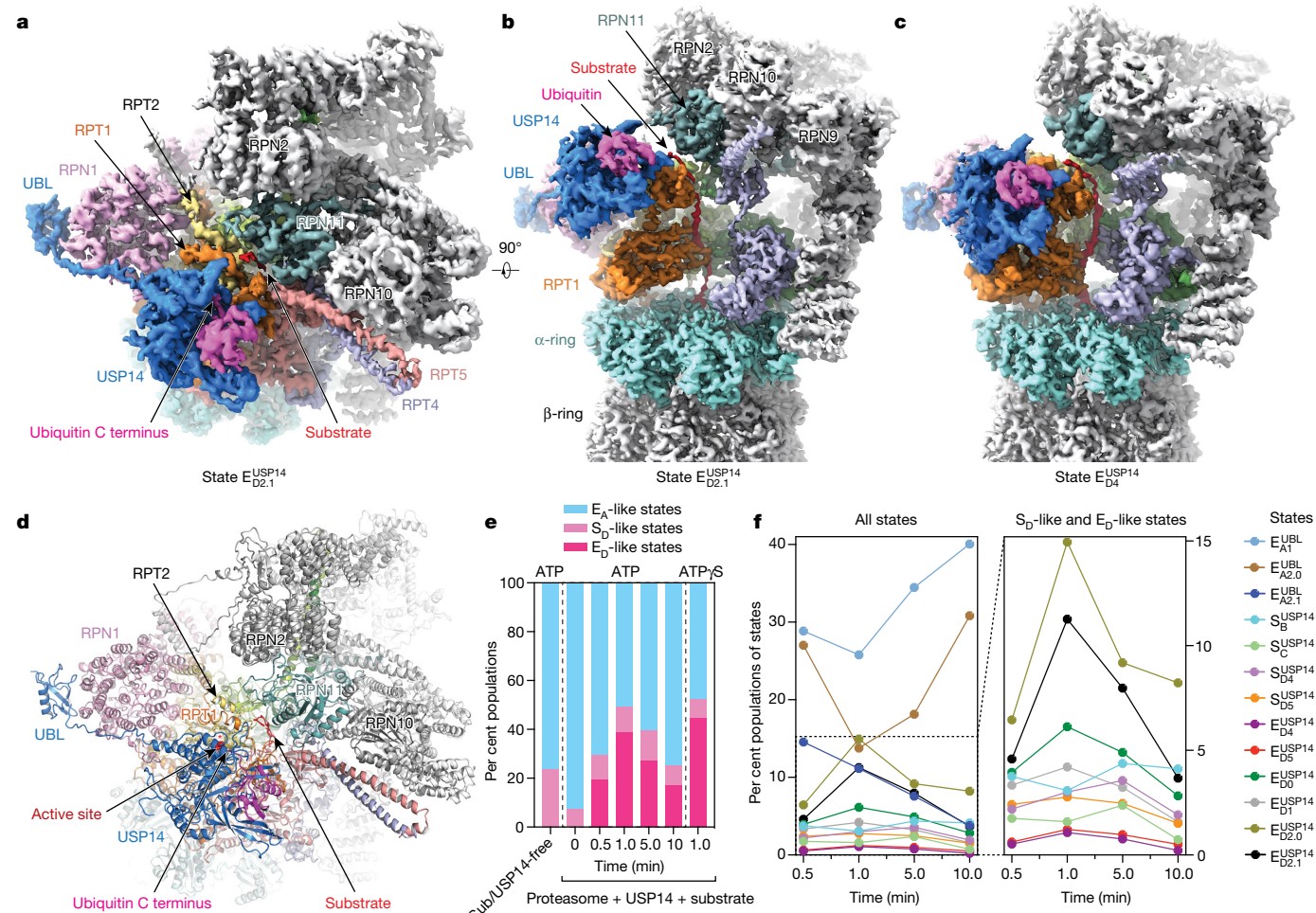

**Fig. 1 | Time-resolved cryo-EM analysis of the conformational landscape of USP14–proteasome complexes in the act of substrate degradation.**
**a**, **b**, Cryo-EM density map of the substrate-engaged USP14–proteasome complex in state $E_{D2.1}^{USP14}$, viewed from the top (**a**) and side (**b**). **c**, Side view of the cryo-EM density map of the substrate-engaged USP14–proteasome complex in state $E_{D4}^{USP14}$. Compared to the view of $E_{D2.1}^{USP14}$ in **b**, USP14 is rotated about 30° to dock onto the AAA domain of RPT1. To visualize the substrate density inside the AAA-ATPase motor, the density of RPT5 is omitted in both **b** and **c**. **d**, Atomic model of state $E_{D2.1}^{USP14}$ viewed from the same perspective as in **a**. **e**, Kinetic changes of overall particle populations of $S_D$-like and $E_D$-like states versus $E_A$-like states obtained from time-resolved cryo-EM analysis. $E_A$-like states include $E_{A1}^{UBL}$, $E_{A2.0}^{UBL}$ and $E_{A2.1}^{UBL}$. $S_D$-like states include $S_B^{USP14}$, $S_C^{USP14}$ $S_{D4}^{USP14}$ and $S_{D5}^{USP14}$.

$E_D$-like states include $E_{D4}^{USP14}$, $E_{D5}^{USP14}$, $E_{D0}^{USP14}$, $E_{D1}^{USP14}$, $E_{D2.0}^{USP14}$ and $E_{D2.1}^{USP14}$. The control consists of previously reported data for substrate-free, USP14-free proteasome[8]. **f**, Kinetic changes of the particle populations of 13 coexisting conformational states of USP14-bound proteasome from the cryo-EM samples made at different time points after mixing the substrate with the USP14–proteasome complex in the presence of 1 mM ATP at 10 °C. Three substrate-inhibited intermediates ($S_B^{USP14}$, $S_C^{USP14}$ and $S_{D4}^{USP14}$) reach their maximal populations at around 5 min, in contrast to state $S_{D5}^{USP14}$ and six substrate-engaged states, which all reach their maximal populations at approximately 1 min. The number of particles used in **e** and **f** are provided in Extended Data Fig. 2b, c.

of human USP14 in complex with functional proteasome in the act of substrate degradation. Our structural and functional analyses portray a dynamic picture of USP14–proteasome interactions and reveal the mechanism of allosteric 'tug-of-war' between USP14 and the proteasome for deciding substrate fate.

## Visualizing intermediates of USP14–proteasome

To prepare a substrate-engaged USP14–proteasome complex, we separately purified human USP14, RPN13 (also known as ADRM1) and USP14-free 26S proteasome (Extended Data Fig. 1). Although the ubiquitin receptor RPN13 appears to be present in the purified USP14-free proteasome (Extended Data Fig. 1e), it was absent from previous cryo-EM reconstructions of human proteasomes[8–10,25,27–29], probably due to its sub-stoichiometric levels. In an attempt to saturate the proteasome with RPN13, and thereby enhance substrate recruitment by the USP14-bound proteasome, we mixed stoichiometric excesses of both

purified USP14 and RPN13 with the USP14-free proteasome before substrate addition. We used Sic1[PY] conjugated with Lys63-linked polyubiquitin chains ($Ub_n$–Sic1[PY]) as a model substrate[7]. Lys63-linked chains are the second most abundant ubiquitin linkages in mammals and regulate essential intracellular functions such as endocytosis, DNA repair and immune responses[1]. We collected cryo-EM datasets for samples prepared by cryo-plunging at 0.5, 1, 5 and 10 min after mixing $Ub_n$–Sic1[PY] with the USP14-bound proteasome in the presence of 1 mM ATP at 10 °C. Because 3D classification of cryo-EM data indicated that intermediate states were maximized approximately 1 min after substrate addition, we collected considerably more data at this time point (Fig. 1e, f). To facilitate structural determination at high resolution, we collected another large dataset by exchanging ATP with the slowly hydrolysed ATPγS 1 min after substrate addition at 10 °C, which is expected to stall all coexisting intermediate conformations[10]. Deep learning-enhanced 3D classification[27] enabled us to determine 13 distinct conformational states of the USP14-bound proteasome, including 3 $E_A$-like, 4 $S_D$-like and

6 $E_D$-like conformers, at nominal resolutions of 3.0–3.6 Å (Fig. 1a–d, Extended Data Figs. 2–4, Extended Data Table 1). As expected, each of the 13 states was consistently observed under both the ATP-only and ATP-to-ATPγS exchange conditions despite the differences in state populations (Extended Data Figs. 1r, 2, 5a).

The six $E_D$-like substrate-engaged states—designated $E_{D0}^{USP14}$, $E_{D1}^{USP14}$, $E_{D2.0}^{USP14}$, $E_{D2.1}^{USP14}$, $E_{D4}^{USP14}$ and $E_{D5}^{USP14}$—captured sequential intermediates during processive substrate unfolding and translocation, which are compatible with the hand-over-hand translocation model[10,30] (Extended Data Figs. 6, 7, Extended Data Table 2). The cryo-EM density of USP14 in state $E_{D2.1}^{USP14}$ is of sufficient quality to allow atomic modelling of full-length USP14 (Extended Data Fig. 4c, d). An unfolded polypeptide substrate is observed in the AAA-ATPase channel in all of the $E_D$-like states. These states also exhibit an open CP gate, with five C-terminal tails (C-tails) of ATPase subunits (RPT1–RPT6, excluding RPT4) inserted into the inter-subunit surface pockets on the α-ring (α-pockets) of the CP[8–10,30,31] (Fig. 3a, Extended Data Fig. 6h).

There are no substrate densities observed in the AAA-ATPase channel of the four $S_D$-like states, designated $S_B^{USP14}$, $S_C^{USP14}$, $S_{D4}^{USP14}$ and $S_{D5}^{USP14}$. Substrate insertion into the AAA-ATPase channel is sterically inhibited in these states, because RPN11 blocks the substrate entrance at the oligonucleotide- or oligosaccharide-binding (OB) ring of the AAA-ATPase motor (Extended Data Fig. 7f). The AAA-ATPase conformations of $S_B^{USP14}$, $S_C^{USP14}$, $S_{D4}^{USP14}$ and $S_{D5}^{USP14}$ resemble those of $S_B$, $S_{D1}$, $E_{D4}^{USP14}$ and $E_{D5}^{USP14}$, respectively[8,9] (Extended Data Figs. 5f, 7d). However, the ATPase–CP interfaces of states $S_B^{USP14}$, $S_C^{USP14}$ and $S_{D4/5}^{USP14}$ resemble those of states $E_A$, $E_C$ and $E_D$, where two, four and five RPT C-tails are inserted into the α-pockets, respectively[10] (Extended Data Fig. 6h). Thus, the CP gate remains closed in $S_B^{USP14}$ and $S_C^{USP14}$, but is open in $S_{D4}^{USP14}$ and $S_{D5}^{USP14}$.

Human USP14 comprises 494 amino acids and features a 9-kDa ubiquitin-like (UBL) domain at its N terminus, followed by a 43-kDa USP domain joined via a flexible linker region of 23 amino acids. The overall USP14 structure bridges the RPN1 and RPT1 subunits in all the $S_D$-like and $E_D$-like states, in agreement with previous low-resolution studies[23–26] (Extended Data Fig. 5c). By contrast, the three $E_A$-like states, designated $E_{A1}^{UBL}$, $E_{A2.0}^{UBL}$ and $E_{A2.1}^{UBL}$, show no visible density for the catalytic USP domain of USP14, although a density consistent with the UBL domain is observed at the T2 site of RPN1[27,32] (also known as PSMD2) (Extended Data Figs. 4a, 7e).

## Time-resolved conformational continuum

Time-dependent quantification of the degraded $Ub_n$–Sic1$^{PY}$ substrate indicates that USP14 reduced the degradation rate by approximately two-fold compared with that of the USP14-free proteasome[7] (Extended Data Fig. 1k). The majority (about 80%) of substrates were eventually degraded by the USP14–proteasome mixture after 30 min of reaction under the same conditions used for our time-resolved cryo-EM analysis. Notably, the substrate-engaged and substrate-inhibited intermediates coexisted at all measured time points and reached their maximal per cent levels at approximately 1 and 5 min, respectively, after substrate addition (Fig. 1e). The population of $S_B^{USP14}$ became the largest among the substrate-inhibited states in 5–10 min (Fig. 1f). The overall population of substrate-inhibited intermediates varied within a small range of 7.7–10.3% from 0.5 to 5 min and was comparable to the 7.3% of the $S_D$-like states before substrate addition (Fig. 1e, f, Extended Data Fig. 2b, c), suggesting that the substrate-inhibited pathway was induced in parallel to the substrate-engaged pathway and was promoted by USP14 despite the stoichiometric excess of ubiquitylated substrates. Consistent with previous studies[23], these results indicate that the substrate-engaged intermediates were converted to substrate-inhibited states upon termination of substrate translocation.

Despite exhaustive 3D classification, we did not observe in any USP14-loaded experimental conditions the proteasome states $E_B$ and $E_C$, which represent RPN11-mediated deubiquitylation and translocation initiation prior to the CP gate opening, respectively[10]. This implies that USP14 prevents the proteasome from assuming the conformation of RPN11-catalysed deubiquitylation. In contrast to the RPN11-bound ubiquitin in states $E_{A2.0}^{UBL}$ and $E_{A2.1}^{UBL}$, no ubiquitin was observed on RPN11 in any state in which USP14 is engaged with ubiquitin (Fig. 1a–d, Extended Data Figs. 4a, 7a–c), suggesting that activated USP14 and RPN11 do not bind ubiquitin simultaneously.

## Dynamic USP14–proteasome interactions

The UBL domain of USP14 binds the RPN1 T2 site via a hydrophobic patch centred on residue Leu70, which is structurally homologous to the Ile44 patch of ubiquitin (Fig. 2a). The RPN1 T2 site is composed of residues Asp423, Leu426, Asp430, Tyr434, Glu458, Asp460 and Leu465 on two adjacent helix-loop regions, in agreement with previous findings[27,32]. The N-terminal stretch of the linker (residues Ala77 to Phe88) in USP14 appears to bind the ridge of the toroid domain of RPN1 (Fig. 2a).

USP14 interacts with both the OB and AAA domains of the ATPase ring. The USP–OB interaction is mediated by the blocking loops 1 (BL1), 2 (BL2) and 3 (BL3) of the USP domain (Fig. 2b), which buries a solvent-accessible area of approximately 527 Å$^2$ on USP14. BL1 makes the most extensive interface with the OB ring around residues Gln128 and Asp133 of RPT1 (also known as PSMC2), whereas Ser430 in BL2 and Trp472 in BL3 interact with Gln128 of RPT1 and Asp118 of RPT2 (also known as PSMC1), respectively (Fig. 2c). The movement of the USP domain between different proteasome states appears to adapt to the rocking motion of the AAA-ATPase motor and maintains its interaction with the OB ring (Fig. 3a–d, Extended Data Fig. 8a).

In contrast to the USP–OB interface, which is nearly invariant in all $S_D$-like and $E_D$-like states, USP-AAA interactions vary prominently (Fig. 3a, b). In states $E_{D0}^{USP14}$, $E_{D1}^{USP14}$, $E_{D2.0}^{USP14}$ and $E_{D2.1}^{USP14}$, the USP domain is flipped up and completely detached from the AAA domain of RPT1. By contrast, the USP domain exhibits differential interactions with the AAA domain of RPT1 in other states (Fig. 3h, Extended Data Fig. 8a), suggesting that USP14 preferentially recognizes these AAA conformations and stabilizes these states by direct interactions. The most extensive USP–AAA interactions are observed in state $E_{D4}^{USP14}$ (Fig. 3h–k). In this conformation, a helix-loop region (residues 371–391) protruding from the USP domain contacts the AAA domain of RPT1, which buries a solvent-accessible area of approximately 850 Å$^2$ on USP14.

## USP14 activation by the proteasome

Occlusion of the ubiquitin C terminus-binding groove in USP14 by BL1 and BL2 auto-inhibits the DUB activity in the absence of the proteasome[5]. Comparison of the USP14 structure in state $E_{D2.1}^{USP14}$ with the crystal structures of the USP domain in isolated and ubiquitin aldehyde-bound forms[5] reveals differential conformational changes in BL1 and BL2 (Fig. 2b). The BL1 region is an open loop in the crystal structures but is folded into a β-hairpin sandwiched between the OB ring and ubiquitin in the proteasome (Fig. 2b, d). By contrast, the BL2 loop is moved approximately 4 Å to make way for docking of the ubiquitin C terminus into the active site groove. Stabilized by the USP–OB interface, BL1 and BL2 together hold the ubiquitin C terminus in a β-strand conformation (Fig. 2c), in which the main chain carbon of ubiquitin C-terminal Gly76 is placed approximately 3.4 Å away from the sulfur atom of the catalytic Cys114 and is fully detached from the substrate, indicating that the structure represents a post-deubiquitylation state of USP14 (Fig. 2e). This ternary architecture of the ubiquitin–USP–OB sandwich indicates that ubiquitin binding stabilizes the BL1 β-hairpin conformation and the USP interaction with the OB ring. Indeed, microscale thermophoresis (MST) measurements show that the dissociation constant (approximately 44 nM) of USP14 with the proteasome in the presence of $Ub_n$–Sic1$^{PY}$ is approximately half of that (approximately 95 nM) in the absence of ubiquitin conjugates and one third of that (approximately

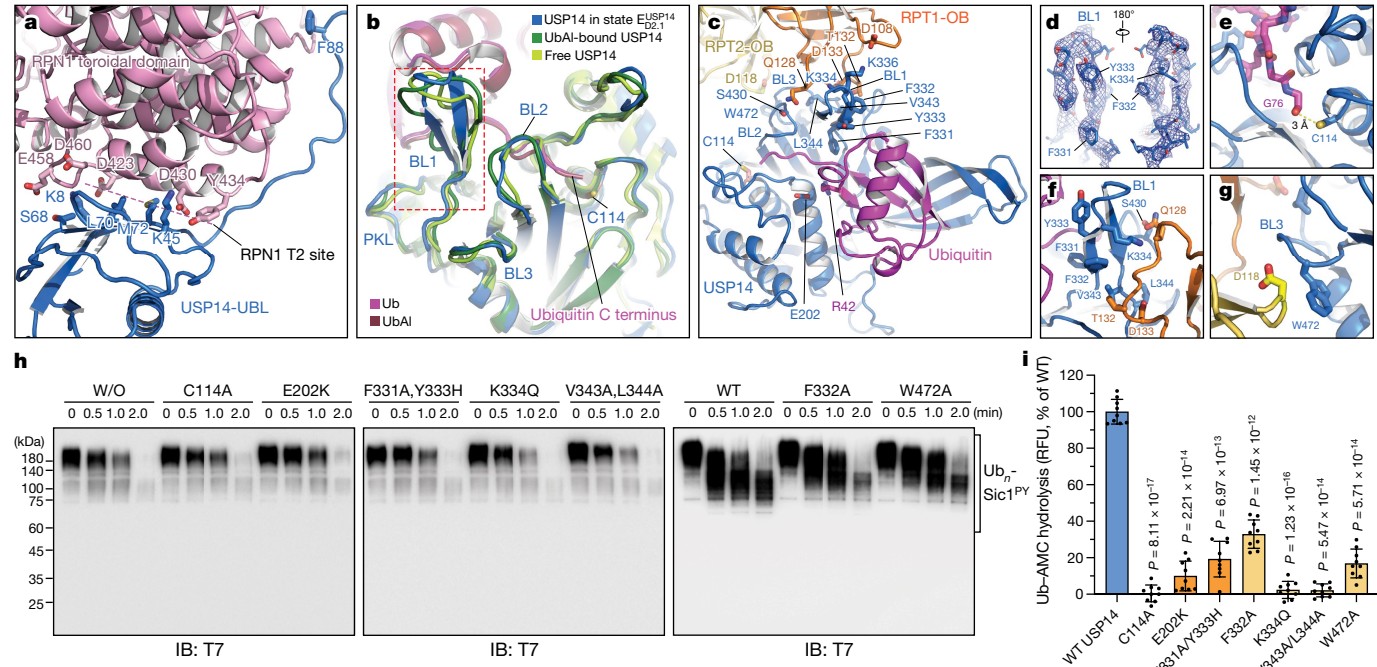

**Fig. 2 | Structural basis of proteasome-mediated activation of USP14.**
**a**, Side-chain interactions between the USP14 UBL domain and the RPN1 T2 site in the proteasome state $E_{D2.1}^{USP14}$. **b**, Structural comparison of the blocking loops by superimposing the USP14 structure in state $E_{D2.1}^{USP14}$ with two crystal structures of USP14 in its isolated form (PDB ID: 2AYN) and in complex with ubiquitin aldehyde[5] (UbAl) (PDB ID: 2AYO). **c**, Magnified view of the ubiquitin–USP–OB sandwich architecture in state $E_{D2.1}^{USP14}$. **d**, Local cryo-EM density of the BL1 motif in state $E_{D2.1}^{USP14}$ in mesh representation superimposed with its atomic model in cartoon representation from two opposite orientations, showing its β-hairpin conformation. **e–g**, Magnified views of the interfaces between the catalytic Cys114 of USP14 and the C-terminal Gly76 of ubiquitin (**e**), between the USP14 BL1 motif and the RPT1 OB domain (**f**), and between the USP14 BL3 motif and the RPT2 OB domain (**g**). Key residues mediating the inter-molecular interactions are shown in stick representation in **a–g**. **h**, In vitro degradation of

$Ub_n$–$Sic1^{PY}$ by the human proteasome assembled with USP14 variants at 37 °C, analysed by SDS–PAGE and western blot using anti-T7 antibody to visualize the fusion protein T7–$Sic1^{PY}$. See Supplementary Fig. 1 for gel source data. These experiments were repeated independently three times with consistent results. The proteasome without USP14 (no USP14; labelled W/O above the leftmost lanes) is used as a negative control. Lanes labelled WT correspond to the proteasome bound to wild-type USP14. **i**, Ubiquitin–AMC hydrolysis by the USP14 mutants dictates their DUB activity in the proteasome. RFU, relative fluorescence units. RFU values at 60 min are shown. All labelled $P$ values were computed against the wild-type USP14 using a two-tailed unpaired $t$-test. Data are mean ± s.d. from three independent experiments. Each experiment includes three replicates. The quantification of wild-type USP14 was used as a denominator to normalize all measurements.

137 or 135 nM, respectively) of either UBL or USP domain alone (Extended Data Fig. 1l–o). Thus, USP14 affinity towards the proteasome reflects interactions of both its UBL and USP domains and is enhanced by $Ub_n$–$Sic1^{PY}$, consistent with previous studies[33].

To test the functional importance of the ubiquitin–USP–OB interface in USP14 activation, we performed structure-based site-directed mutagenesis (Extended Data Fig. 9). Indeed, the USP mutations, including the single mutant K334Q and the double mutant V343A/L344A at the BL1–OB interface (Fig. 2f), F331A/Y333A at the ubiquitin–BL1 interface, as well as the single mutant E202K that disrupts the salt bridge between Glu202 of USP and Arg42 of ubiquitin (Fig. 2c), all abrogated the DUB activity of USP14—similar to the effect by C114A mutation that removes the active site of USP14—and showed no obvious inhibition of proteasomal degradation (Fig. 2h, i, Extended Data Fig. 9e, f). By contrast, the single mutants BL1(F332A) and BL3(W472A) retained a reduced DUB activity and suppressed proteasome function to a lesser extent (Fig. 2f–i). These contrasting phenotypes substantiate our structural finding that the ubiquitin–USP–OB interactions are essential for USP14 activation towards efficacious deubiquitylation.

The overall structure of full-length USP14 exhibits three major states (Extended Data Fig. 8b), far fewer than the number of USP14-bound proteasome states, suggesting that the conformational entropy of the linker region is greatly reduced upon USP14 assembly and activation on the proteasome. In the linker region, the residues Glu90 and Asp91 are potentially involved in transient interactions with the RPN1 toroid

(Extended Data Figs. 7g, 9a). Deletion of residues 93–96 and insertion of TEEQ after residue 92 in the linker suppressed proteasomal degradation of $Ub_n$–$Sic1^{PY}$ more potently than the double point mutation E90K/D91A (Extended Data Fig. 9d, g). All three USP14 mutants showed 20–30% reductions in DUB activity relative to the wild-type USP14. These observations suggest that the linker length and composition of USP14 may have been evolutionarily optimized for DUB activity, although the UBL and USP domain architecture appears to be functionally robust against variations of the linker region.

## Allosteric regulation of ATPase activity

USP14 interacts with the AAA domain of RPT1 via two discrete interfaces in state $E_{D4}^{USP14}$ (Fig. 3a, Extended Data Fig. 4f). The primary USP–AAA interface is composed of a negatively charged surface on a helix (residues 371–383) and a loop (residues 384–391) protruding out of the USP domain, which buries the majority of the USP–AAA interface (Fig. 3g, h). The helix region appears to pack against a convex surface of the large AAA subdomains of RPT1, on which Arg371, Lys375 and Asn383 of USP14 interact with Glu185, His197 and Glu199 of RPT1, whereas the loop region (384–391) reaches the small AAA subdomain of RPT1 (Fig. 3i). The second USP–AAA interface is centred on Tyr285 in the PKL loop region of USP14 that interacts with Lys267 and Lys268 of RPT1 (Fig. 3j).

In state $E_{D4}^{USP14}$, ADP is bound to the Walker A motif between the large and small AAA subdomain of RPT1, the dihedral angle of which is directly

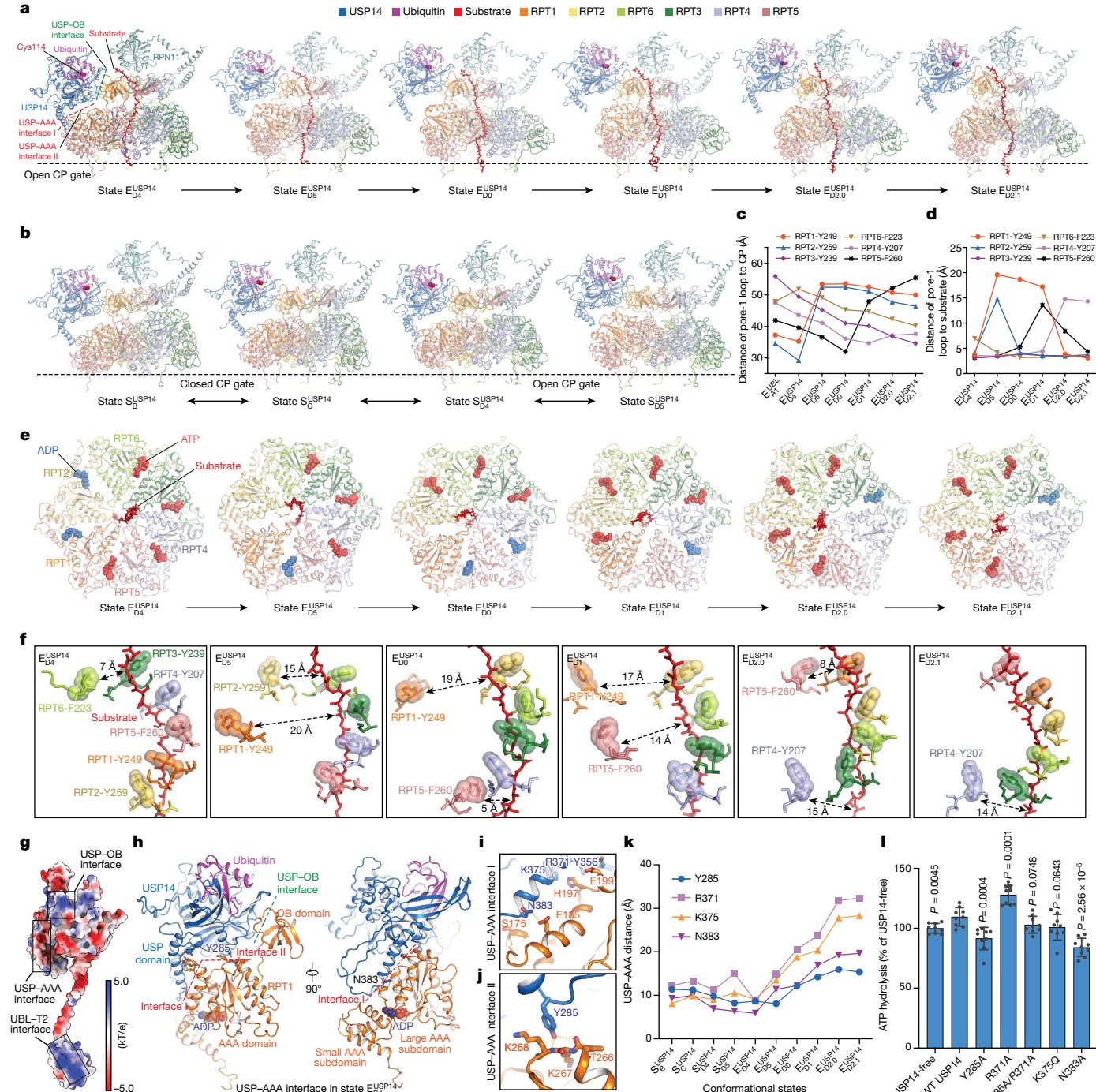

**Fig. 3 | Structural dynamics and mechanism of allosteric regulation of the AAA-ATPase motor by USP14. a, b,** Side-by-side comparison of the USP14–ATPase subcomplex structures aligned against the CP in six substrate-engaged states (**a**) and four substrate-inhibited states (**b**). **c, d,** Plots of distance from the pore-1 loop of each ATPase to the CP (**c**) and to the substrate (**d**) in distinct states. **e,** The AAA domain structures of the ATPase motor in six substrate-engaged states. **f,** Varying architecture of the pore-1 loop staircase interacting with the substrate in distinct states. The distances from disengaged pore-1 loops to the substrate are marked. The side chains of the pore-1 loop residues, featuring a consensus sequence of K/M-Y/F-V/L/I, are shown in stick representation, with the aromatic residues highlighted in transparent sphere representation. **g,** Electrostatic surface representation of the full-length USP14, coloured according to electrostatic potential from red ($-5.0$ kT e$^{-1}$,

negatively charged) to blue ($5.0$ kT e$^{-1}$, positively charged). **h,** Atomic model of the USP14–RPT1 subcomplex in state $E_{D4}^{USP14}$ in cartoon representation. **i, j,** Magnified views of the USP-AAA interface I (**i**) and II (**j**) in state $E_{D4}^{USP14}$, with the interacting pairs of residues in stick representation. **k,** Changes of the USP-AAA interface in distinct states, characterized by measuring the shortest distance between four USP14 residues (Y285, R371, K375 and N383) and the main chains of RPT1 AAA domain. **l,** ATPase activity was quantified by measuring the release of phosphate from ATP hydrolysis of the proteasome. All labelled $P$ values were computed by comparison with wild-type USP14 using a two-tailed unpaired $t$-test. Data are mean ± s.d. from three independent experiments, each with three replicates. The quantification of USP14-free proteasome was used as a denominator to normalize all measurements in each experiment.

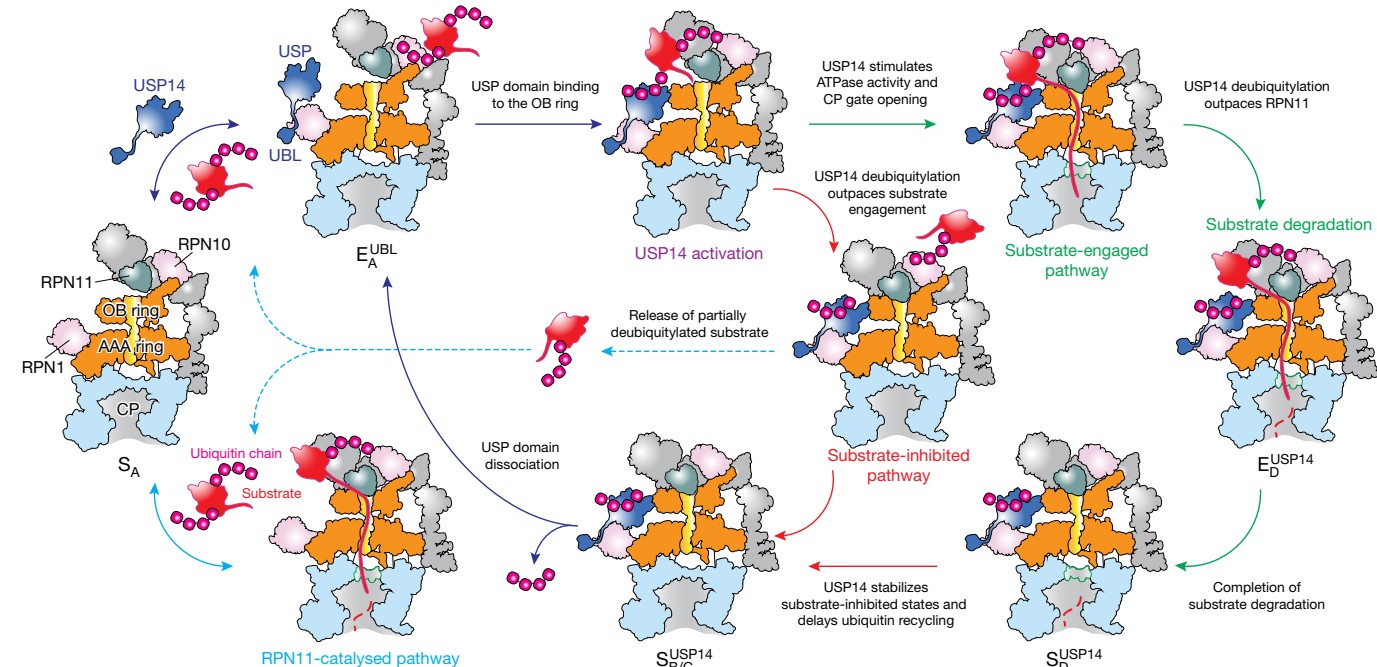

**Fig. 4 | Proposed model of USP14-mediated regulation of proteasome function.** USP14 binding to the RPN1 and RPT1 subunits of the proteasome primes USP14 activation, whereas ubiquitin–substrate conjugates recruited to the proteasome's ubiquitin receptors facilitate ubiquitin recognition by USP14. RPN11-catalysed pathway (turquoise solid arrow) is allosterically excluded once USP14 is recruited to the proteasome (dark blue arrows). USP14 binding creates two parallel state-transition pathways of the proteasome. Along the substrate-inhibited pathway (red arrows), which has RPN11 blocking the substrate entrance of the OB ring before any substrate insertion takes place, USP14 trims ubiquitin chains and releases the substrate from the proteasome, thus preventing the substrate degradation (dashed turquoise arrows). Along

the substrate-engaged pathway (green arrows), a substrate has already been inserted into the ATPase ring and RPN11 narrows down on the OB ring but does not block substrate translocation through the OB ring (Extended Data Fig. 7f). Although our data do not intuitively explain why USP14 trims ubiquitin until the last one on a substrate remains, the structures provide geometric constraints for polyubiquitin chain binding to both ubiquitin receptors and USP14 and suggest that ubiquitin recognition by USP14 in the proteasome requires at least one additional helper ubiquitin chain that is already anchored on a nearby ubiquitin receptor. This helper ubiquitin chain may not be available for USP14 binding but can be readily trimmed by RPN11.

stabilized by the USP–AAA interaction (Fig. 3h). In other substrate-engaged states, in which the USP domain is detached from the AAA domain, the nucleotide-binding pocket of RPT1 is either empty or bound to ATP (Fig. 3e, Extended Data Fig. 6j). Thus, the USP–AAA interaction energetically stabilizes the ADP-bound conformation of RPT1 and USP–AAA dissociation promotes nucleotide exchange in RPT1. We postulate that the USP–AAA interaction may allosterically promote the ATPase activity during substrate degradation. Indeed, wild-type USP14 enhanced the ATPase activity during proteasomal degradation of $Ub_n$–$Sic1^{PY}$ by approximately 10% relative to that of the USP14-free proteasome (Fig. 3l).

To test the functional roles of the USP–AAA interfaces, we mutated several USP14 residues at the USP–AAA interfaces to alanine. Although none of these USP14 mutants exhibited any observable defects in the DUB activity of USP14 or the peptidase activity of USP14-bound proteasome (Extended Data Fig. 9c, h), they perturbed the ATPase activity in the presence of polyubiquitylated substrate, in line with previous studies[21–23,34,35]. Two mutants, Y285A and N383A, showed approximately 20–30% reduction of the ATPase activity to a level 10–20% below that of the USP14-free proteasome (Fig. 3l). As Tyr285 is located halfway between the primary USP–AAA and USP–OB interfaces (Fig. 3f, h), it may be mechanically important in transmitting the allosteric effect from USP to AAA. Both Y285A and N383A mutations potentially reduce the overall USP association with the large AAA subdomain of RPT1 (Fig. 3i). By contrast, the single mutant R371A enhanced the ATPase activity by approximately 20% relative to the wild-type USP14, probably by tightening the USP–AAA interactions; whereas K375Q showed no significant effect on the ATPase activity, probably because only the main chain of K375 interacts with RPT1 His197 (Fig. 3j). Consistently, the double mutant Y285A/R371A restored the ATPase rate to the approximate level

of the USP14-free proteasome, presumably owing to the cancellation of two counteracting allosteric effects between the two mutated residues (Fig. 3l). Altogether, our structure-guided mutagenesis supports the notion that the USP–AAA interfaces mediate the non-catalytic, allosteric regulation of the ATPase activity in the proteasome.

## Asymmetric ATP hydrolysis around ATPase ring

The six substrate-engaged USP14–proteasome structures characterize a detailed intermediate sequence of substrate translocation, which deviates substantially from the pathway of translocation initiation of the USP14-free proteasome[10] (Extended Data Figs. 5, 6). Whereas the RP conformations of states $E_{D1}^{USP14}$ and $E_{D2.1}^{USP14}$ closely resemble those of USP14-free proteasomes in previously reported states $E_{D1}$ and $E_{D2}$, respectively, other substrate-engaged states of USP14–proteasome present distinct, hitherto unknown RP conformations[10] (Extended Data Figs. 5d–f, 6). Of note, the AAA-ATPase conformations in states $E_{D4}^{USP14}$ and $E_{D5}^{USP14}$ can be compared to those of USP14-free, closed-CP states $E_B$ and $E_{C1}$, respectively, albeit with defined structural differences[10,27] (Extended Data Fig. 5d, f). In particular, the pore-1 loop of RPT6 in state $E_{D4}^{USP14}$ has considerably moved up towards the substrate as compared with that in state $E_B$ (Extended Data Fig. 5e). Given the absence of the $E_B$- and $E_C$-like states in the presence of USP14, our structural data together prompt the hypothesis that $E_{D4}^{USP14}$ and $E_{D5}^{USP14}$ may replace the USP14-free states $E_B$ and $E_C$ during initiation of substrate translocation. Thus, we infer that these states present a continuum of USP14-altered conformations following the state-transition sequence $E_{A1}^{UBL} \rightarrow E_{D4}^{USP14} \rightarrow E_{D5}^{USP14} \rightarrow E_{D0}^{USP14} \rightarrow E_{D1}^{USP14} \rightarrow E_{D2.0}^{USP14} \rightarrow E_{D2.1}^{USP14}$ (Fig. 3c, Extended Data Fig. 7i, j). In support of this sequence assignment, the RPT3 pore-1 loop is moved

sequentially from the top to the bottom of the substrate–pore loop staircase, whereas coordinated ATP hydrolysis navigates around the ATPase ring for a near-complete cycle (Fig. 3c–f, Extended Data Fig. 6a–f).

The newly resolved states fill a major gap in our understanding of substrate translocation by the proteasomal AAA-ATPase motor. It has remained unclear whether a strict sequential hand-over-hand mechanism is used by the AAA-ATPase motor for processive substrate translocation[36]. Of note, four of the six $E_D$-like states—$E_{D5}^{USP14}$, $E_{D0}^{USP14}$, $E_{D1}^{USP14}$ and $E_{D2.0}^{USP14}$—exhibit an AAA-ATPase conformation with two adjacent RPT pore-1 loops disengaged from the substrate: one moving away from the substrate while releasing ADP for nucleotide exchange, and the other moving up towards the substrate upon binding ATP for substrate re-engagement (Fig. 3d–f). Recent studies of the proteasome have found that substrate re-engagement of an ATPase subunit can be the rate-limiting step in single-nucleotide exchange dynamics[27]. Coupling of conformational changes between adjacent AAA domains may vary from tightly coupled kinetics—that is, only one dissociated pore-1 loop at a time—to loosely coupled kinetics exhibiting two adjacent pore-1 loops dissociated from the substrate simultaneously[10,27]. Thus, two adjacent pore-1 loops dissociated from the substrate represent an inevitable intermediate in the sequential hand-over-hand model of substrate translocation, which would be very short-lived in tightly coupled kinetics[37]. Under the allosteric influence of USP14, such kinetics can become loosely coupled around RPT1, leading to meta-stabilization of these short-lived intermediate states. The observations of both tightly and loosely coupled kinetics at different locations around the ATPase ring rationalize broken symmetry of coordinated ATP hydrolysis in the AAA-ATPase motor—an effect that has been previously informed by functional studies[38,39].

## Insights into proteasome regulation by USP14

Our structural, kinetic and functional data collectively provide insights into how USP14 regulates proteasome activity at multiple checkpoints by inducing parallel pathways of proteasome state transitions (Fig. 4, Extended Data Fig. 7j). The first checkpoint is at the initial ubiquitin recognition before substrate engagement with the ATPases. USP14 assembly on the proteasome is initiated by its UBL domain binding to the T2 site of the RPN1 toroid[27,32] as suggested in state $E_{A1}^{UBL}$. Binding of the UBL domain alone has been previously suggested to allosterically stimulate proteasome activity[34,35]. This is followed by a distinct association step of its USP domain with the OB ring of the AAA-ATPase motor in a ubiquitin-dependent manner, which allosterically activates USP14 and reciprocally stabilizes the proteasome in the substrate-inhibited states. It appears that the USP-ATPase association is allosterically precluded at large once substrate-conjugated ubiquitin is recruited to RPN11, as suggested by states $E_{A2.0}^{UBL}$ and $E_{A2.1}^{UBL}$ (Extended Data Fig. 7a, b). Such a competition in ubiquitin recruitment forces the proteasome to bypass the conformational transition pathway represented by USP14-free states $E_B$ and $E_C$, which are needed to couple RPN11-catalysed deubiquitylation with ATP-dependent substrate engagement and translocation initiation[10]. Thus, the first checkpoint allows USP14 to allosterically compete against RPN11 in accepting substrate-conjugated ubiquitin.

However, the stabilization of substrate-inhibited states by USP14 activation does not completely exclude the possibility of substrate engagement with the ATPases. Should an unfolded or loosely folded initiation region of the substrate be available[40], substrate insertion into the AAA-ATPase channel can stochastically occur before USP14 activation by the proteasome, which presumably depends on the specific structures of substrate-ubiquitin conjugates. In the presence of ubiquitylated substrates, USP14 stimulates the ATPase rate, tightens the RP–CP interface and induces early CP gate opening[21–23], as represented in states $E_{D4}^{USP14}$ and $E_{D5}^{USP14}$. Thus, USP14 creates a second kinetic checkpoint and drives the proteasome to choose between two alternative pathways of USP14-regulated conformational transitions—one

that sterically impedes substrate commitment and the other that kinetically antagonizes RPN11 by outpacing the coupling of RPN11-directed deubiquitylation and substrate translocation, rather than directly inhibiting the DUB activity of RPN11. In support of this model, all $S_D$-like and $E_D$-like states of the USP14-bound proteasome showed no ubiquitin binding to RPN11.

Previous studies have demonstrated that USP14 trims ubiquitin chains from the substrate on a millisecond time scale[7]. Consistently, all observed USP14-bound $S_D$-like and $E_D$-like states appear to present USP-bound ubiquitin in a post-cleavage state, perhaps owing to the stabilization of trimmed ubiquitin by ternary interactions of the ubiquitin–USP–OB sandwich (Fig. 2c). The lack of observation of ubiquitin-free USP14 in the proteasome suggests that the rate of ubiquitin release from USP14 on the proteasome is slow as compared with that of ubiquitin-free USP14 from the proteasome. Thus, it is conceivable that the cleaved ubiquitin chain is released only upon dissociation of the USP domain from the OB ring[21,22,33]. This is in contrast to RPN11 in all the USP14-free $E_D$-like states[10], which exhibit no RPN11-bound ubiquitin, implying rapid release of cleaved ubiquitin. Therefore, USP14 creates a third checkpoint, where the polyubiquitin-bound USP14 kinetically delays ubiquitin release from the proteasome and suppresses additional substrate recruitment. The third checkpoint may further control the ubiquitin-recycling function of the proteasome that is critical for regulating the free ubiquitin reservoir in cells[41].

In summary, USP14 acts as an adaptive regulator of proteasomal DUB and ATPase activities that times intermediate steps of substrate processing, as USP14 bridges the gap between RPN1 and the AAA-ATPase motor in a ubiquitin-dependent, switchable fashion. USP14 interactions create three branching checkpoints on the proteasome, at the steps of initial ubiquitin recognition, substrate translocation initiation by the AAA-ATPase motor, and recycling of trimmed ubiquitin chains. This multi-checkpoint mechanism integrates catalytic and non-catalytic effects of USP14-mediated proteasome regulation into a comprehensively coordinated, elegantly timed process of substrate degradation. As partly supported by another study on the yeast Ubp6-proteasome interactions[42], such a mechanism is expected to be conserved from yeast to human and to inform on how reversibly associated DUBs and other proteins regulate proteasome function in general (Extended Data Fig. 9i). Importantly, our time-resolved cryo-EM studies, which had been enhanced by deep learning-improved 3D classification[27], present an emerging paradigm that enables atomic-level visualization of hitherto inaccessible functional kinetics and complex dynamics in general.

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

## Methods

### Expression and purification of human USP14

Wild-type USP14 and its mutants were cloned into pGEX-4T vector obtained from GenScript (Nanjing, China). For purification of recombinant USP14 and mutants, BL21-CondonPlus (DE3)-RIPL cells (Shanghai Weidi) transformed with plasmids encoding wild-type or mutant USP14 were grown to an $OD_{600}$ of 0.6–0.7 in LB medium supplemented with 100 mg ml$^{-1}$ ampicillin. Cultures were cooled to 20 °C and induced with 0.2 mM IPTG overnight. Cells were collected by centrifugation at 3,000$g$ for 15 min and resuspended in lysis buffer (25 mM Tris-HCl (pH 8.0), 150 mM NaCl, 0.2% NP-40, 1 mM DTT, 10% glycerol and 1× protease inhibitor cocktail). Cells were lysed by sonication and the lysate was cleared through centrifugation at 20,000$g$ for 30 min at 4 °C. The supernatant was incubated with glutathione Sepharose 4B resin (GE Healthcare) for 3 h at 4 °C. For the purification of wild-type USP14, USP14 UBL domain (USP14-UBL) and USP14 USP domain (USP14-USP), the resin was washed with 20 column volumes of washing buffer (25 mM Tris-HCl (pH 8.0), 150 mM NaCl, 1 mM DTT, 10% glycerol), then incubated with cleavage buffer (20 mM Tris-HCl (pH 8.0), 150 mM NaCl) containing thrombin (Sigma) overnight at 4 °C. The eluted samples were further purified on a gel-filtration column (Superdex 75, GE Healthcare) equilibrated with 25 mM Tris-HCl (pH 8.0), 150 mM NaCl, 1 mM DTT, 10% glycerol. For the purification of USP14 mutants, the resin was washed with 20 column volumes of washing buffer (25 mM Tris-HCl (pH 8.0), 300 mM NaCl, 1 mM DTT), then incubated with cleavage buffer (20 mM Tris-HCl (pH 8.0), 150 mM NaCl) containing thrombin (Sigma) overnight at 4 °C. To remove thrombin, the GST eluent was incubated with Benzamidine-Sepharose (GE Healthcare) for 30 min at 4 °C.

### Expression and purification of human 26S proteasome

Hexahistidine, TEV cleavage site, biotin and hexahistidine (HTBH)-tagged human 26S proteasomes were affinity purified as described[8–10] from a stable HEK 293 cell line (a gift from L. Huang). Further authentication of cell lines was not performed for this study. Mycoplasma testing was not performed for this study. In brief, HEK 293 cells were Dounce-homogenized in a lysis buffer (50 mM PBS (77.4% $Na_2HPO_4$, 22.6% $NaH_2PO_4$, pH 7.4), 5 mM $MgCl_2$, 5 mM ATP, 0.5% NP-40, 1 mM DTT and 10% glycerol) containing 1× protease inhibitor cocktail. The cleared lysates were incubated with Streptavidin Agarose resin (Yeasen) for 3 h at 4 °C. The resin was washed with 20 bed volumes of lysis buffer to remove endogenous USP14 and UCH37 associated with the proteasome[7,10]. The 26S proteasomes were cleaved from the beads by TEV protease (Invitrogen) and further purified by gel filtration on a Superose 6 10/300 GL column. Western blot was used to detect RPN13 and USP14 in the proteasomes using anti-RPN13 antibody (Abcam, 1:10,000 dilution) and anti-USP14 antibody (Novus, 1:1,000 dilution). For ubiquitin–vinyl-sulfone (Ub–VS)-treated human proteasome, 1 μM Ub–VS (Boston Biochem) was added to the proteasome-binding resin and incubated for 2 h at 30 °C. Residual Ub–VS was removed by washing the beads with 30 bed volumes of wash buffer (50 mM Tris-HCl (pH 7.5), 1 mM $MgCl_2$ and 1 mM ATP). The proteasomes were cleaved from the beads using TEV protease (Invitrogen) and used to measure the DUB activity of USP14 using the Ub–AMC hydrolysis assay.

### Preparation of polyubiquitylated Sic1$^{PY}$

Sic1$^{PY}$ and WW-HECT were purified as previously described[10]. The PY motif (Pro-Pro-Pro-Tyr) is recognized by the WW domains of the Rsp5 family of E3 ligases. In the Sic1$^{PY}$ construct, a PY motif was inserted to the N-terminal segment (MTPSTPPSRGTRYLA) of the Cdk inhibitor Sic1, resulting in a modified N terminus of MTPSTPPPPPYSRGTRYLA[43,44] (the PY motif is underlined). Human UBE1 (plasmid obtained as a gift from C. Tang) and human UBCH5A (obtained from GenScript) were expressed as GST fusion proteins from pGEX-4T vectors. In brief, UBE1-expressing BL21-CondonPlus (DE3)-RIPL cell cultures were

induced with 0.2 mM IPTG for 20 h at 16 °C, whereas UBCH5A expression was induced with 0.2 mM IPTG overnight at 18 °C. Cells were collected in lysis buffer (25 mM Tris-HCl (pH 7.5), 150 mM NaCl, 10 mM $MgCl_2$, 0.2% Triton-X-100, 1 mM DTT) containing 1× protease inhibitor cocktail and lysed by sonication. The cleared lysates were incubated with glutathione Sepharose 4B resin for 3 h at 4 °C and subsequently washed with 20 bed volumes of lysis buffer. The GST tag was removed by thrombin protease (Sigma) in cleavage buffer (20 mM Tris-HCl (pH 8.0), 150 mM NaCl, 1 mM DTT) overnight at 4 °C. The eluted samples were further purified by gel-filtration column (Superdex 75, GE Healthcare) equilibrated with 25 mM Tris-HCl (pH 7.5), 150 mM NaCl, 1 mM DTT, 10% glycerol.

To ubiquitylate Sic1$^{PY}$, 1.2 μM Sic1$^{PY}$, 0.5 μM UBE1, 2 μM UBCH5A, 1.4 μM WW-HECT and 1 mg ml$^{-1}$ ubiquitin (Boston Biochem) were incubated in reaction buffer (50 mM Tris-HCl (pH 7.5), 100 mM NaCl, 10 mM $MgCl_2$, 2 mM ATP, 1 mM DTT and 10% glycerol) for 2 h at room temperature. His-tagged Sic1$^{PY}$ conjugates (polyubiquitylated Sic1$^{PY}$, $Ub_n$–Sic1$^{PY}$) were purified by incubating with Ni-NTA resin (Qiagen) at 4 °C for 1 h. Afterwards the resin was washed with 20 column volumes of the wash buffer (50 mM Tris-HCl (pH 7.5), 100 mM NaCl, 10% glycerol). The $Ub_n$–Sic1$^{PY}$ was eluted with the same buffer containing 150 mM imidazole, and finally exchanged to the storage buffer (50 mM Tris-HCl (pH 7.5), 100 mM NaCl, 10% glycerol) using an Amicon ultrafiltration device with 30K molecular cut-off (Millipore).

### Expression and purification of human RPN13

To purify human RPN13, pGEX-4T-RPN13-transformed BL21-CondonPlus (DE3)-RIPL cells were cultured to an $OD_{600}$ of 0.6 and then induced by 0.2 mM IPTG for 20 h at 16 °C. Cells were resuspended in lysis buffer (25 mM Tris-HCl (pH 7.5), 300 mM NaCl, 1 mM EDTA, 0.2% Triton-X-100, 1 mM DTT) containing 1× protease inhibitor cocktail and lysed by sonication. A 20,000$g$ supernatant was incubated with glutathione Sepharose 4B resin (GE Healthcare) for 3 h at 4 °C. The resin was washed with 20 column volumes of washing buffer (25 mM Tris-HCl (pH 8.0), 150 mM NaCl, 1 mM DTT, 10% glycerol) and 10 column volumes of cleavage buffer (20 mM Tris-HCl (pH 8.0), 150 mM NaCl). The GST tag was cleaved by incubating with thrombin (Sigma) overnight at 4 °C. The eluted samples were further purified by gel-filtration column (Superdex 75, GE Healthcare) equilibrated with 25 mM Tris-HCl (pH 8.0), 150 mM NaCl, 1 mM DTT, 10% glycerol.

### In vitro degradation assay

Purified human proteasomes (~30 nM) were incubated with RPN13 (~300 nM), $Ub_n$–Sic1$^{PY}$ (~300 nM) in degradation buffer (50 mM Tris-HCl (pH 7.5), 5 mM $MgCl_2$ and 5 mM ATP) at 37 °C. Purified recombinant USP14 variants (~1.2 μM) were incubated with proteasome for 20 min at room temperature before initiating the degradation reaction. The reaction mixtures were incubated at 37 °C for 0, 0.5, 1.0 and 2.0 min, or 10 °C for 0, 0.5, 1.0, 5.0, 10 and 30 min, then terminated by adding SDS loading buffer and subsequently analysed by western blot using anti-T7 antibody (Abcam, 1:1,000 dilution), which was used to examine fusion protein T7–Sic1$^{PY}$.

### Ubiquitin–AMC hydrolysis assay

Ubiquitin–AMC (Ub–AMC; Boston Biochem) hydrolysis assay was used to quantify the deubiquitylating activity of wild-type and mutant USP14 in the human proteasome. The reactions were performed in reaction buffer (50 mM Tris-HCl (pH 7.5), 5 mM $MgCl_2$, 1 mM ATP, 1 mM DTT, 1 mM EDTA and 1 mg ml$^{-1}$ ovalbumin (Diamond)), containing 1 nM Ub–VS-treated proteasome, 0.2 μM USP14 variants and 10 nM RPN13. The reaction was initiated by adding 1 μM Ub–AMC. Ub–AMC hydrolysis was measured in a Varioskan Flash spectral scanning multimode reader (Thermo Fisher) by monitoring an increase of fluorescence excitation at 345 nm with an emission at 445 nm. For free USP14 activity, the reaction was performed using 1 μM USP14 variants and 1 μM Ub–AMC (BioVision).

### ATPase activity assay

ATPase activity was quantified using malachite green phosphate assay kits (Sigma). Human proteasomes (30 nM), RPN13 (300 nM) and USP14 variants (1.2 μM) were incubated in assembly buffer (50 mM Tris-HCl (pH 7.5), 5 mM MgCl$_2$ and 0.5 mM ATP) for 20 min at room temperature. Ub$_n$–Sic1$^{PY}$ (300 nM) was subsequently added, and the sample was incubated for 1 min at 37 °C. The reaction mixtures were mixed with malachite green buffers as described by the manufacturer (Sigma). After 30 min of room temperature incubation, the absorbance at 620 nm was determined using a Varioskan Flash spectral scanning multimode reader (Thermo Fisher).

### Peptidase activity assay

Peptide hydrolysis by the human proteasomes was measured using fluorogenic substrate Suc-LLVY-AMC (MCE). Human proteasomes (1 nM) were incubated with USP14 variants (1 μM) in buffer (50 mM Tris-HCl (pH 7.5), 100 mM KCl, 0.5 mM MgCl$_2$, 0.1 mM ATP and 25 ng μl$^{-1}$ BSA) for 20 min at room temperature. 10 μM Suc-LLVY-AMC was added to the reaction mixture, which was incubated for 30 min at 37 °C in the dark. Peptide activity was measured in a Varioskan Flash spectral scanning multimode reader (Thermo Fisher) by excitation at 380 nm with an emission at 460 nm.

### Microscale thermophoresis

The human proteasomes were labelled with red fluorescent dye NT-650-NHS using the Monolith NT Protein Labeling Kit (NanoTemper). After labelling, excess dye was removed by applying the sample on column B (provided in the kit) equilibrated with reaction buffer (50 mM Tris-HCl (pH 7.5), 5 mM MgCl$_2$ and 1 mM ATP). 0.05% Tween-20 was added to the sample before MST measurements. For interaction of NT-650-NHS-labelled proteasomes with USP14, USP14-UBL or USP14-USP, concentration series of USP14, USP14-UBL or USP14-USP were prepared using a 1:1 serial dilution of protein in reaction buffer containing 0.05% Tween-20. The range of USP14, USP14-UBL or USP14-USP concentration used began at 8 μM, with 16 serial dilution in 10-μl aliquots. The interaction was initiated by the addition of 10 μl of 30 nM NT-650-NHS-labelled proteasomes to each reaction mixture and measured by Monolith NT.115 (NanoTemper) at 20% LED excitation power and 40% MST power. To evaluate the effect of Ub$_n$–Sic1$^{PY}$ on the interaction of USP14 with the proteasome, 30 nM Ub$_n$–Sic1$^{PY}$ was added to the reaction mixture. Data were analysed using MO Control software provided by NanoTemper.

### Cryo-EM sample preparation

To prepare cryo-EM samples, all purified proteins were exchanged to imaging buffer (50 mM Tris-HCl (pH 7.5), 5 mM MgCl$_2$ and 1 mM ATP). Human proteasomes (1 μM) were incubated with 10 μM RPN13, 10 μM USP14 in imaging buffer (50 mM Tris-HCl (pH 7.5), 5 mM MgCl$_2$ and 1 mM ATP) for 20 min at 30 °C, then cooled to 10 °C. 10 μM Ub$_n$–Sic1$^{PY}$ was added to the mixture and incubated at 10 °C for 0.5, 1, 5 and 10 min. 0.005% NP-40 was added to the reaction mixture immediately before cryo-plunging. Cryo-grids made without the addition of substrate corresponded to the condition of 0 min of reaction time and were used as a baseline control for time-resolved analysis (Fig. 1e). For ATP-to-ATPγS exchange and ATPase quenching, after the reaction mixture was incubated at 10 °C for 1 min, 1 mM ATPγS was added to the reaction mixture at once, and incubated for another 1 min, then NP-40 was added to the mixture to a final concentration of 0.005% before cryo-plunging.

### Cryo-EM data collection

The cryo-grids were initially screened in a 200 kV Tecnai Arctica microscope (Thermo Fisher). Good-quality grids were then transferred to a 300 kV Titan Krios G2 microscope (Thermo Fisher) equipped with the post-column BioQuantum energy filter (Gatan) connected to a K2

Summit direct electron detector (Gatan). Coma-free alignment and parallel illumination were manually optimized prior to each data collection session. Cryo-EM data were acquired automatically using SerialEM software[45] in a super-resolution counting mode with 20 eV energy slit, with the nominal defocus set in the range of −0.8 to −2.0 μm. A total exposure time of 10 s with 250 ms per frame resulted in a 40-frame movie per exposure with an accumulated dose of ~50 electrons per Å$^2$. The calibrated physical pixel size and the super-resolution pixel size were 1.37 Å and 0.685 Å, respectively. For time-resolved sample conditions, 1,781, 2,298, 15,841, 2,073 and 2,071 movies were collected for cryo-grids made with the reaction time of 0, 0.5, 1, 5, and 10 min, respectively. For the condition of exchanging ATP to ATPγS at 1 min after substrate addition, 21,129 movies were collected.

### Reference structures

Comparisons to protein structures from previous publications used the atomic models in the PDB under accession codes: 2AYN (USP domain of USP14 in its isolated form[5]), 2AYO (USP domain of USP14 bound to ubiquitin aldehyde[5]), 6MSB (state E$_{A1}$ of substrate-engaged human proteasome[10]), 6MSD (state E$_{A2}$), 6MSE (state E$_B$), 6MSG (state E$_{C1}$), 6MSJ (state E$_{D1}$), 6MSK (state E$_{D2}$), 5VFT (state S$_B$ of substrate-free human proteasome[8,9]), 5VFU (state S$_C$), 5VFP (state S$_{D1}$) and 5VFR (state S$_{D3}$). Cryo-EM maps from previous publications used in comparison are available from EMDB under access codes EMD-9511 (USP14–UbAl-bound proteasome[25]), EMD-3537 (Ubp6-bound proteasome map[26]) and EMD-2995 (Ubp6–UbVS-bound proteasome[23]).

### Cryo-EM data processing

Drift correction and dose weighting were performed using the MotionCor2 program[46] at a super-resolution pixel size of 0.685 Å. Drift-corrected micrographs were used for the determination of the micrograph CTF parameters with the Gctf program[47]. Particles were automatically picked on micrographs that were fourfold binned to a pixel size of 2.74 Å using an improved version of the DeepEM program[48]. Micrographs screening and auto-picked particles checking were both preformed in the EMAN2 software[49]. A total of 213,901, 106,564, 1,494,869, 212,685, 141,257 and 1,387,530 particles were picked for the 0 min, 0.5 min, 1 min, 5 min, 10 min and ATPγS datasets, respectively. Reference-free 2D classification and 3D classification were carried out in software packages RELION[50] version 3.1 and ROME[51]. Focused 3D classification, CTF and aberration refinement, and high-resolution auto-refinement were mainly done with RELION 3.1, whereas the AlphaCryo4D software[27] was used to analyse the conformational changes and conduct the in-depth 3D classification for time-resolved analysis. Particle subtraction and re-centering were performed using RELION 3.1 and SPIDER[52] software. We applied a hierarchical 3D classification strategy to analyse the data (Extended Data Fig. 2), which were optimized as previously described[10]. The entire data-processing procedure consisted of five steps. Datasets of different conditions were processed separately at steps 1 and 2 and combined at steps 3 and 4.

Step 1: doubly capped proteasome particles were separated from singly capped ones through several rounds of 2D and 3D classification. These particles were aligned to the consensus models of the doubly and singly capped proteasome to obtain their approximate shift and angular parameters. With these parameters, each doubly capped particle was split into two pseudo-singly capped particles by re-centring the box onto the RP–CP subcomplex. Then the box sizes of pseudo-singly capped particles and true singly capped particles were both shrunk to 640 × 640 pixels with a pixel size of 0.685 Å, and down-sampled by two-fold to a pixel size of 1.37 Å for the following processing. A total of 3,429,154 particles from all datasets were obtained after this step.

Step 2: particles were aligned to the CP subcomplex through auto-refinement, followed by one round of CTF refinement to correct optical aberration (up to the fourth order), magnification anisotropy, and per-particle defocus together with per-particle astigmatism. After

another run of the CP-masked auto-refinement, an alignment-skipped RP-masked 3D classification was performed to separate the $S_A$-like states from the $S_D$-like states. Poor 3D classes showing broken 26S proteasome were removed for further analysis at this step. The RP subcomplex of the $S_D$-like states rotated by a large angle compared to the $S_A$-like states, and only in $S_D$-like states was the USP domain of USP14 observed to bind the OB ring of the proteasome. There were 1,774,110 particles in total in $S_A$-like states and 1,360,329 particles in total in $S_D$-like states in all datasets after this step.

Step 3: considering the particle number of some datasets were not enough to ensure high accuracy of independent 3D classification, in the following procedure we pooled particles together from all datasets except for the 0-min condition, in which the substrate was not yet added into the reaction system. For the $S_D$-like state, CP-masked auto-refinement was performed, followed with two rounds of CTF refinement and another run of CP-masked auto-refinement. Alignment-skipped RP-masked 3D classification was then performed, while conformational changes were analysed using AlphaCryo4D[27], which yielded three clusters, designated $S_B$-like, $S_D$-like, and $E_D$-like states. These names were correspondingly referred to their similar states in previously published studies[9,10]. The $S_A$-like particles were processed in the same way, resulting in a cluster named $E_A$-like state; bad classes showed blurred densities in RPN10 and part of the lid. The 0-min dataset was processed independently for the lack of substrate, resulting in three classes, named $S_A$-like (92.8%), $S_B$-like (4.3%) and $S_D$-like (3.0%).

Step 4: particles in different clusters were individually refined with the CP masked. The CP density was then subtracted, and the particle box was recentred to the RP subcomplex and shrunk to 240 × 240 pixels, with a pixel size of 1.37 Å. For each cluster, the CP-subtracted particles were subjected to several rounds of RP-masked auto-refinement and alignment-skipped RP-masked 3D classification followed by AlphaCryo4D analysis[27], finally resulting in thirteen major conformational states of the USP14-bound proteasome, named $E_{A1}^{UBL}$, $E_{A2.0}^{UBL}$, $E_{A2.1}^{UBL}$, $S_B^{USP14}$, $S_C^{USP14}$, $S_{D4}^{USP14}$, $S_{D5}^{USP14}$, $E_{D0}^{USP14}$, $E_{D1}^{USP14}$, $E_{D2.0}^{USP14}$, $E_{D2.1}^{USP14}$, $E_{D4}^{USP14}$ and $E_{D5}^{USP14}$. For state $E_{A1}^{UBL}$, particles with blurred RPN1 were excluded for final high-resolution reconstruction. For state $E_{D2.1}^{USP14}$, particles with blurred RPN2 were excluded for final high-resolution reconstruction. These states exhibit remarkable conformational changes of the AAA ring and the full RP, as well as the interactions of the RP and USP14. Time-resolved analysis of conformational changes and comparison in the presence and absence of ATPγS were both done after this step, by simply separating the particles for each state based on their time labels. Namely, the proportion of particles of each state at a given time point was obtained by summing up the number of particles for each state at the same time point and then calculating the fraction of particles of each state with respect to the total number of particles at this time point[53,54]. Similarly, final analysis of state percentage for the ATP-to-ATPγS exchange condition was done by counting the particles of each state under this condition, with the particles of each state used for separate refinement, reconstruction and comparison with those under ATP-only conditions (Extended Data Figs. 2c, 5a, 6i).

Final refinement of each state was performed using pseudo-singly capped particles with the pixel size of 1.37 Å. Two types of local mask were applied for the auto-refinement, one focusing on the complete RP and the other focusing on the CP, resulting in two maps for each state, which were merged in Fourier space into one single map. Based on the in-plane shift and Euler angle of each particle from the auto-refinement, we reconstructed the two half-maps of each state using pseudo-singly capped particles with the pixel size of 0.685 Å. The Fourier shell correlation (FSC) curves of thirteen states were calculated from two separately refined half maps in a gold-standard procedure, yielding the nominal resolution ranging from 3.0 to 3.6 Å, and the local RP resolution ranging from 3.3 to 4.6 Å (Extended Data Figs. 2a, 3b–d). Before visualization, all density maps were sharpened by applying a negative B-factor ranging from −10 to −50 Å$^2$.

In order to further improve the local density quality of USP14 and RPN1, which suffered from local conformational dynamics, another round of RP-masked 3D classification was done using CP-subtracted particles for some states to exclude 3D classes with blurred USP14 and RPN1. These locally improved maps were only used for visualization and adjustment of atomic models of USP14 and RPN1. For states $E_{A1}^{UBL}$, $E_{A2.0}^{UBL}$ and $E_{A2.1}^{UBL}$, 3D classes with unblurred RPN1 and especially visible UBL on the RPN1 T2 site (including 152,802, 66,966 and 61,930 particles, respectively) were selected and refined by applying a mask on the RPN1-UBL component. The resulting RPN1-UBL density in these states were compared with previously reported $E_{A1}$ state (Extended Data Fig. 7e). For states $E_{D0}^{USP14}$ and $E_{D2.0}^{USP14}$, 3D classes with unblurred RPN1 density (including 61,447 and 53,145 particles, respectively) were selected and refined to 4.1 and 4.2 Å for the RP, respectively. For state $S_C^{USP14}$, $E_{D4}^{USP14}$ and $E_{D2.1}^{USP14}$, 3D classes with improved USP14 densities (including 34,659, 54,642, 142,814 particles, respectively) were selected and refined to 4.5, 4.2 and 3.8 Å for the RP component, respectively, for better visualization of the full-length USP14 in the proteasome.

### Atomic model building and refinement

Atomic model building was based on the previously published cryo-EM structures of the human proteasome[10]. For the CP subcomplex, initial models of the closed-gate CP and open-gate CP were respectively derived from the $E_{A1}$ model (PDB 6MSB) and the $E_{D2}$ model (PDB 6MSK). For the RP subcomplex, the previous $E_{D2}$ model was used as an initial model. All subunits of the initial model were individually fitted as a rigid body into each of the reconstructed maps with UCSF Chimera[55], followed by further adjustment of the main chain traces using Coot[56]. Initial model of the full-length USP14 was first derived from a predicted one by AlphaFold[57], which was verified by comparing to a crystal structure[5] (PDB 2AYO). The USP14 model was then merged with the initial proteasome model by independently fitting models of the USP14 UBL and USP domains as rigid bodies into the cryo-EM maps, and manually fitting the linker between the UBL and USP domains in Coot[56]. Despite the presence of RPN13 in our purified proteasome (Extended Data Fig. 1g) and the addition of excessive RPN13 to saturate the proteasome, no reliable density was observed for RPN13 in all cryo-EM maps, thus precluding the atomic modelling of RPN13 and likely reflecting its highly dynamic association with the proteasome. The atomic models of subunit SEM1 and the N terminus of subunit RPN5 fitted into their corresponding local densities of lower resolution were rebuilt by considering the prediction of AlphaFold[57]. The atomic model of USP14 was first rebuilt and refined against the map of state $E_{D2.1}^{USP14}$ with improved local USP14 density, the resulting model of which was then used as starting USP14 models to fit into the USP14 densities in other states. For some structures with partially blurred or missing subunit densities, the atomic models were revised by removing these regions, for example, the UBL domain of USP14 was removed in the models of $E_{D5}^{USP14}$ and $S_{D5}^{USP14}$. Given that the substrates were not stalled in a homogeneous location during their degradation and that substrate translocation through the proteasome is not sequence-specific, the substrate densities were modelled using polypeptide chains without assignment of amino acid sequence. For states $E_{A1}^{UBL}$, $E_{A2.0}^{UBL}$, $E_{A2.1}^{UBL}$, $E_{D0}^{USP14}$, $E_{D1}^{USP14}$, $E_{D2.0}^{USP14}$, $E_{D2.1}^{USP14}$ and $E_{D4}^{USP14}$, the nucleotide densities are of sufficient quality for differentiating ADP from ATP, which enabled us to build the atomic models of ADP and ATP into their densities (Extended Data Fig. 6j). For other states with the local RP resolution worse than 4.0 Å, the nucleotide types or states were hypothetically inferred from their adjacent states at higher resolution with the closest structural similarity, based on the local densities, the openness of corresponding nucleotide-binding pockets as well as their homologous structural models of higher resolution if available.

After manually rebuilding, atomic models were all subjected to the real-space refinement in Phenix[58]. Both stimulated annealing and global minimization were applied with non-crystallographic symmetry (NCS),

rotamer and Ramachandran constraints. Partial rebuilding, model correction and density-fitting improvement in Coot[56] were then iterated after each round of atomic model refinement in Phenix[58]. The refinement and rebuilding cycle were often repeated for three rounds or until the model quality reached expectation (Extended Data Table 1).

## Structural analysis and visualization

All structures were analysed in Coot[56], PyMOL[59], UCSF Chimera[55], and ChimeraX[60]. Inter-subunit interactions and interfacial areas were computed and analysed using the PISA server[61] (https://www.ebi.ac.uk/pdbe/prot_int/pistart.html). Local resolution variations were estimated using ResMap[62]. Figures of structures were plotted in PyMOL[59], ChimeraX[60], or Coot[56]. Structural alignment and comparison were performed in PyMOL[59] and ChimeraX[60].

## Data reporting

No statistical methods were used to predetermine sample size. The experiments were not randomized, and investigators were not blinded to allocation during experiments and outcome assessment.

## Statistical analysis

Statistical analyses of mutagenesis data were performed using two-tailed unpaired $t$-tests with SPSS v.27.0 unless otherwise indicated. Statistical significance was assessed with a 95% confidence interval and a $P$ value of < 0.05 was considered significant.

## Reporting summary

Further information on research design is available in the Nature Research Reporting Summary linked to this paper.

## Data availability

Cryo-EM density maps of USP14–proteasome complexes resolved in this study have been deposited in the Electron Microscopy Data Bank (EMDB) (www.emdataresource.org) under accession codes EMD-32272 ($E_{A1}^{UBL}$), EMD-32273 ($E_{A2.0}^{UBL}$), EMD-32274 ($E_{A2.1}^{UBL}$), EMD-32275 ($E_{D4}^{USP14}$), EMD-32276 ($E_{D5}^{USP14}$), EMD-32277 ($E_{D0}^{USP14}$), EMD-32278 ($E_{D1}^{USP14}$), EMD-32279 ($E_{D2.0}^{USP14}$), EMD-32280 ($E_{D2.1}^{USP14}$), EMD-32281 ($S_{B}^{USP14}$), EMD-32282 ($S_{C}^{USP14}$), EMD-32283 ($S_{D4}^{USP14}$), EMD-32284 ($S_{D5}^{USP14}$), EMD-32285 ($E_{A1}^{UBL}$ with the local RPN1 density improved), EMD-32286 ($E_{A2.0}^{UBL}$ with the local RPN1 density improved), EMD-32287 ($E_{A2.1}^{UBL}$ with the local RPN1 density improved), EMD-32288 ($E_{D5}^{USP14}$ with the USP14 density improved), EMD-32289 ($E_{D0}^{USP14}$ with the RPN1 density improved), EMD-32290 ($E_{D1}^{USP14}$ with the RPN1 density improved), EMD-32291 ($E_{D2.1}^{USP14}$ with the USP14 density improved) and EMD-32292 ($S_{C}^{USP14}$ with the USP14 density improved). The corresponding coordinates have been deposited in the Protein Data Bank (PDB) (https://www.pdb.org) under accession codes 7W37 ($E_{A1}^{UBL}$), 7W38 ($E_{A2.0}^{UBL}$), 7W39 ($E_{A2.1}^{UBL}$), 7W3A ($E_{D4}^{USP14}$), 7W3B ($E_{D5}^{USP14}$), 7W3C ($E_{D0}^{USP14}$), 7W3F ($E_{D1}^{USP14}$), 7W3G ($E_{D2.0}^{USP14}$), 7W3H ($E_{D2.1}^{USP14}$), 7W3I ($S_{B}^{USP14}$), 7W3J ($S_{C}^{USP14}$), 7W3K ($S_{D4}^{USP14}$) and 7W3M ($S_{D5}^{USP14}$). Uncropped versions of all gels and blots are provided in Supplementary Fig. 1. All other data are available from the corresponding author upon reasonable request. Source data are provided with this paper.

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

**Acknowledgements** We thank Y. Saeki for the plasmids expressing Sic1[PY] and WWHECT; L. Huang for the proteasome-expressing cell lines; C. Tang for the plasmid expressing UBE1; X. Li, Z. Guo, X. Pei, B. Shao, G. Wang, Y. Ma, C. Fan, B. Liu and L. Lai for technical assistance; and A. Goldberg and Q. Ouyang for discussions and comments. This work was funded in part by grants from the Beijing Natural Science Foundation (Z180016/Z18J008) and the National Natural Science Foundation of China (11774012, 12125401) to Y.M., and by a grant from the National Institute of Health (R01 GM043601) to D.F. The cryo-EM data were collected at the Cryo-EM Core Facility Platform and Laboratory of Electron Microscopy at Peking University. The data processing was performed in the High-Performance Computing Platform, the Weiming No. 1 and Life Science No. 1 supercomputing systems, at Peking University.

**Author contributions** Y.M. conceived this study. S. Zou purified the protein complexes and prepared the cryo-EM samples. S. Zou and L.Z. conducted the biochemical experiments and mutagenesis study. S. Zhang, S. Zou, and D.Y. collected cryo-EM data and analysed the experimental cryo-EM datasets. S. Zhang refined the density maps. S. Zhang and Y.M. built and refined the atomic models. D.F. and Z.W. contributed to the analysis of the data. Y.M. supervised this study, analysed the data and wrote the manuscript with input from all authors.

**Competing interests** The authors declare no competing interests.

**Additional information**
**Correspondence and requests for materials** should be addressed to Youdong Mao.

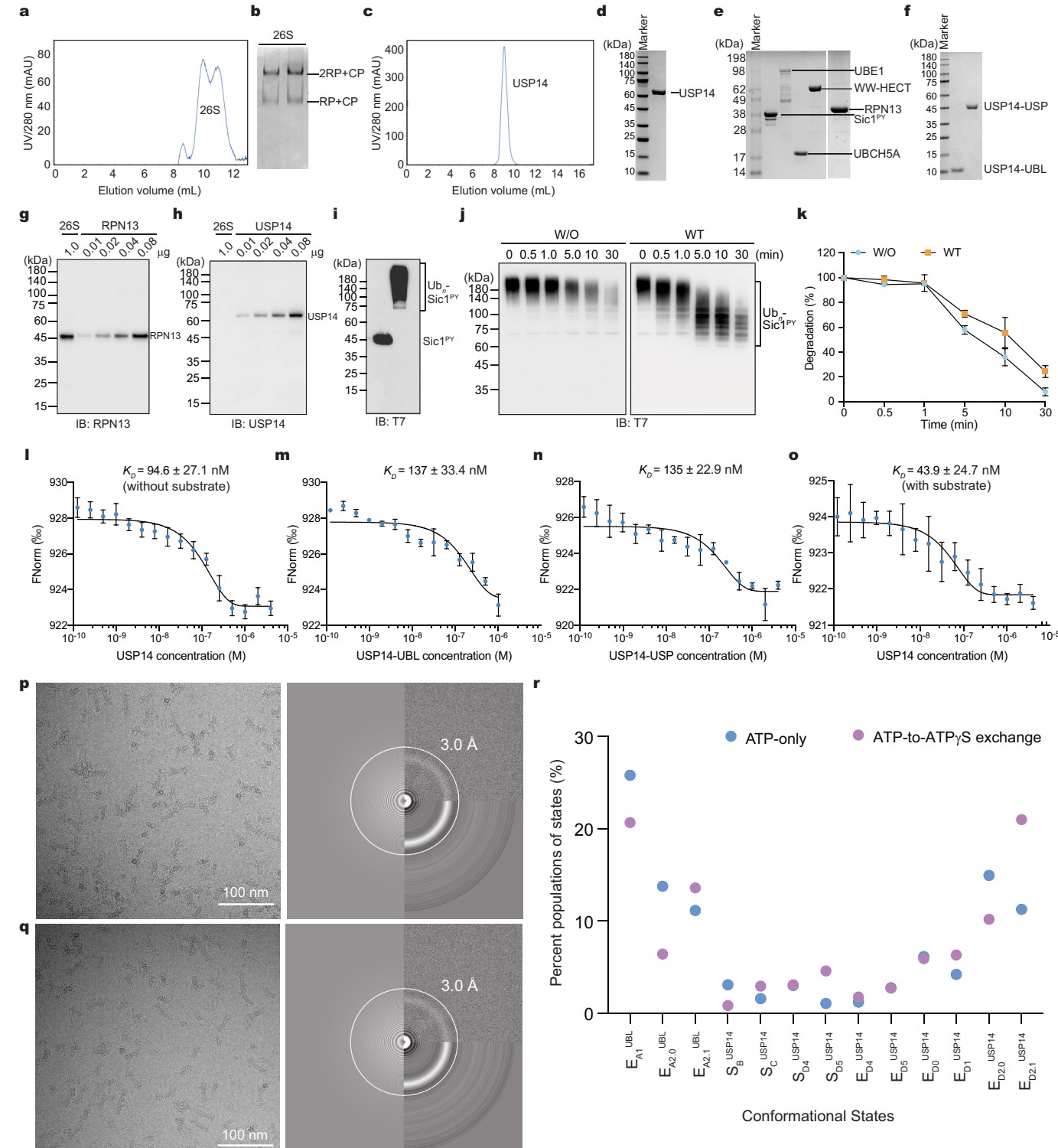

**Extended Data Fig. 1** | See next page for caption.

**Extended Data Fig. 1 | Protein purification and cryo-EM imaging. a**, The human 26S proteasome was purified through gel-filtration column (Superose 6 10/300 GL). **b**, Native gel analysis of the human 26S proteasome from (**a**). **c**, FPLC purification of human USP14 on Superdex 75 10/300 GL column. **d**–**f**, SDS-PAGE and Coomassie blue stain analysis of purified USP14 (**d**), Sic1$^{PY}$, UBE1, UBCH5A, WW-HECT, human RPN13 (**e**), USP14 UBL and USP domains (**f**). **g** and **h**, Western blot was used to evaluate the content of RPN13 (**g**) and USP14 (**h**) in the purified human proteasomes. The results indicate the presence of RPN13 and the absence of USP14 in the purified human 26S proteasome. **i**, Western blot was used to verify polyubiquitylation of Sic1$^{PY}$ (Ub$_n$-Sic1$^{PY}$) using anti-T7 antibody, indicating that most Sic1$^{PY}$ was ubiquitylated. **j**, *In vitro* degradation of Ub$_n$-Sic1$^{PY}$ by the purified 26S proteasome at 10 °C in the absence and presence of USP14. The concentration and ratio of each component was same as cryo-EM sample preparation. The experiments were repeated three times. Samples in (**i**) and (**j**) were analyzed by SDS–PAGE/ Western blot using anti-T7 antibody. W/O, the proteasome without binding to USP14. WT, the wildtype USP14-bound proteasome. **k**, Kinetics of Ub$_n$-Sic1$^{PY}$ degradation was plotted by measuring Ub$_n$-Sic1$^{PY}$ density in (**j**) using ImageJ software. Each point is representative of three independent experiments. Data are presented as mean ± s.d. **l**–**o**, MST analysis of USP14 binding to the human proteasome. A dissociation constant of 94.6 ± 27.1 nM (**l**, full-length USP14 in the absence of Ub$_n$-Sic1$^{PY}$), 137 ± 33.4 nM (**m**, USP14 UBL domain only), 135 ± 22.9 nM (**n**, USP14 USP domain only) and 43.9 ± 24.7 nM (**o**, full-length USP14 in the presence of Ub$_n$-Sic1$^{PY}$) were calculated from three independent experiments (shown as mean ± s.d.). **p** and **q**, Typical motion-corrected cryo-EM micrographs (left) of the substrate-engaged human USP14–proteasome complex in the presence of 1 mM ATP (**p**) or after ATP-to-ATPγS exchange (**q**). Power spectrum evaluation of the corresponding micrographs are shown on the right. The exact numbers of micrographs collected under different experimental conditions are provided in Extended Data Fig. 2a. Each experiment was repeated independently at least five times with similar results. **r**, Comparison of percent population of each conformational state in the presence of 1 mM ATP or ATP-to-ATPγS exchange in 1 min after mixing the substrate with the USP14-bound proteasome. Our cryo-EM analysis suggests that ATP-to-ATPγS exchange enriches states $E_{A2.1}^{UBL}$, $S_C^{USP14}$, $S_{D5}^{USP14}$, $E_{D1}^{USP14}$, $E_{D2.1}^{USP14}$ and $E_{D4}^{USP14}$, but reduces states $E_{A1}^{UBL}$, $E_{A2.0}^{UBL}$, $S_B^{USP14}$ and $E_{D2.0}^{USP14}$. The particle numbers used to derive this plot are provided in Extended Data Fig. 2c. For gel source data, see Supplementary Fig. 1.

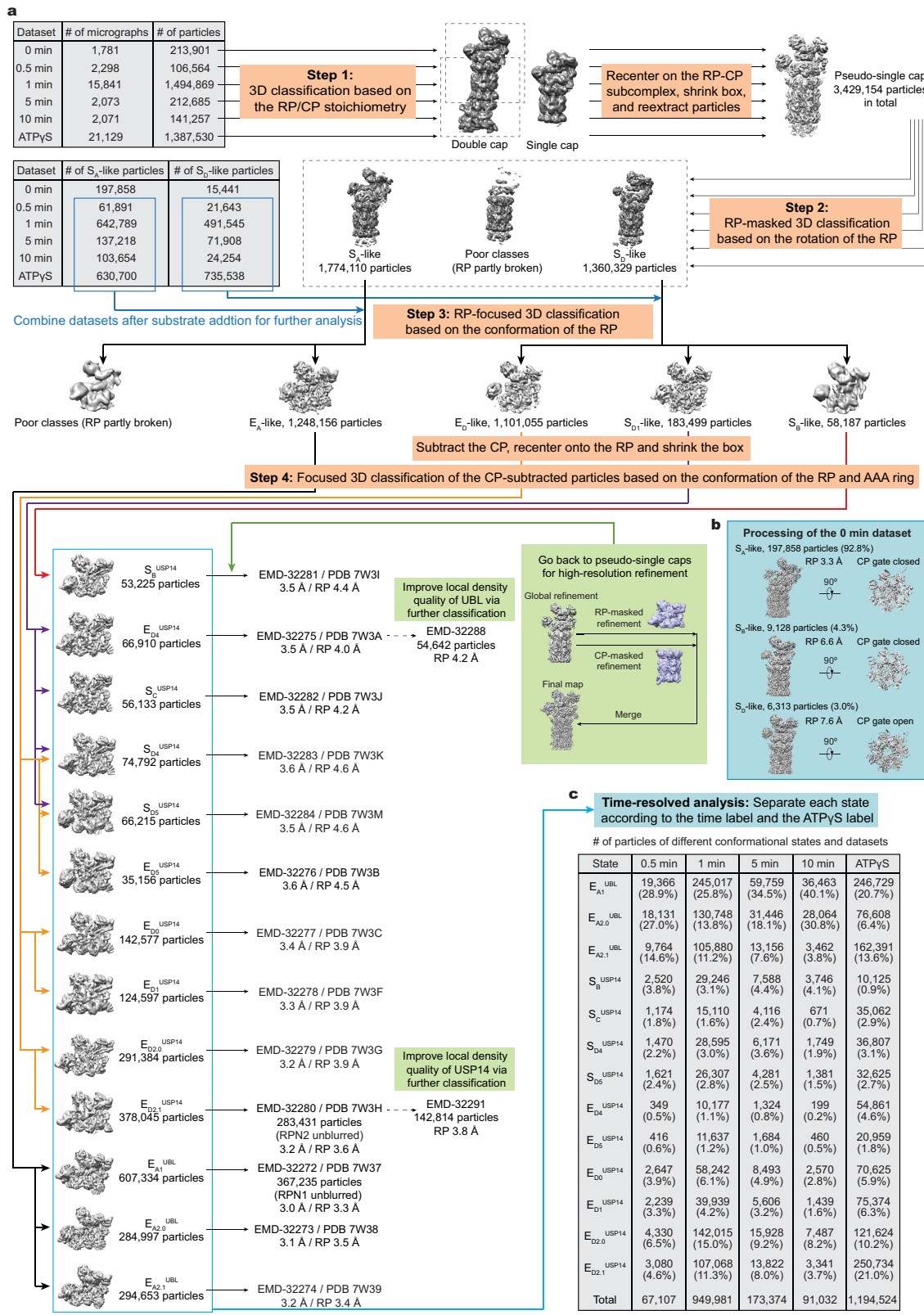

**Extended Data Fig. 2 | Cryo-EM data processing workflow and time-resolved analysis. a**, The workflow diagram illustrates the major steps of our focused 3D classification strategy for cryo-EM data analysis. Particle numbers after 3D classification and final reconstruction and the resolutions of the complete RP-CP and RP-masked reconstructions of each state are labelled.

**b**, 3D classification of the dataset corresponding to 0 minute before substrate addition as an overall control. **c**, Time-resolved analysis of all states by restoring the time label (for the buffer condition with 1 mM ATP) or ATPγS label (for the condition with ATP-to-ATPγS exchange). The percentages were computed using the total particle number corresponding to a given time point.

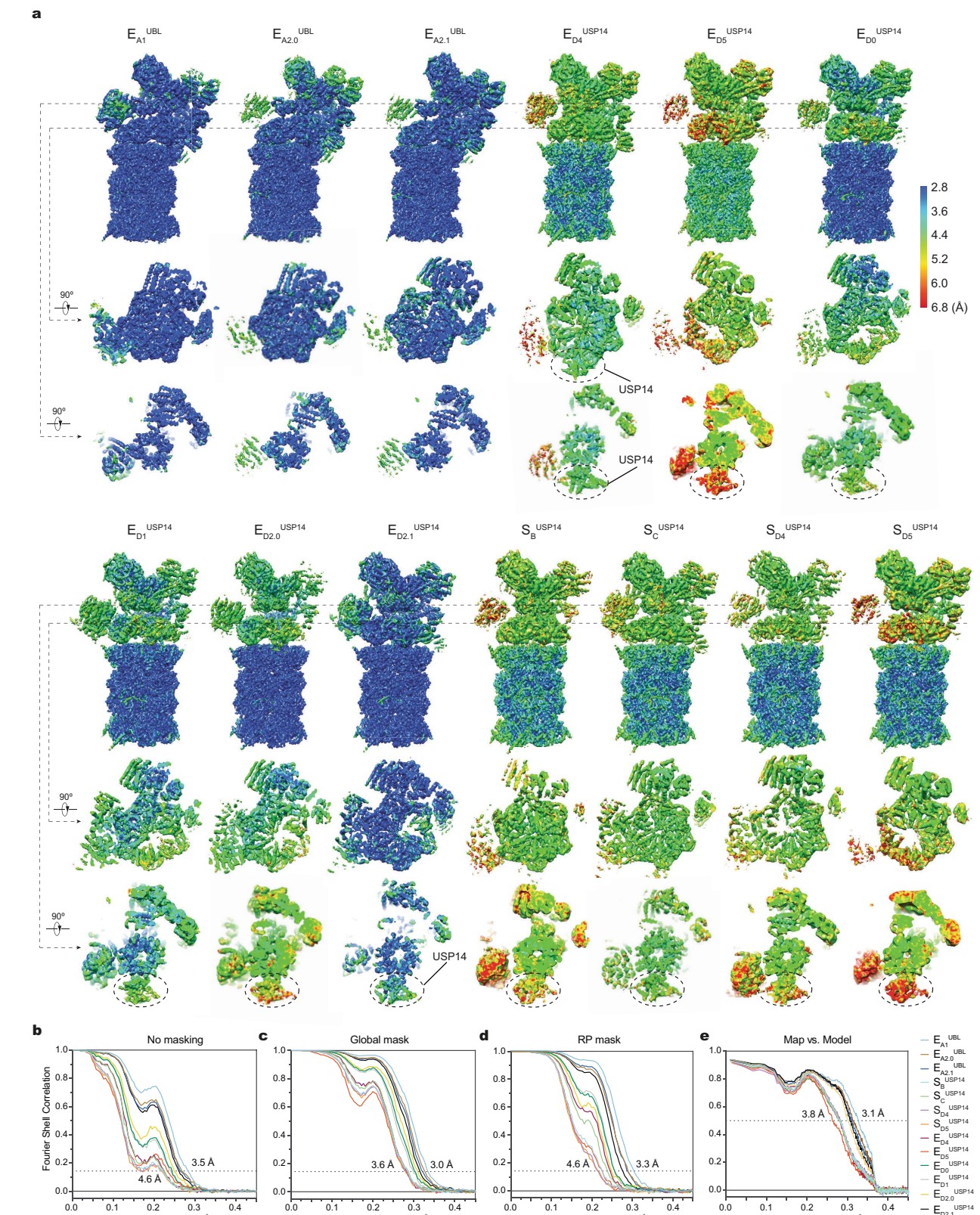

**Extended Data Fig. 3 | Cryo-EM reconstructions and resolution measurement. a**, Local resolution estimation of the RP reconstructions of thirteen states calculated by ResMap[62]. Each state is shown in two orthogonal orientations, with the second orientation (top view) shown in two different cross-sections at the AAA (middle row) and OB (lower row) domains. All local resolutions are shown with the same color bar in the upper right inset. **b** and **c**, Gold-standard Fourier shell correlation (FSC) plots of the complete RP-CP maps of all states calculated without (**b**) or with (**c**) masking the separately refined half-maps. **d**, Gold-standard FSC plots of the RP-masked reconstructions of all states. The RP maps were refined by focusing the mask on the RP subcomplex. **e**, Model-map FSC plots calculated by Phenix[58] between each RP-CP map (masked) and its corresponding atomic model. For each state, separately refined RP and CP maps (using RP and CP masks, respectively) were merged in Fourier space into a single RP-CP map, which was used for the model-map FSC calculation. The same color code is used in (**b**–**e**).

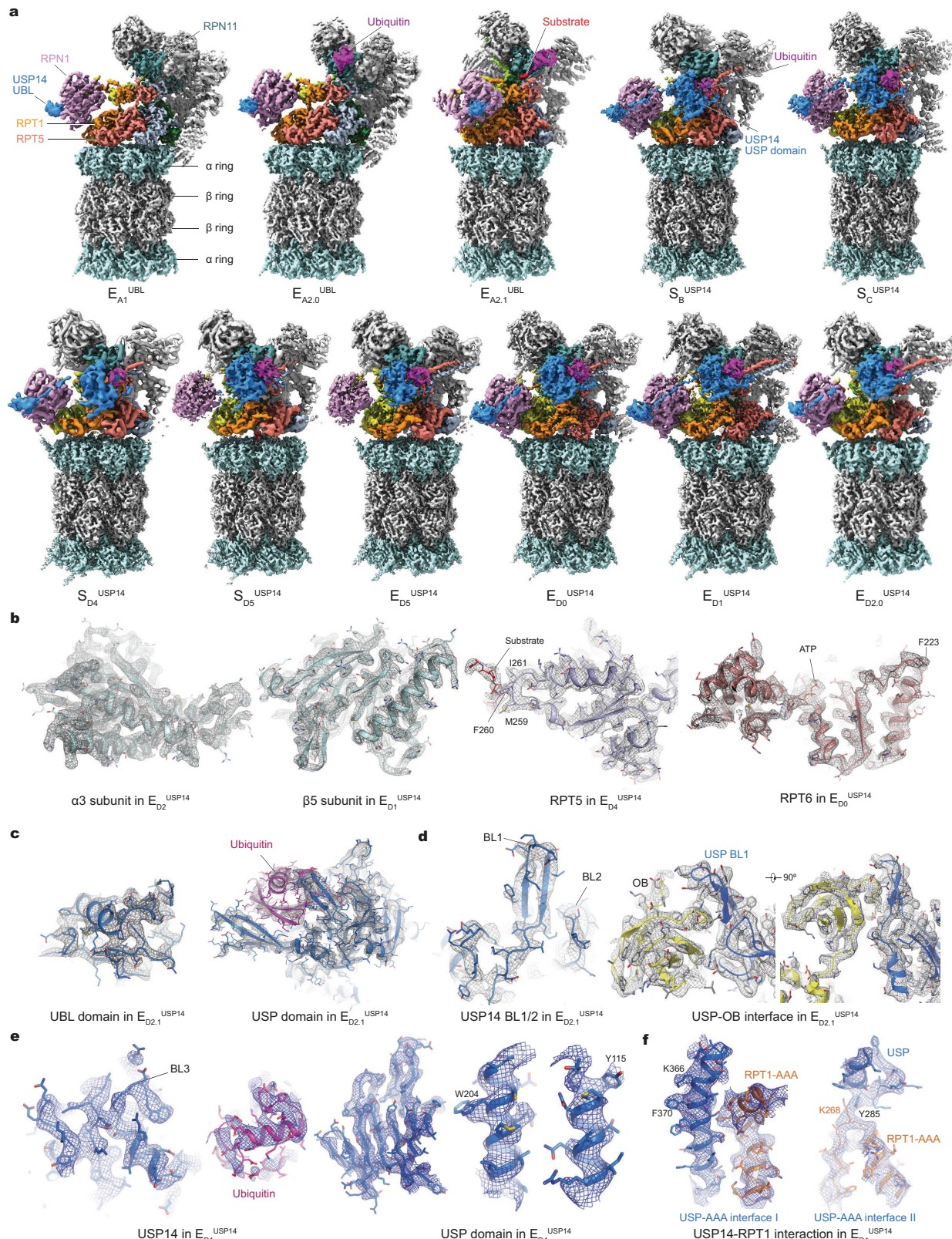

**Extended Data Fig. 4 | Cryo-EM maps and quality assessment. a**, Gallery of all refined cryo-EM maps not shown in the main figures. **b**–**f**, Typical high-resolution cryo-EM densities (mesh) of secondary structures superimposed with their atomic models. Different subunits of the proteasome are shown in (**b**), where the substrates shown in the middle panel are modelled using polypeptide chains without assignment of amino acid sequence. **c**, High-resolution cryo-EM densities of the two domains of USP14 in state $E_{D2.1}^{USP14}$. **d**, High-resolution cryo-EM densities of USP14 BL1 and BL2 loops and USP-OB interface in state $E_{D2.1}^{USP14}$. **e**, High-resolution cryo-EM densities of the BL3 loop, ubiquitin and representative secondary structure elements in the USP domain of USP14 in state $E_{D4}^{USP14}$. **f**, High-resolution cryo-EM densities of USP-AAA interfaces in state $E_{D4}^{USP14}$.

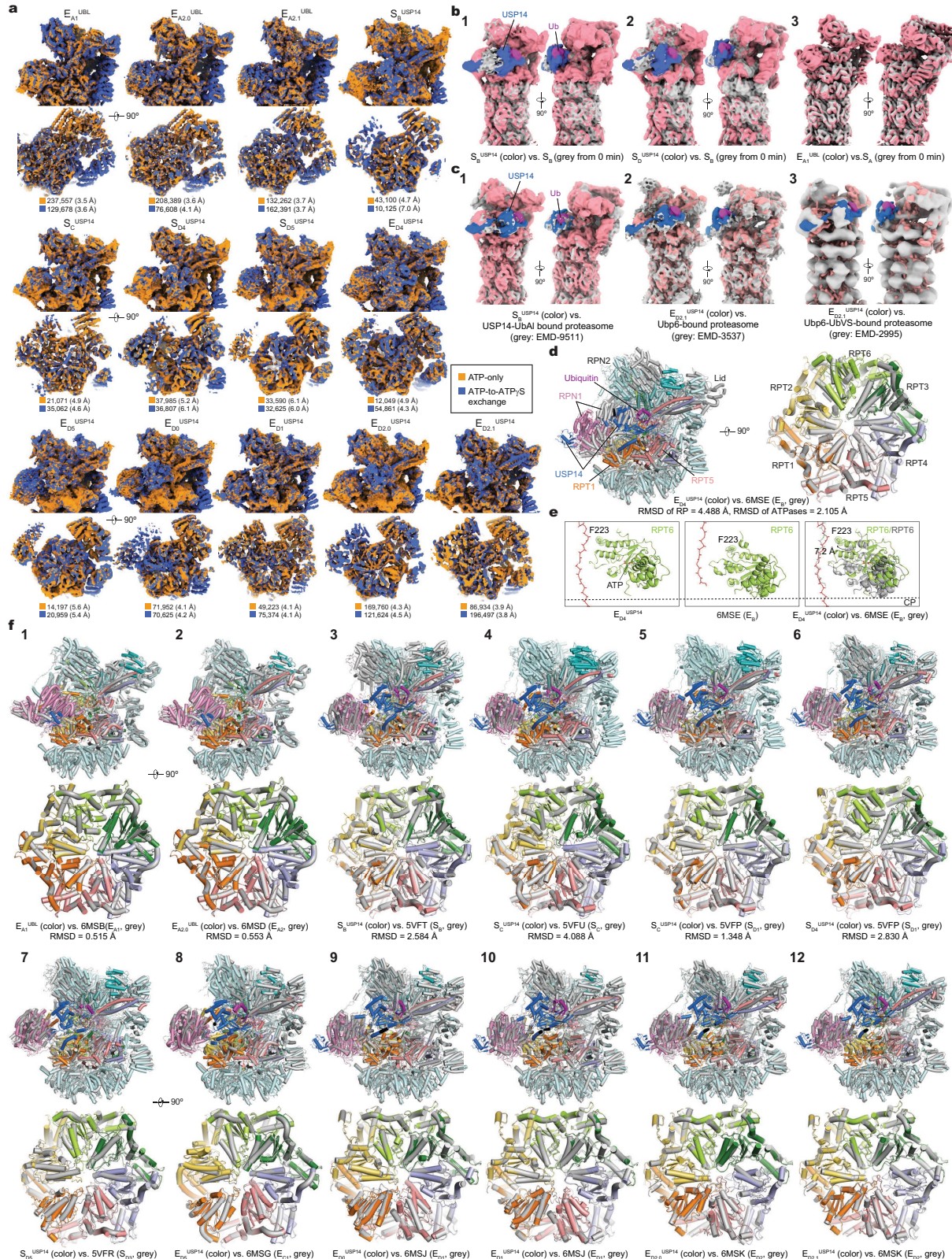

**Extended Data Fig. 5** | See next page for caption.

**Extended Data Fig. 5 | Structural comparison of the proteasome under distinct conditions. a**, Comparison of cryo-EM reconstructions of the thirteen states of USP14-proteasome complex from the conditions of including 1 mM ATP (ATP-only, no ATPγS in the degradation buffer) (orange) and with ATP-to-ATPγS exchange at 1 min after substrate addition (blue). Particle number and the RP resolution for each class are labelled below each panel of structural comparison. The results show that consistent features of the same states were obtained from the two conditions, although the N-terminal densities of RPN5 are slightly stronger under the condition with ATP-to-ATPγS exchange. **b**, Comparison of the three major states at 0 min (before substrate addition and after USP14 was bound to the proteasome) with those after the substrate was mixed with the USP14-proteasome complex. For visual clarity, all maps were low-pass filtered to 8 Å. **c**, Comparison of state $S_B^{USP14}$ with the USP14-UbAl-bound proteasome map EMD-9511 (panel **1**)[25] and of state $E_{D2.1}^{USP14}$ with the Ubp6-bound proteasome map EMD-3537 (panel **2**)[26] and Ubp6-UbVS-bound proteasome map EMD-2995 (panel **3**)[23]. Due to the low-resolution nature of the previously published maps, the cryo-EM maps of states $S_B^{USP14}$ and $E_{D2.1}^{USP14}$ were low-pass filtered to 8 Å for the visual clarity of comparison. **d**, Structural comparison of state $E_{D4}^{USP14}$ with state $E_B$ (6MSE) suggests that they have comparable ATPase conformations with defined differences in RPT6 (shown in panel **e**), whereas the lid subcomplexes are in very different conformations with large rotations. **e**, Structural comparison of RPT6 in states $E_{D4}^{USP14}$ and $E_B$ shows that the RPT6 pore-1 loop, highlighted by transparent sphere representation of Phe223, is moved about 7.2 Å toward the substrate in state $E_{D4}^{USP14}$ relative to state $E_B$. The right panel show the two RPT6 structures superimposed after aligning the entire ATPase motor subcomplex structures together (as shown in the right of panel **d**). **f**, Comparison of the RP and ATPase structures in different states and previously published cryo-EM structures[8–10]. These structures are aligned together against their CPs. Each pair of compared structures are shown in two orthogonal orientations, with the root-mean-squared-deviation (RMSD) values for the ATPase components shown below each panel of structural comparison. Previous USP14-free structures (PDB ID) used for the comparison include substrate-free states $S_B$ (5VFT), $S_C$ (5VFU), $S_{D1}$ (5VFP), $S_{D3}$ (5VFR) and substrate-bound states $E_{A1}$ (6MSB), $E_{A2}$ (6MSD), $E_B$ (6MSE), $E_{C1}$ (6MSG), $E_{D1}$ (6MSJ), and $E_{D2}$ (6MSK).

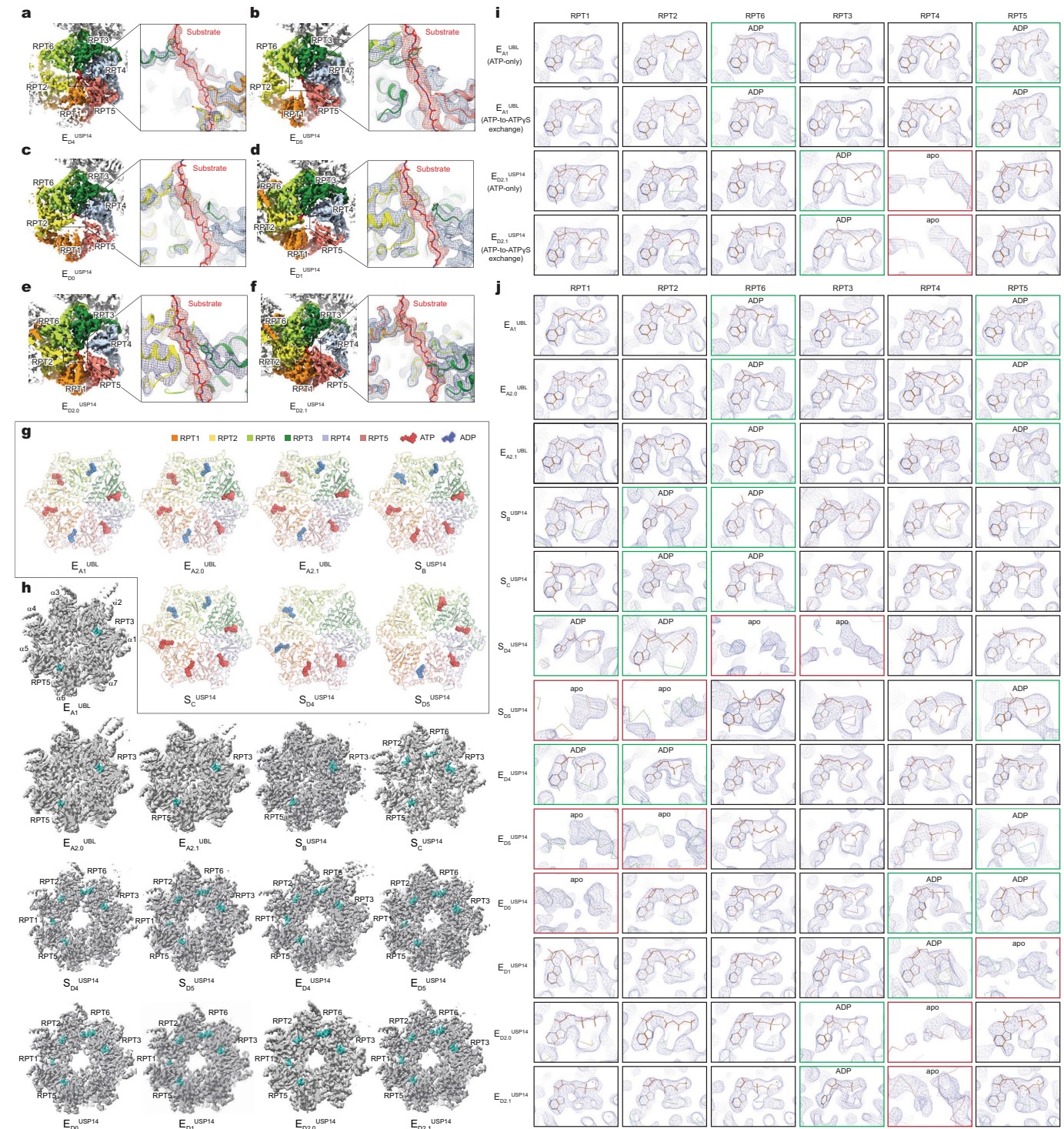

**Extended Data Fig. 6** | See next page for caption.

**Extended Data Fig. 6 | Substrate interactions with the pore loops in the AAA-ATPase motor, the RP-CP interfaces and the nucleotide densities in different USP14-bound states. a–f**, Cryo-EM densities of the ATPase motor bound to the substrate. Substrate densities are colored in red. Right insets show the zoomed-in side views of the substrate interacting with the pore loops of the ATPases. The substrate in each state is modelled as a polypeptide backbone structure and is represented with red sticks. The ATPases and substrates are rendered as surface (left) and mesh (right) representations. **g**, Top views of the ATPase motor structures of all states not shown in the main figure are shown in cartoon representations. Nucleotides are shown in sphere representations. The sphere representations of ADP and ATP are in blue and in red, respectively. The structures are aligned together against their CP components. Top right, color codes of subunits used in all panels. **h**, Comparison of the RP−CP interface and RPT C-terminal tail insertions into the α-pockets of the CP in different states. The cryo-EM densities of the CP subcomplexes are shown as grey surface representations, while the RPT C-tails are colored in teal blue. **i**, Comparison of the nucleotide densities in the cryo-EM reconstructions of USP14-bound proteasome under two different nucleotide conditions. The side-by-side comparison of the nucleotide densities in two best resolved states show that the ATP-to-ATPγS exchange did not change the observed nucleotide states under our experimental conditions relative to that only used 1 mM ATP in the degradation buffer. **j**, Comparison of the nucleotide densities in thirteen distinct states of UBP14-bound proteasome. The nucleotide densities fitted with atomic models are shown in blue mesh representation. All close-up views were directly screen-copied from Coot after atomic modelling into the density maps without modification. At the contour level commonly used for atomic modelling, the potential nucleotide densities in the apo-like subunits mostly disappear, although they can occasionally appear as partial nucleotide shapes at a much lower contour level. The states with limited local resolutions are hypothetically assigned for nucleotide types based on the densities, the openness of corresponding nucleotide-binding pockets as well as their homologous structural models of higher resolution from states with similar conformations.

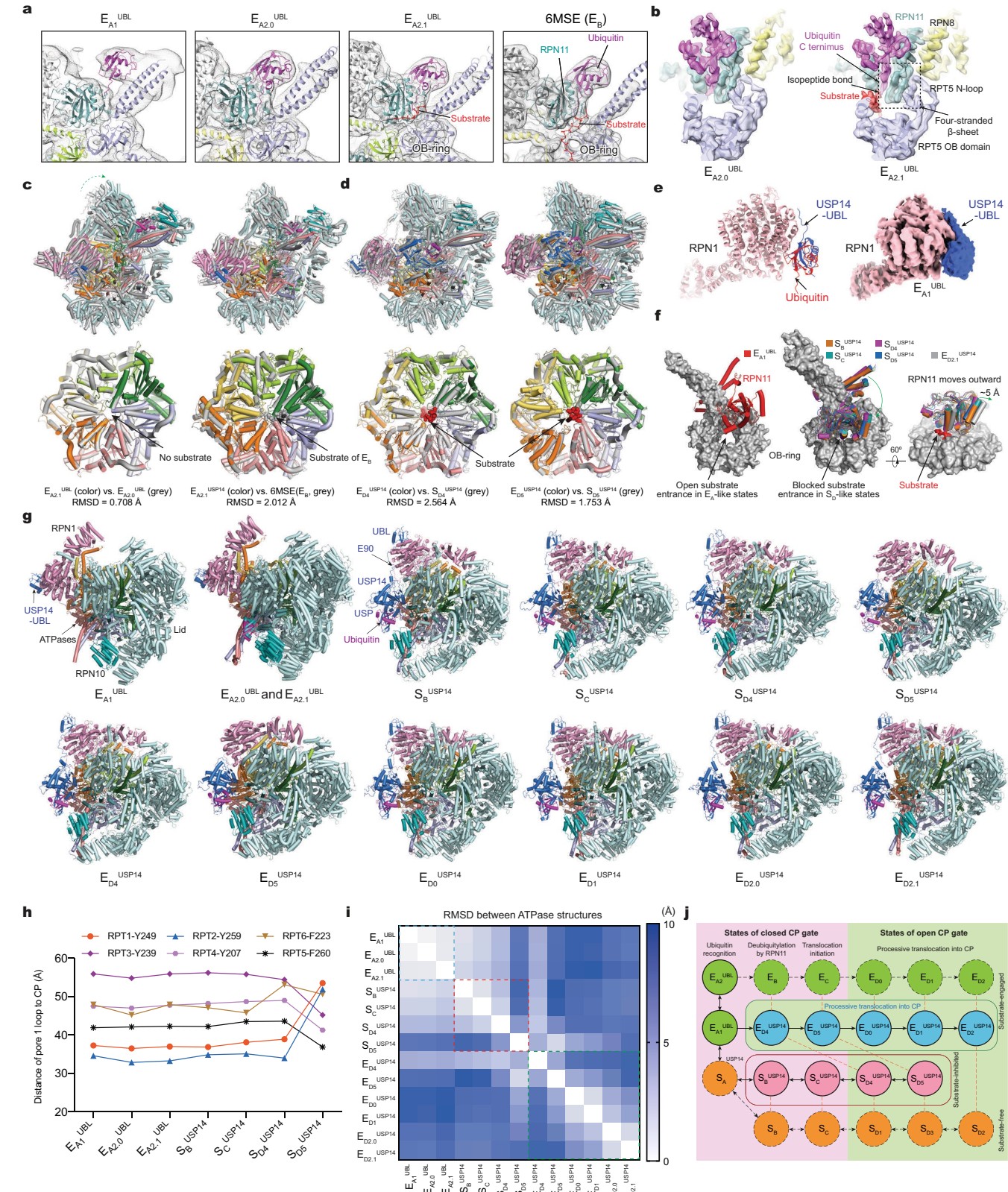

**Extended Data Fig. 7** | See next page for caption.

**Extended Data Fig. 7 | Structural comparison of the USP14-bound proteasome of different states. a**, Structural comparison of states $E_{A1}^{UBL}$, $E_{A2.0}^{UBL}$ and $E_{A2.1}^{UBL}$ shows the ubiquitin transfer from the RPT4-RPT5 coiled-coil (CC) domain to RPN11. The cryo-EM densities rendered as grey mesh representations are low-pass-filtered to 8 Å for visual clarity of comparison. **b**, Structural comparison of ubiquitin-RPN11-RPT5 interaction in state $E_{A2.0}^{UBL}$ and $E_{A2.1}^{UBL}$. Cryo-EM densities rendered transparent surfaces are superimposed with the atomic models. The overall conformation of $E_{A2.0}^{UBL}$ resides between $E_{A1}^{UBL}$ and $E_{A2.1}^{UBL}$. Both states $E_{A2.0}^{UBL}$ and $E_{A2.1}^{UBL}$ exhibited RPN11-bound ubiquitin and no substrate densities in the AAA-ATPase motor. A short stretch of ubiquitin-linked substrate density is bound to the cleft between RPN11 and the OB ring in $E_{A2.1}^{UBL}$ but not $E_{A2.0}^{UBL}$. In both states, the RPT5 N-loop (residues 99-119) pairs with one side of the Insert-1 β-hairpin of RPN11, the other side of which is paired with the C terminus of ubiquitin. They together form a four-stranded β-sheet, a feature that was previously visualized at atomic level only in state $E_B$ of the USP14-free proteasome[10]. **c**, Structural comparison of the RP and ATPase between $E_{A2.0}^{UBL}$ and $E_{A2.1}^{UBL}$ and between $E_{A2.1}^{UBL}$ and $E_B$ (PDB ID 6MSE). **d**, Structural comparison of the RP and ATPase between $E_{D4}^{USP14}$ and $S_{D4}^{USP14}$ and between $E_{D5}^{USP14}$ and $S_{D5}^{USP14}$. The root-mean-squared-deviation (RMSD) values for the ATPase components are shown below each panel of structural comparison in (**c**) and (**d**). **e**, Binding of UBL of USP14 to the T2 site of RPN1 in state $E_{A1}^{UBL}$ as compared to the binding of ubiquitin to RPN1 in USP14-free state $E_{A1}$ from previous studies[10]. Right inset shows the cryo-EM density of UBL-bound RPN1 in $E_{A1}^{UBL}$. **f**, Comparison of the RPN11-OB ring interface in different states. Left panel, the interface in the $E_A$-like states shows an open OB ring for substrate entrance. Middle panel, the substrate entrance of the OB ring is blocked by RPN11 in the substrate-inhibited states. Right panel, RPN11 is rotated outward slightly by ~5 Å to make way for substrate translocation through the OB ring in state $E_{D2.1}^{USP14}$ (grey cartoon) as compared to those in the substrate-inhibited states (colorful cartoon). **g**, Side-by-side structural comparison of the USP14-bound RP in all states from the top view showing differential rotation of the lid and RPN1. **h**, Plots of the distance of pore-1 loop to the CP for those states not shown in Fig. 3c. The comparison shows that the pore-loop staircase architecture in state $S_B^{USP14}$, $S_C^{USP14}$ or $S_{D4}^{USP14}$ is similar to that of $E_A$-like state. **i**, Root-mean-squared-deviation (RMSD) values of the ATPase structures are mapped between any pairs of the thirteen states. **j**, An integrated schematic diagram of proteasome state transitions illustrates the full functional cycles of the proteasome in the presence and absence of USP14. The solid circles are the states observed in the current study, whereas the dashed circles are the states observed in previous studies of substrate-free[8] (orange) or substrate-engaged human proteasome[10] (green) in the absence of USP14. Color blue and salmon label the substrate-engaged and substrate-inhibited USP14-proteasome states, respectively. The states with closed and open CP gate are placed in pink and limon backgrounds, respectively. Vertical orange dashed lines link the state pairs with comparable AAA-ATPase structures. Black solid arrows and dashed arrows represent the possible structural transitions connecting the states observed in the current study and pervious USP14-free studies[8,10], respectively.

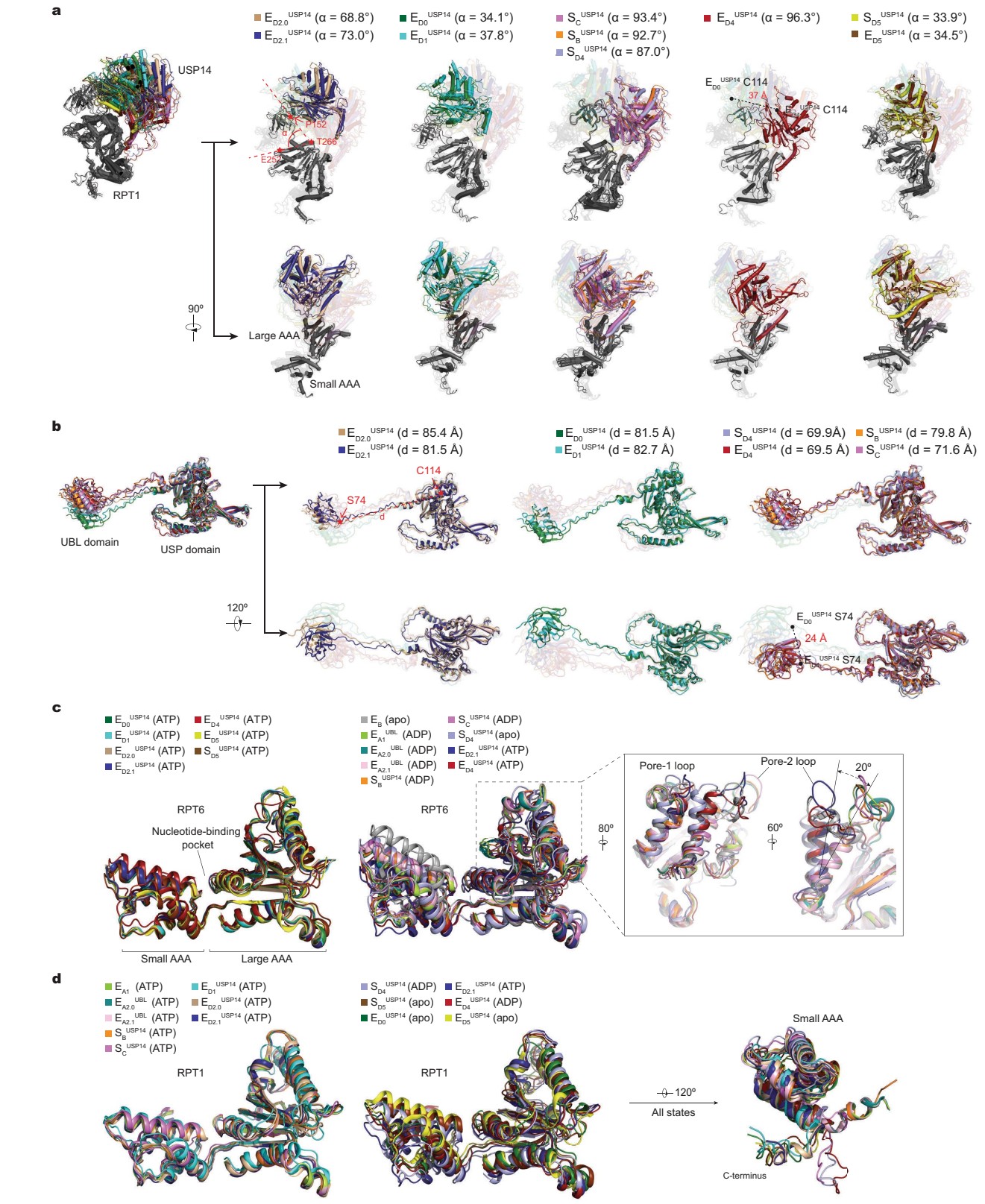

**Extended Data Fig. 8 |** See next page for caption.

**Extended Data Fig. 8 | Dynamics of USP14 and key ATPase subunits in the proteasome. a**, Superposition of the USP14-RPT1 subcomplex structures from different states aligned against the RPT1 large AAA subdomain. USP14 rotates together with the RPT1 OB domain and moves up over 37 Å (from $E_{D4}^{USP14}$ to $E_{D0}^{USP14}$) relative to the RPT1 AAA domain. The angle between the OB domain and the AAA domain is measured and labelled for each state. **b**, Superposition of the USP14 structures from different states aligned against their USP domain. The UBL domain moves up over 24 Å (from $E_{D4}^{USP14}$ to $E_{D0}^{USP14}$) relative to the USP domain. The distance between Ser74 and Cys114 is measured and labelled for each state. **c**, Superposition of the RPT6 AAA domain structures from different states aligned against the large AAA subdomain. Left, comparison of the RPT6 AAA structures in the ATP-bound states. Middle, comparison of states $S_{D4}^{USP14}$, $E_{D2.1}^{USP14}$ and $E_{D4}^{USP14}$, the ADP-bound states and state $E_B$ (PDB ID: 6MSE) shows conformational changes of the AAA domain driven by the ATP hydrolysis and nucleotide exchange. Right, the open-gate states $E_{D2.1}^{USP14}$, $E_{D4}^{USP14}$, and $S_{D4}^{USP14}$ show different refolding of both the pore-1 and pore-2 loops. **d**, Superposition of the RPT1 AAA domain structures from different states aligned against the large AAA subdomain. Left, comparison of the structures in the ATP-bound states. Middle, comparison of the structures in different nucleotide-binding states. Right, the C-terminal tails of RPT6 exhibit three major orientations.

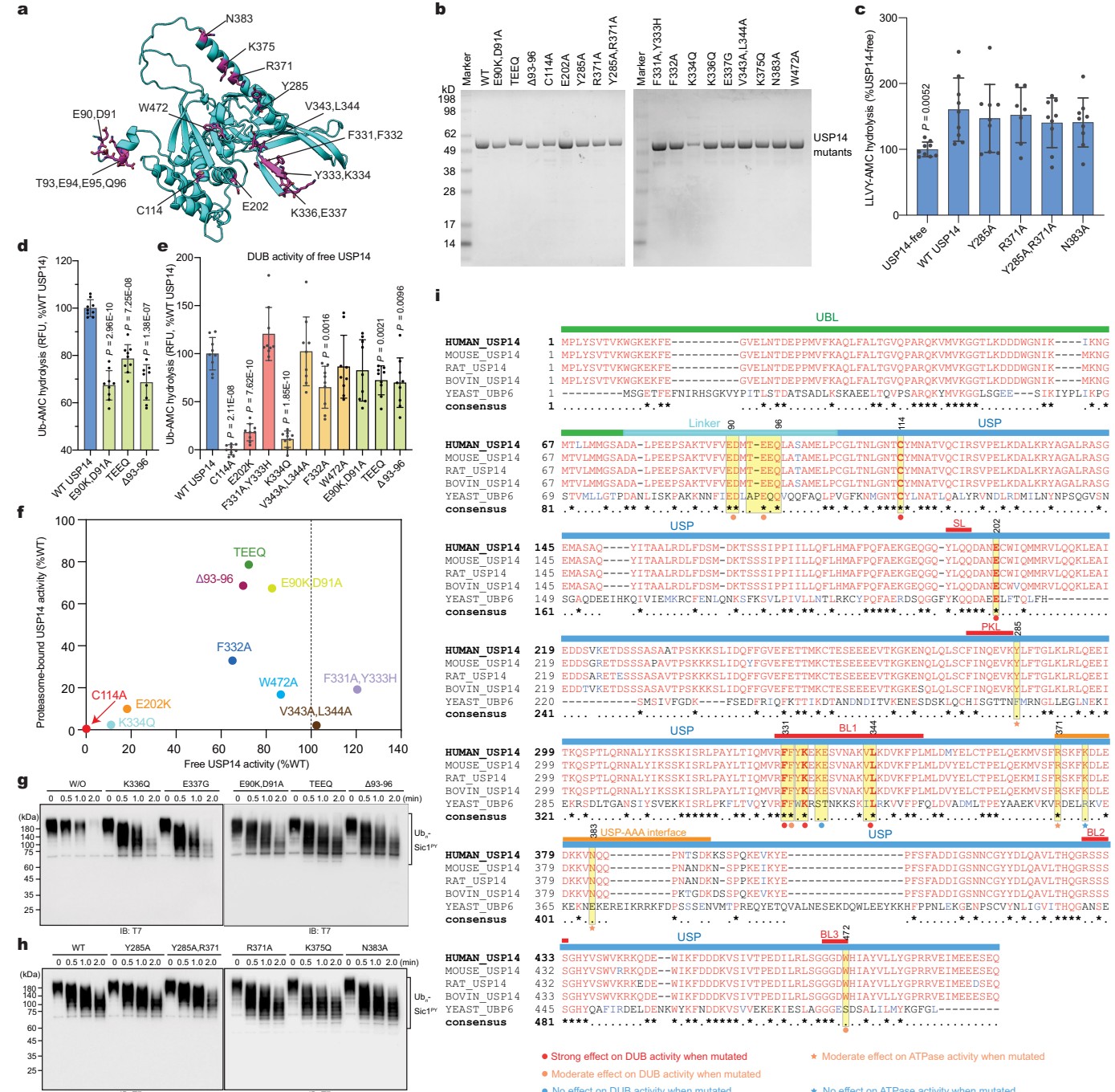

**Extended Data Fig. 9 | Structure-based site-directed mutagenesis.**
**a**, Mapping of the potential RPT-binding sites and other residues affecting USP14 activation onto the USP14 structure in the $E_{D4}^{USP14}$ model. The UBL domain of USP14 is not shown. **b**, Purification of USP14 mutants and analyzed by SDS/PAGE and stained with Coomassie blue. **c**, Peptidase activity assay was used to evaluate the effects of USP14 variants on regulating the CP gate opening. *P* values were analyzed using a two-tailed unpaired *t*-test between wild-type USP14-bound and USP14-free proteasomes. The results suggest that the mutants promote the CP gate opening to the same degree as that of wild-type USP14 as compared to that of the USP14-free proteasome. **d** and **e**, Ubiquitin-AMC hydrolysis assay to measure the DUB activity of USP14 mutants in the presence (panel **d**) or absence (panel **e**) of the human proteasome. Data are presented as mean ± s.d. from three independent experiments. *P* values were analyzed using a two-tailed unpaired *t*-test between USP14 mutants and wild-type USP14. *P* value is not labelled for data with *P* > 0.05, which is not significant. Data in **c**–**e** are presented as mean ± s.d. from three independent experiments, each with three replicates. Dots, individual data points. **f**, The

DUB activity of USP14 mutants in the presence or absence of proteasome. Data points are the average of individual data points shown in panels (**d**) and (**e**) and Fig. 2i. **g** and **h**, *In vitro* degradation of Ub$_n$-Sic1$^{PY}$ by the 26S proteasome in the presence of USP14 mutants testing the USP-OB interface or linker region (panel **g**) and affecting the USP-AAA interfaces (panel **h**) (repeated three times with similar results). Samples were analyzed by SDS–PAGE/Western blot using anti-T7 antibody. TEEQ, insertion of TEEQ after residue 92 in the linker region. Δ93–96, mutant with deletion of residues 93-96 in the linker region. W/O, the proteasome without binding to USP14. WT, the wildtype USP14-bound proteasome. **i**, Multiple sequence alignment of USP14 from five species was performed by Chimera. Annotation is based on the structural and mutational data from Figs. 2 and 3. The mutations with the strongest phenotypes (marked by red stars) all correspond to the amino acids (highlighted bold) that are fully conserved from yeast to human. Those mutations with moderate phenotypes correspond to the amino acids that are well conserved in mammals but may vary in yeast. For gel source data, see Supplementary Fig. 1.

# Extended Data Table 1 | Cryo-EM data collection, refinement and validation statistics

| | $E_{A1}^{UBL}$ (EMD-32272) (PDB 7W37) | $E_{A2.0}^{UBL}$ (EMD-32273) (PDB 7W38) | $E_{A2.1}^{UBL}$ (EMD-32274) (PDB 7W39) | $S_{B}^{USP14}$ (EMD-32281) (PDB 7W3I) | $S_{C}^{USP14}$ (EMD-32282) (PDB 7W3J) | $S_{D4}^{USP14}$ (EMD-32283) (PDB 7W3K) | $S_{D5}^{USP14}$ (EMD-32284) (PDB 7W3M) | $E_{D0}^{USP14}$ (EMD-32277) (PDB 7W3C) | $E_{D1}^{USP14}$ (EMD-32278) (PDB 7W3F) | $E_{D2.0}^{USP14}$ (EMD-32279) (PDB 7W3G) | $E_{D2.1}^{USP14}$ (EMD-32280) (PDB 7W3H) | $E_{D4}^{USP14}$ (EMD-32275) (PDB 7W3A) | $E_{D5}^{USP14}$ (EMD-32276) (PDB 7W3B) |
|---|---|---|---|---|---|---|---|---|---|---|---|---|---|
| **Data collection and processing** | | | | | | | | | | | | | |
| Magnification | 105,000 | 105,000 | 105,000 | 105,000 | 105,000 | 105,000 | 105,000 | 105,000 | 105,000 | 105,000 | 105,000 | 105,000 | 105,000 |
| Voltage (kV) | 300 | 300 | 300 | 300 | 300 | 300 | 300 | 300 | 300 | 300 | 300 | 300 | 300 |
| Electron exposure ($e^-$/Å$^2$) | 50 | 50 | 50 | 50 | 50 | 50 | 50 | 50 | 50 | 50 | 50 | 50 | 50 |
| Defocus range (μm) | -0.4 to -5.0 | -0.4 to -5.0 | -0.4 to -5.0 | -0.4 to -5.0 | -0.4 to -5.0 | -0.4 to -5.0 | -0.4 to -5.0 | -0.4 to -5.0 | -0.4 to -5.0 | -0.4 to -5.0 | -0.4 to -5.0 | -0.4 to -5.0 | -0.4 to -5.0 |
| Pixel size (Å) | 0.685 | 0.685 | 0.685 | 0.685 | 0.685 | 0.685 | 0.685 | 0.685 | 0.685 | 0.685 | 0.685 | 0.685 | 0.685 |
| Symmetry imposed | C1 | C1 | C1 | C1 | C1 | C1 | C1 | C1 | C1 | C1 | C1 | C1 | C1 |
| Initial particle images (no.) | 3,228,506 | 3,228,506 | 3,228,506 | 3,228,506 | 3,228,506 | 3,228,506 | 3,228,506 | 3,228,506 | 3,228,506 | 3,228,506 | 3,228,506 | 3,228,506 | 3,228,506 |
| Final particle images (no.) | 367,235 | 284,997 | 294,653 | 53,225 | 56,133 | 74,792 | 66,215 | 142,577 | 124,597 | 291,384 | 294,653 | 66,910 | 35,156 |
| Map resolution (Å) | 3.0 | 3.1 | 3.2 | 3.5 | 3.5 | 3.6 | 3.5 | 3.4 | 3.3 | 3.2 | 3.2 | 3.5 | 3.6 |
| FSC threshold | 0.143 | 0.143 | 0.143 | 0.143 | 0.143 | 0.143 | 0.143 | 0.143 | 0.143 | 0.143 | 0.143 | 0.143 | 0.143 |
| Map resolution range (Å) | 2.5-6.0 | 2.5-6.0 | 2.5-6.0 | 3.0-8.0 | 3.0-8.0 | 3.0-8.0 | 3.0-8.0 | 2.5-8.0 | 2.5-8.0 | 2.5-8.0 | 2.5-6.0 | 3.0-8.0 | 3.0-8.0 |
| **Refinement** | | | | | | | | | | | | | |
| Initial model used | 6MSB | 6MSD | 6MSD | 6MSB | 6MSB | 6MSK, 6MSE | 6MSK, 6MSH | 6MSK, 6MSJ | 6MSK, 6MSJ | 6MSK | 6MSK | 6MSK, 6MSB | 6MSK, 6MSJ |
| Model resolution (Å) | 3.1 | 3.2 | 3.2 | 3.6 | 3.6 | 3.6 | 3.6 | 3.6 | 3.6 | 3.3 | 3.3 | 3.6 | 3.8 |
| FSC threshold | 0.5 | 0.5 | 0.5 | 0.5 | 0.5 | 0.5 | 0.5 | 0.5 | 0.5 | 0.5 | 0.5 | 0.5 | 0.5 |
| Model resolution range (Å) | 2.5-6.0 | 2.5-6.0 | 2.5-6.0 | 3.0-8.0 | 3.0-8.0 | 3.0-8.0 | 3.0-8.0 | 2.5-8.0 | 2.5-8.0 | 2.5-8.0 | 2.5-6.0 | 3.0-8.0 | 3.0-8.0 |
| Map sharpening $B$ factor (Å$^2$) | -30 | -30 | -30 | -50 | -50 | -30 | -30 | -20 | -20 | -10 | -50 | -50 | -30 |
| **Model composition** | | | | | | | | | | | | | |
| Non-hydrogen atoms | 106,025 | 106,178 | 106,442 | 110,481 | 110,559 | 110,657 | 109,020 | 110,747 | 110,751 | 110,741 | 110,771 | 110,889 | 109,441 |
| Protein residues | 13,527 | 13,539 | 13,535 | 14,054 | 14,071 | 14,090 | 13,879 | 14,074 | 14,074 | 14,074 | 14,074 | 14,090 | 13,915 |
| Ligands | 13 | 13 | 12 | 7 | 7 | 5 | 5 | 6 | 6 | 6 | 10 | 7 | 5 |
| **$B$ factors (Å$^2$)** | | | | | | | | | | | | | |
| Protein | 132.08 | 128.18 | 130.96 | 122.34 | 137.24 | 167.16 | 170.46 | 187.97 | 124.16 | 150.21 | 143.50 | 125.03 | 149.95 |
| Ligand | 83.04 | 79.69 | 116.22 | 115.48 | 154.82 | 221.15 | 209.13 | 190.72 | 159.21 | 243.44 | 168.32 | 130.08 | 222.16 |
| **R.m.s. deviations** | | | | | | | | | | | | | |
| Bond lengths (Å) | 0.004 | 0.004 | 0.005 | 0.005 | 0.005 | 0.004 | 0.003 | 0.004 | 0.003 | 0.004 | 0.005 | 0.004 | 0.003 |
| Bond angles (°) | 0.803 | 0.800 | 0.816 | 0.822 | 0.844 | 0.818 | 0.775 | 0.754 | 0.742 | 0.756 | 0.782 | 0.809 | 0.785 |
| **Validation** | | | | | | | | | | | | | |
| MolProbity score | 1.66 | 1.69 | 1.68 | 1.75 | 1.80 | 1.73 | 1.74 | 1.73 | 1.71 | 1.72 | 1.75 | 1.75 | 1.72 |
| Clashscore | 5.00 | 5.47 | 5.25 | 5.71 | 6.45 | 5.88 | 5.94 | 5.66 | 5.42 | 5.58 | 5.60 | 5.70 | 5.52 |
| Rotamers outliers (%) | 0.15 | 0.18 | 0.15 | 0.26 | 0.55 | 0.18 | 0.14 | 0.13 | 0.16 | 0.31 | 0.24 | 0.38 | 0.13 |
| **Ramachandran plot** | | | | | | | | | | | | | |
| Favored (%) | 94.10 | 94.23 | 94.05 | 93.22 | 93.04 | 93.93 | 93.77 | 93.54 | 93.73 | 93.75 | 93.00 | 93.18 | 93.60 |
| Allowed (%) | 5.83 | 5.69 | 5.90 | 6.72 | 6.91 | 5.98 | 6.12 | 6.39 | 6.20 | 6.13 | 6.86 | 6.72 | 6.33 |
| Outliers (%) | 0.07 | 0.08 | 0.05 | 0.06 | 0.05 | 0.09 | 0.11 | 0.07 | 0.07 | 0.12 | 0.14 | 0.10 | 0.07 |

## Extended Data Table 2 | Summary of key structural features of USP14-bound proteasome states

| States | $E_{A1}^{UBL}$ | $E_{A2.0}^{UBL}$ | $E_{A2.1}^{UBL}$ | $S_B^{USP14}$ | $S_C^{USP14}$ | $S_{D4}^{USP14}$ | $S_{D5}^{USP14}$ | $E_{D4}^{USP14}$ | $E_{D5}^{USP14}$ | $E_{D0}^{USP14}$ | $E_{D1}^{USP14}$ | $E_{D2.0}^{USP14}$ | $E_{D2.1}^{USP14}$ |
|---|---|---|---|---|---|---|---|---|---|---|---|---|---|
| Site with ubiquitin density observed | RPT5 CC | RPN11 | RPN11 | USP14 | USP14 | USP14 | USP14 | USP14 | USP14 | USP14 | USP14 | USP14 | USP14 |
| Substrate density observed | None | None | Cleft between RPN11 and RPT4 OB | None | None | None | None | AAA channel | AAA channel | AAA channel | AAA channel | AAA channel | AAA channel |
| USP14 binding site | RPN1 T2 site (UBL) | RPN1 T2 site (UBL) | RPN1 T2 site (UBL) | RPN1 T2, RPT1 OB-AAA | RPN1 T2, RPT1 OB-AAA | RPN1 T2, RPT1 OB-AAA | RPN1 T2, RPT1 OB-AAA | RPN1 T2, RPT1 OB-AAA | RPN1 T2, RPT1 OB-AAA | RPN1 T2, RPT1 OB | RPN1 T2, RPT1 OB | RPN1 T2, RPT1 OB | RPN1 T2, RPT1 OB |
| Pore-loop staircase contacting substrate (from top to bottom) | None | None | None | None | None | None | None | RPT3, RPT4, RPT5, RPT1, RPT6 | RPT6, RPT3, RPT4, RPT5 | RPT2, RPT6, RPT3, RPT4 | RPT2, RPT6, RPT3, RPT4 | RPT1, RPT2, RPT6, RPT3 | RPT5, RPT1, RPT2, RPT6, RPT3 |
| RPT subunits disengaged from pore loop staircase | RPT6 | RPT6 | RPT6 | RPT6 | RPT6 | RPT6 | RPT1, RPT2 | RPT6 | RPT1, RPT2 | RPT5, RPT1 | RPT5, RPT1 | RPT4, RPT5 | RPT4 |
| RPT subunits with ADP bound | RPT6, RPT5 | RPT6, RPT5 | RPT6, RPT5 | RPT2, RPT6 | RPT2, RPT6 | RPT1, RPT2 | RPT5 | RPT2, RPT6 | RPT5 | RPT4, RPT5 | RPT4 | RPT3 | RPT3 |
| RPT subunits in apo-like state | None | None | None | None | None | RPT6, RPT3 | RPT1, RPT2 | None | RPT1, RPT2 | RPT1 | RPT5 | RPT4 | RPT4 |
| RPT C-tails insertion into α-pockets | RPT3, RPT5 | RPT3, RPT5 | RPT3, RPT5 | RPT3, RPT5 | RPT3, RPT5, RPT2, RPT6 | RPT3, RPT5, RPT1, RPT2, RPT6 | RPT3, RPT5, RPT1, RPT2, RPT6 | RPT3, RPT5, RPT1, RPT2, RPT6 | RPT3, RPT5, RPT1, RPT2, RPT6 | RPT3, RPT5, RPT1, RPT2, RPT6 | RPT3, RPT5, RPT1, RPT2, RPT6 | RPT3, RPT5, RPT1, RPT2, RPT6 | RPT3, RPT5, RPT1, RPT2, RPT6 |
| CP gate state | Closed | Closed | Closed | Closed | Closed | Open | Open | Open | Open | Open | Open | Open | Open |
| Resemblance to the USP14-free substrate-free conformations | $S_A$ (lid position, ATPase and RP-CP interface) | $S_A$ (ATPase and RP-CP interface) | $S_A$ (ATPase and RP-CP interface) | $S_B$ (ATPase and RP-CP interface) | $S_{D1}$ (lid position and ATPase) | $S_{D1}$ (lid position, pore-loop staircase and RP-CP interface) | $S_{D3}$ (lid position, pore-loop staircase and RP-CP interface) | $S_D$ (RP-CP interface) | $S_D$ (RP-CP interface) | $S_D$ (RP-CP interface) | $S_D$ (RP-CP interface) | $S_D$ (RP-CP interface) | $S_{D2}$ (lid position, pore-loop staircase and RP-CP interface) |
| Resemblance to the USP14-free substrate-bound conformations | $E_{A1}$ (ubiquitin binding, lid position, ATPase and RP-CP interface) | $E_{A2}$ (ubiquitin binding, lid position, ATPase and RP-CP interface) | $E_{A2}$ (ubiquitin binding, ATPase and RP-CP interface) | $E_{A1}$ (ATPase and RP-CP interface) | $E_{A1}$ (ATPase only); $E_C$ (RP-CP interface) | $E_B$ (ATPase only); $E_D$ (RP-CP interface) | $E_{C2}$ (lid position and ATPase only); $E_D$ (RP-CP interface) | $E_B$ (ATPase only); $E_D$ (RP-CP interface) | $E_{C1}$ (lid position and ATPase only); $E_D$ (RP-CP interface) | $E_{D0}$ (lid position, disengaged RPT subunits and RP-CP interface) | $E_{D1}$ (lid position, disengaged RPT subunits and RP-CP interface) | $E_{D2}$ (nucleotide de-binding states and RP-CP interface) | $E_{D2}$ (lid position, ATPase and RP-CP interface) |
| Putative functions | Ubiquitin recognition | Ubiquitin transfer to RPN11 | Priming for substrate insertion and deubiquitylation by RPN11 | 1) Substrate deubiquitylation by USP14, followed by substrate release from the proteasome without degradation  2) Prevention of substrate engagement with the AAA-ATPase motor | | | | 1) Substrate deubiquitylation by USP14 with processive substrate translocation and degradation  2) Stimulation of ATPase activity and CP gate opening | | | | |

Comparison is meant for the human proteasome under different conditions. The nucleotide states in the substrate-engaged states share certain comparable features consistent with the previously reported results[10]. ATP is generally bound to the substrate-engaged ATPases above the penultimate contact along the substrate. ADP is bound to the ATPase subunit at the lowest position in contact with substrate, which is poised for dissociation from substrate, whereas an apo-like or ADP-bound state is generally found in the substrate-disengaged ATPase subunits that are in the seam of the lock-washer-like ATPase ring. CC, coiled coil. CP, core particle.

# Reporting Summary

## Statistics

For all statistical analyses, confirm that the following items are present in the figure legend, table legend, main text, or Methods section.

| n/a | Confirmed | |
|---|---|---|
| ☐ | ☒ | The exact sample size (*n*) for each experimental group/condition, given as a discrete number and unit of measurement |
| ☒ | ☐ | A statement on whether measurements were taken from distinct samples or whether the same sample was measured repeatedly |
| ☐ | ☒ | The statistical test(s) used AND whether they are one- or two-sided<br>*Only common tests should be described solely by name; describe more complex techniques in the Methods section.* |
| ☒ | ☐ | A description of all covariates tested |
| ☐ | ☒ | A description of any assumptions or corrections, such as tests of normality and adjustment for multiple comparisons |
| ☐ | ☒ | A full description of the statistical parameters including central tendency (e.g. means) or other basic estimates (e.g. regression coefficient) AND variation (e.g. standard deviation) or associated estimates of uncertainty (e.g. confidence intervals) |
| ☐ | ☒ | For null hypothesis testing, the test statistic (e.g. *F*, *t*, *r*) with confidence intervals, effect sizes, degrees of freedom and *P* value noted<br>*Give P values as exact values whenever suitable.* |
| ☒ | ☐ | For Bayesian analysis, information on the choice of priors and Markov chain Monte Carlo settings |
| ☐ | ☒ | For hierarchical and complex designs, identification of the appropriate level for tests and full reporting of outcomes |
| ☒ | ☐ | Estimates of effect sizes (e.g. Cohen's *d*, Pearson's *r*), indicating how they were calculated |

*Our web collection on statistics for biologists contains articles on many of the points above.*

## Software and code

Policy information about availability of computer code

| Data collection | SerialEM v3.6.11 |
|---|---|
| Data analysis | MotionCor2 v1.2.1, GCTF v1.06, DeepEM v1.0, EMAN v2.22, SPIDER v22.10, ROME v1.1.2, RELION v3.1.3, AlphaCryo4D v0.1.0, COOT v0.9.5, Phenix v1.19.2, ResMap v1.1.4, Pymol v2.2.3, UCSF Chimera v1.16, ChimeraX v1.2.5, Image J v1.53, SPSS v27.0 |

For manuscripts utilizing custom algorithms or software that are central to the research but not yet described in published literature, software must be made available to editors and reviewers. We strongly encourage code deposition in a community repository (e.g. GitHub). See the Nature Portfolio guidelines for submitting code & software for further information.

## Data

Policy information about availability of data

All manuscripts must include a data availability statement. This statement should provide the following information, where applicable:

- Accession codes, unique identifiers, or web links for publicly available datasets
- A description of any restrictions on data availability
- For clinical datasets or third party data, please ensure that the statement adheres to our policy

Cryo-EM density maps of USP14-proteasome complexes have been deposited in the Electron Microscopy Data Bank (EMDB) (www.emdatasource.org) under accession codes EMD-32272 (EA1UBL), EMD-32273 (EA2.0UBL), EMD-32274 (EA2.1UBL), EMD-32275 (ED4USP14), EMD-32276 (ED5USP14), EMD-32277 (ED0USP14), EMD-32278 (ED1USP14), EMD-32279 (ED2.0USP14), EMD-32280 (ED2.1USP14), EMD-32281 (SBUSP14), EMD-32282 (SCUSP14), EMD-32283 (SD4USP14) and EMD-32284 (SD5USP14), EMD-32285 (EA1UBL with the local RPN1 density improved), EMD-32286 (EA2.0UBL with the local RPN1 density improved), EMD-32287 (EA2.1UBL with the local RPN1 density improved), EMD-32288 (ED4USP14 with the USP14 density improved), EMD-32289 (ED0USP14 with the RPN1 density improved), EMD-32290 (ED2.0USP14 with the RPN1 density improved), EMD-32291 (ED2.1USP14 with the USP14 density improved), and

# Field-specific reporting

Please select the one below that is the best fit for your research. If you are not sure, read the appropriate sections before making your selection.

☒ Life sciences     ☐ Behavioural & social sciences     ☐ Ecological, evolutionary & environmental sciences

For a reference copy of the document with all sections, see nature.com/documents/nr-reporting-summary-flat.pdf

# Life sciences study design

All studies must disclose on these points even when the disclosure is negative.

| | |
|---|---|
| Sample size | No statistical methods were used to estimate appropriate sample size. The actual sample size was increased until the finally achieved resolution and quality of 3D reconstructions of the cryo-EM structures meet the expectation, that is, no worse than 3.6 Å by gold-standard Fourier shell correlation (FSC) measurement. |
| Data exclusions | No data were excluded from the analysis. During cryo-EM data clustering, good cryo-EM images were chosen for further 3D analysis based on their achieved resolution and reconstruction quality. Poorer images were excluded in the final reconstructions based on the criteria of maximizing the map resolution and quality. |
| Replication | All functional experiments were repeated at least three times. All attempts at replication were successful. |
| Randomization | Single-particle sets of cryo-EM images were randomly split for the purposes of estimating overall resolution during Fourier shell correlation calculation. Otherwise randomization was not relevant to these studies. |
| Blinding | Blinding was not relevant to this study. |

# Reporting for specific materials, systems and methods

We require information from authors about some types of materials, experimental systems and methods used in many studies. Here, indicate whether each material, system or method listed is relevant to your study. If you are not sure if a list item applies to your research, read the appropriate section before selecting a response.

## Materials & experimental systems

| n/a | Involved in the study |
|---|---|
| ☐ | ☒ Antibodies |
| ☐ | ☒ Eukaryotic cell lines |
| ☒ | ☐ Palaeontology and archaeology |
| ☒ | ☐ Animals and other organisms |
| ☒ | ☐ Human research participants |
| ☒ | ☐ Clinical data |
| ☒ | ☐ Dual use research of concern |

## Methods

| n/a | Involved in the study |
|---|---|
| ☒ | ☐ ChIP-seq |
| ☒ | ☐ Flow cytometry |
| ☒ | ☐ MRI-based neuroimaging |

## Antibodies

| | |
|---|---|
| Antibodies used | Anti-T7 (abcamAnti-T7 (abcam, cat#: ab9115, Rabbit polyconal, 1:1000 )<br>Anti-RPN13 (abcam, cat#: ab157185, Rabbit monoclonal, 1;10000 )<br>Anti-USP14(Novus, cat#: NBP2-20826, Rabbit polyconal, 1:1000 ) |
| Validation | Information of the antibody validation is available through manufacturer's online database. No further validation was done on the antibody in the reported experiments. |

# Eukaryotic cell lines

Policy information about cell lines

| | |
|---|---|
| Cell line source(s) | A stable HEK293 cell line harboring HTBH (hexahistidine, TEV cleavage site, biotin, and hexahistidine) tagged hRPN11 , a gift from L. Huang, Departments of Physiology and Biophysics and of Developmental and Cell Biology, University of California, Irvine, California 92697. |
| Authentication | Further authentication was not performed for this study. |
| Mycoplasma contamination | Mycoplasma testing was not performed for this study. |
| Commonly misidentified lines (See ICLAC register) | No commonly misidentified cell lines were used in this study. |

