## [Peer Review File · Nature]

Manuscript Title: USP14-regulated allostery of the human proteasome by time-resolved cryo-EM

Reviewer Comments & Author Rebuttals

Reviewer Reports on the Initial Version:

Referees' comments:

Referee #1 (Remarks to the Author):

USP14/UBP6 is an evolutionarily conserved and proteasome-activated deubiquitinase (DUB). It has been shown to trim multiple ubiquitin chains off a substrate recruited to the proteasome and block or delay its proteasomal degradation. In this manuscript, the authors performed time-resolved cryo-EM analysis of the USP14-proteasome complex in the presence of a polyubiquitinated model substrate protein. A total of eleven conformational states were captured, some with substrate engaged and translocated and some without. By preparing samples with different incubation times and exchanging ATP with a nonhydrolyzable analog, the authors were able to further monitor the change of population distributions of those conformational states. Based on these structural results, the authors propose three mechanisms by which USP14 prevents RPN11-mediated substrate processing and control substrate deubiquitination and degradation by the proteasome. With structure-guided mutational studies, the authors have also mapped the USP14-proteasome interface, which is involved in USP14 activation by the proteasome and the regulation of ATPase activity of the proteasome core particle by the deubiquitinase. Overall speaking, the study has revealed the structural basis of the reciprocal regulation between USP14 and the proteasome. Although some of the mechanistic questions remain to be answered, this paper represents a major advance in the field, which justifies its publication in Nature if the authors can address the following concerns.

1. ED Figure 1g shows the blocking effect of USP14 on the proteasomal degradation of ubiquitinated-Sic1(PY). At 10 minutes incubation time, a significant amount of ubiquitinated Sic1 remained. It is not clear whether this is also true for the reactions used for preparing the cryo-EM grids, which include components at a different ratio and concentration. The authors should show the fate of the substrate during the time-resolved structural studies. Figure 1f indicates that the majority (80%) of the resolved particles at 10 min is in the "EA1" state. At this point, is the majority of the substrate already degraded through the ED-USP14 states? If so, which state represents the final step of RPN11-mediated cleavage of the last Ub chain as highlighted in Figure 4b? If there is still ubiquitinated substrate left, why the majority of the particles is in state EA1?

2. What is the affinity of USP14 toward the proteasome? What about its two individual domains? What is the ubiquitin binding-enhanced affinity of USP14 USP domain toward the proteasome? The authors use "USP14 binding" and "USP14 bound" to describe the states in which the USP14 USP domain is visible. However, isn't the proteasome associated with USP14 via its UBL domain in the EA1 and EA2 states? ED Figure 4a shows that USP14 UBL domain is associated with the proteasome

in both EA1 and EA2 states, which lack the density of the USP14 USP domain, indicating that the USP domain is flexibly tethered. ED5.1 and ED5.2, on the other hand, have the USP14 USP domain (bound to ubiquitin) clearly resolved. But the USP14 UBL domain is absent probably because RPN1, its docking site on the proteasome, is poorly resolved. Quantitative measurements are needed to assess whether the proteasome is expected to be saturated with USP14 under the experimental conditions described for the cryo-EM studies. The numbers will also help the readers to understand the dynamic interactions between the full length USP14 enzyme and the proteasome and validate the claim that “USP14 assembly on the proteasome is facilitated by ubiquitin binding on USP14”.

3. Figure 2e shows that, C114A, the catalytically inactive mutant of USP14, fails to block substrate degradation. This is in stark contrast to a previously study reporting the yeast USP14 ortholog, UBP6, can inhibit proteasome degradation independently from its catalytic activity. If the difference stems from the substrate, can it be explained by the structures?

4. How do the structures explain the previous observation that USP14 fails to deubiquitinate and inhibit the degradation of Sic1-PY with a single ubiquitin chain?

5. The rationale for making the E90K/D91A double point-mutations in the USP14 linker region is not clear. Are these two residues interacting with any part of the proteasome?

6. Figure 2ef, what is the E202K mutant for?

Referee #2 (Remarks to the Author):

Zhang and colleagues report the time resolved cryoEM analysis of USP14 in complex with the 26S proteasome. USP14 is one of three deubiquitinating enzymes associated with the proteasome and while it has been suggested that ubiquitin binding to USP14 induces ATPase activity of the proteasome, there are no molecular details concerning this activation. Here Zhang and colleagues report nine conformational states of the proteasome with USP14 bound and a model ubiquitinated substrate (Sic1) included to capture the conformational states adopted by proteasome bound to USP14. Of significance were the observations that no ubiquitin binding to RPN11 was observed in all these states, demonstrating that USP14 engagement of ubiquitin from a ubiquitinated substrate decouples the activity of RPN11 and favours new USP14-bound proteasomal state(s) to efficiently process the ubiquitinated substrates while preventing a second ubiquitinated substrate from binding. The functional importance of USP14 are validated through point mutations to probe the interfaces between USP14 and the core particle.

Overall, this is a significant study that finally reveals atomic details of USP14 activation by ubiquitinated substrates and how this is translated into proteasomal activation. This reviewer would recommend publication in Nature.

Some minor points that would be worth clarifying.

USP14 is not found constitutively bound to the proteasome as is evident due to the reconstitution of

the proteasome with USP14 and RPN13. It would be helpful for a more general audience for the authors to explain the rationale for addition of RPN13 to purified proteasomes. In addition, how do the authors know that the purified 26S proteasomes are in fact USP14 free? Given that these samples are not cross-linked prior to cryo plunging suggests that USP14 should be stably bound to the proteasome? Is the interaction between USP14 and the proteasome regulated through other mechanisms within cells?

The time resolved study shows that most movies were collected at 1 min (13,862) whereas at other time points an average 2k movies were collected. Could the lack of particles prevent adequate sub-classification of all the different states? Given the high percentage of non-USP14 bound particles (greater than 50%) would there be a significant number of particles to independently subclassify the different states at these timepoints? Or are these sub-classifications performed using prior knowledge of states from the larger datasets? Finally, have any kinetic analyses been performed that reflects the speed of proteasomal processing in the presence of USP14? It would be good to put this into context with reported biochemical observations, if known.

From Extended data Figure 2 five different datasets were collected to obtain over 2.4M particles. However, it is not clear from the methods, could the authors articulate this a little more clearly in the methods section.

On page 5 the authors state that, "USP14 binding appears to induce a greater degree of proteasome dynamics and expand its conformation space to sample both substrate-engaged and substrate-inhibited states". Can this be inferred from the numbers of states that have been subclassified for USP14 bound particles versus the non-bound and more populous Ea1 and EA2 states? Surely could these states not be sub-classified further given the large number of particles present. The authors may want to either clarify this or remove this assertive statement.

Extended data 3, please clearly label which regions of the RP maps correspond to USP14.

Referee #3 (Remarks to the Author):

Ubiquitin-specific protease 14 (USP14) is one of the three proteasome-associated deubiquitinating enzymes (DUBs). It is the most frequently observed DUB that reversibly associates with the human proteasome. The present work describes cryo-EM structures of human full-length USP14 in complex with functional proteasome in the act of substrate deubiquitination. Structure-guided mutagenesis experiments substantiate the functional importance of dynamic USP14-proteasome interfaces. In my opinion, the manuscript reads comprehensive and I suspect that by sending the submission for peer review, the editors have already ensured that the study is suitable for Nature. However, I want to mention that a lot is already known in the field of USP14, but as I am not familiar with this research topic, I cannot reliably judge the new findings of the present work and I am also not an expert in electron microscopy. Therefore, I recommend to additionally consult an expert for verifying the methodology and the novelty of the present study.

Mammalian USP14 (Ubp6 in yeast) is activated catalytically upon specific association with the 26S proteasome, and the molecular mechanism underlying the proteasome-mediated USP14 activation and its reciprocal regulation of the proteasome function remains not thoroughly understand. The manuscript by S. Zhang *et al.* resolve a series of cryo-EM structures of human 26S proteasome associated USP14, some of which engaged with the model substrate Sic1^{py}, representing conformational states in the substrate-engaged pathway. They reveal USP14 activation allosterically reprograms the conformation of the AAA-ATPase motor and stimulates the gate opening of the CP. The authors further propose several regulatory checkpoints on the proteasome, and the mechanism on USP14-mediated regulation of proteasome function. Overall, the structural work gives new insight into allosteric coordination between USP14 and 26S proteasome. However, the cryo-EM data processing is incomplete and many of the conclusions lack supportive data. Also, with several key conformational states not captured in the current study and lack evidence for the proper ordering of different states especially for the substrate-inhibited pathway, the proposed mechanism remains questionable. The following are comments and questions the authors need to address.

1) Due to lacking introduction and clear description on known structures of related proteasome system and their key features, including the previously published proteasome structures E_{A1}, E_{A2}, E_B, E_{C1} and E_{C2}, etc, frequently used to compare with their current structures, this manuscript is hard to read and confusing. The authors are also suggested to provide a structural comparison between their structures and the known proteasome structures with engaged Usp14/Ubp6.

2) Please provide more introduction on the design of their reaction system: why adding more PRN13, why using Sic1^{py} as the model substrate, what kind of polyUb chains produced in the system and can they represent the physiological polyUb chains?

3) The authors describe that “Our cryo-EM analysis suggests that ATP-to-ATP_γS exchange enriches states S_C^{USP14}, E_{D2}^{USP14} and E_{D4}^{USP14} but reduces states S_B^{USP14}, S_D^{USP14} and E_{D0}^{USP14} with an overall agreement with the ATP-only condition on the characterization of the conformational continuum (Extended Data Fig. 1j).” However, they didn’t provide the original cryo-EM classification and refinement results to delineate how they deduced this conclusion. Please show these data to support their conclusion.

4) The authors mention that the nominal resolutions of their UBP14 located nine cryo-EM structures are at 3.0-3.6 Å range. However, it is not clear which part of the map the authors used to define the resolutions. They can either use the complete 26S proteasome map (including both RP and CP) based on the FSC 0.143 standard to define their nominal resolution, or just use the resolution of the RP portion which are intensively analyzed in the manuscript. However, based on Extended Data Fig. 3, the RP masked resolution ranges at 6.2-3.6 Å, instead of 3.0-3.6 Å. The CP masked resolution is about 3.0-3.6 Å, but this portion is not the key component in their analysis. Thus, the authors are suggested to clarify this and use the proper resolution to describe their maps. In addition, how they performed the CP (focused) refinement should be described.

5) According to Fig. 1f, E_{A1} and E_{A2} are the major conformations in the sample at 1 min timepoint, but there are few descriptions on them. The authors are suggested to include how they obtained the E_{A1} and E_{A2} state maps in data processing workflow and provide data statics, and they deserve to be better analyzed. Moreover, please provide the original data to support their conclusion that “3D classification of cryo-EM data indicated that intermediate states are approximately maximized in 1 min after substrate addition (Fig. 1e)” in addition to the population plot in Fig. 1e.

6) According to their previous study (Fig. 1f in refence 10), ubiquitin was resolved in E_{A1}; however, in the current study, there appears no density for ubiquitin in the E_{A1} map (Extended

Fig. 4a). This is somewhat confusing. Is there bound ubiquitin in E_{A1} in this study? If yes, please illustrate where it is. This affects how to define this state.

As shown in Fig. 4a, the authors propose E_{A2} is the previous step of E_B , and E_{A1} is the previous step of E_{D4}^{USP14} . Please explain the reason for this ordering, why not E_{A1} is previous of E_B ? They also put forward a competition mechanism between Rpn11 and USP14 “given the absence of the E_B - and E_C -like states in the presence of USP14”. Although the majority of the particles (~1.4 million) are the “ S_A -like” state without observed USP domain, they did not present any further processing results on this major dataset. They could perform focused processing on the Rsp11 region on this dataset as well as other particle set without observable USP14 to carefully sort out whether there are E_B and E_C states in their current experimental condition. Otherwise, it is hard to draw a competition mechanism between Rpn11 and USP14 as proposed by the author.

7) According to Extended Table 2, the major difference between $S_{B/C}^{USP14}$ and S_D^{USP14} are the RPT C-tails inserted into alpha pockets of CP. The $S_{B/C}^{USP14}$ could also be defined as states after S_D^{USP14} . This simple model can as well include and explain all the structures resolved in this study. There is no compelling structural evidence to support that there is a branched pathway. The authors should give more convincing evidence to support this conclusion.

8) The drawing of Fig. 4b is conflicted to some statements in the text. It's described on page 11 that “In support of this model, all substrate-engaged USP14-proteasome states showed no ubiquitin binding to RPN11.” But in Fig. 4b, E_D^{USP14} has ubiquitin attached to Rpn11. Moreover, in Fig. 4b legend, it's written that “The substrate-inhibited pathway,.....has RPN11 blocking the substrate entrance of OB ring before any substrate insertion takes place.” and “The substrate-engaged pathway, has RPN11 closed on the OB entrance after a substrate has already inserted into the ATPase ring”. Based on these descriptions, Rpn11 orientation appears to be a key element to distinguish these states/pathways. However, the structural difference of Rpn11 in these states has never been described and illustrated in this manuscript. Also, in Fig. 4b, the Rpn11 are drawn in the same position for substrate-inhibited and substrate-engaged states. What's the difference between “blocking” and “closed on”?

9) There lack a summary and definition on the overall influence of USP14 on UPS, and explanations on its physiological significance.

10) On P. 6, the statement that “These kinetic features indicate ... that S_C^{USP14} and S_D^{USP14} changed into S_B^{USP14} in the late stage of substrate processing” is not supported. Dose the ‘late stage’ refer to the timespan between 5min and 10min? S_B^{USP14} didn't increase after 5min according to Fig. 1f, and the only increased state is E_{1A} if this refer to the time span between 1min and 5min.

11) In Extended Data Fig 5c, please also show the model-map fitting for the substrate and RPT subunits in the contacting regions for all the related states.

12) The authors describe that “While the overall proteasome conformations of states E_{D0}^{USP14} , E_{D1}^{USP14} and E_{D2}^{USP14} closely resemble those of USP14-free, substrate-engaged states $E_{D0.2}$, $E_{D1.2}$ and E_{D2} , only the AAA-ATPase components in states E_{D4}^{USP14} , $E_{D5.1}^{USP14}$ and $E_{D5.2}^{USP14}$ are comparable to those of USP14-free states E_B , E_{C1} and E_{C2} with defined differences, respectively^{10,25} (Extended Data Figs. 5, 6b)”. There is no conformational comparison data to support the similarity between E_{D0}^{USP14} and $E_{D0.2}$, and between $E_{D5.2}^{USP14}$ and E_{C2} in Extended Data Fig. 6b. Also, in Extended Data Fig. 5, we cannot see how “the pore-1 loop of RPT6 in state E_{D4}^{USP14} has considerably moved up toward the substrate as compared to that in state E_B (Extended Data Fig. 5c)”. Please provide data to support these conclusions.

13) Please provide the data for the statement that “RPN11 blocks the substrate entrance at the OB ring of the AAA-ATPase motor in states S_B^{USP14} , S_C^{USP14} and S_D^{USP14} ”, to support their conclusion that “these states sterically inhibit substrate insertion into the AAA-ATPase channel”. Also show data to support their statement that “The AAA-ATPase motors of these substrate-inhibited states all adopt a S_A -like conformation”.

14) The authors show the conformational comparison between E_{D4}^{USP14} and S_D^{USP14} and between E_{D4}^{USP14} and S_C^{USP14} in Extended Data Fig. 6b, and describe that “structural comparison between S_D^{USP14} and E_{D4}^{USP14} , which both have an open CP gate, indicates a highly comparable AAA-ATPase motor conformation, suggesting that S_D^{USP14} is perhaps an intermediate state connecting E_{D4}^{USP14} with other inhibitory states”. However, the aligned structures appear not very similar to each other in the AAA-ATPase motor region. Please provide the RMSD value between the AAA-ATPase motor of the two structures to validate their conclusion.

15) The authors describe that “the pore-1 loop of RPT6 in state E_{D4}^{USP14} has considerably moved up toward the substrate as compared to that in state E_B (Extended Data Fig. 5c)”. But the referred figure can’t tell such difference, since it doesn’t include the compared E_B structure. Please generate a figure that can address this point.

16) It’s written on Page 7 that “structural comparisons exhibit differential rotations of RPN1 and the lid relative to the base induced by USP14 (Extended Data Fig. 5b)”. But Extended Data Fig. 5b shows the models for AAA-ATPase, doesn’t include Rpn1 and the lid. The authors should generate a figure that can support this point.

17) Also on Page 7, the authors describe that “The BL1 region is an open loop in the crystal structures but is refolded into a β -hairpin sandwiched between the OB ring and ubiquitin in the proteasome”. Refolding is a dramatic change. There are only panels rendering the BL1 model. Density details for the “refolded β -hairpin” should also be exhibited to convey their point.

18) In the last panel of Fig. 2e, the *In vitro* degradation experiment, the band intensities for substrates degraded by WT proteasome at the timepoint of 0.5/1.0/2.0 min appear stronger than that at 0 min. Why the band intensity is increased in the later time points if the sample loading is in equivalent amount in a parallel experiment? In addition, the W472A mutant appears promoting proteasome substrate degradation at the timepoint of 2 min, with the substrate seemingly being degraded at 2 min, which doesn’t support their conclusion that “the single mutants F332A in BL1 and W472A in BL3 ... inhibited proteasome function (Fig. 2e, f).” There are similar concerns on their statement that “we made three mutants with deletion of residues 93-96, insertion of TEEQ after residue 92, and double point-mutations of E90K/D91A. While all three USP14 mutants exhibited inhibition of proteasomal degradation of Ub_n -Sic1^{PY}”. However, E90K/D91A appears promoting proteasome mediated degradation if compare the band intensities at the timepoint of 2 min as shown in Extended Data Fig. 8c, d.

19) In Fig. 2f, there are Ubiquitin-AMC hydrolysis results for all the mutants in Fig. 2e, except for C114A, why? There are *in vitro* degradation results for C114A and E202K in Fig. 2e, but there isn’t any comment in the text. So is K375Q in Fig. 3e.

20) On page 8, the authors describe that “States E_{D4}^{USP14} , $E_{D5.1}^{USP14}$ and $E_{D5.2}^{USP14}$ are specifically induced by USP14, since no comparable conformations were reconstructed in the absence of USP14 previously despite exhaustive 3D classification”. They should make it clear why the other USP14 bound states like E_{D1}^{USP14} are not included in this statement, what’s the difference between these three and the others, and how they compare with the previous ones?

21) There should be a more in-depth analysis on the mechanism of how and why the USP mutations influence the ATPase activity differently.

23) The authors describe that “Catalytically, USP14 trims ubiquitin chains from the substrate quite fast⁷. For this reason, we have not captured the USP14 conformation right at the moment of deubiquitylation.” Conformation right at the moment of deubiquitylation at Rpn11 was captured for 26S proteasome in their previous study. So why deubiquitylation mediated by USP14 can't be captured? Does it trim ubiquitin chains much faster than Rpn11? Is there any supporting evidence on this?

24) In Fig.3e, please also provide the P values for the ATPase activity assay.

Minor points:

1) Please make it clear what the purpose is by adding ATP γ S in the system and how it can quench the ATPase activity of the ATPase?

2) The meaning of “W/O” should be defined in the caption of Fig. 2.

3) It's written on page 9 “The primary USP-AAA interface is mediated by a negatively charged surface on a helix (residues 372-383)”. The negatively charged surface should be illustrated. Interacting residues in USP-AAA interfaces are analyzed in this paragraph. What's the criteria to count for interaction, distance between atoms? The authors are also suggested to show how well the model fits in map in the interaction interface. Additionally, the color style in Fig. 3c is confusing. Readers intuitively think the red residues belong to the orange part, and the blue residues belong to the blue chain.

4) The difference between black dashed arrow and black solid arrow are not introduced in the figure caption of Fig.4a.

Author Rebuttals to Initial Comments:

Point-by-Point Responses to Referees' Comments on Manuscript 2021-07-11629A

We thank all referees very much for reviewing our manuscript and constructive comments and for their support of publication of this study. In the revised manuscript, we have thoroughly addressed all referees' comments and questions. The sentences in the revised manuscript that are quoted in the following point-by-point responses are also highlighted in yellow, for the convenience of reviewing.

Referee #1:

USP14/UBP6 is an evolutionarily conserved and proteasome-activated deubiquitinase (DUB). It has been shown to trim multiple ubiquitin chains off a substrate recruited to the proteasome and block or delay its proteasomal degradation. In this manuscript, the authors performed time-resolved cryo-EM analysis of the USP14-proteasome complex in the presence of a polyubiquitinated model substrate protein. A total of eleven conformational states were captured, some with substrate engaged and translocated and some without. By preparing samples with different incubation times and exchanging ATP with a nonhydrolyzable analog, the authors were able to further monitor the change of population distributions of those conformational states. Based on these structural results, the authors propose three mechanisms by which USP14 prevents RPN11-mediated substrate processing and control substrate deubiquitination and degradation by the proteasome. With structure-guided mutational studies, the authors have also mapped the USP14-proteasome interface, which is involved in USP14 activation by the proteasome and the regulation of ATPase activity of the proteasome core particle by the deubiquitinase. Overall speaking, the study has revealed the structural basis of the reciprocal regulation between USP14 and the proteasome. Although some of the mechanistic questions remain to be answered, this paper represents a major advance in the field, which justifies its publication in Nature if the authors can address the following concerns.

Response: We thank the referee very much for constructive comments and supporting publication of this work in Nature. We have revised the manuscript to address all concerns raised, as detailed below.

1. ED Figure 1g shows the blocking effect of USP14 on the proteasomal degradation of ubiquitinated-Sic1(PY). At 10 minutes incubation time, a significant amount of ubiquitinated Sic1 remained. It is not clear whether this is also true for the reactions used for preparing the cryo-EM grids, which include components at a different ratio and concentration. The authors should show the fate of the substrate during the time-resolved structural studies. Figure 1f indicates that the majority (80%) of the resolved particles at 10 min is in the "E_{A1}" state. At this point, is the majority of the substrate already degraded through the E_D-USP14 states? If so, which state represents the final step of RPN11-mediated cleavage of the last Ub chain as highlighted in Figure 4b? If there is still ubiquitinated substrate left, why the majority of the particles is in state E_{A1}?

Response: We thank the referee for this comment. To address this issue, we have rerun the immunoblotting experiments corresponding to the time points used in the time-resolved structural study, under the same reaction condition (buffer composition and concentrations) as that used for time-resolved cryo-EM experiments. The previous

Extended Data Fig. 1g was done with proteasome/substrate concentrations that were quite different from those in time-resolved cryo-EM analysis. In the revised Extended Data Fig. 1j and k, we have provided newly reproduced results with additional time points (30 min) that better characterize the fate of the substrate under the same condition as that of time-resolved structural analysis. This result replaces the previous one (Extended Data Fig. 1g) in this figure. At 10 min, ~40% of the Sic1^{PY} substrate was degraded, whereas the majority of substrate was deubiquitylated. At 30 min, the majority of Sic1^{PY} substrates was eventually degraded, and USP14 apparently slowed down and suppressed the degradation process. These results are consistent with the cryo-EM analysis, in that partially deubiquitylated substrates may be insufficient for retaining E_D-like or S_D-like states. Additional cryo-EM analysis described below may help to further explain this.

The Sic1^{PY} substrate has an unfolded segment that facilitates its engagement with the proteasome. Previous studies from Goldberg's group have shown that the Sic1^{PY} without ubiquitylation can also be degraded by the proteasome but at a very slow rate (ref. 20). Thus, our observation of most of the substrates were degraded at 30 min is compatible with previous results.

In the revised Figure 4, we removed the highlight of the final step of RPN11-mediated cleavage of the last Ub chain, which we believe was insufficiently represented in the particle population and thus was missed by our current cryo-EM analysis. Based on our previous study (ref. 10), we expect that the state E_B structure represents the deubiquitylation state of RPN11.

During manuscript revision, we have conducted additional 3D classification using more data (~3.4 million particles from 45193 micrographs, which is ~41% increase from ~2.5 million particles in 32003 micrographs of the original submission) and found that previous state E_{A1} contained a new sub-state named E_{A2.0}. As shown in the revised Extended Data Figs. 4 and 7a,b, state E_{A2.0} has ubiquitin tightly bound to RPN11, like E_{A2.1}. The overall conformation of E_{A2.0} resides between E_{A1} and E_{A2.1} (Extended Data Fig. 7a-c). As shown in the revised Fig. 1f, this deeper classification allowed us to clarify that at 10 min, there were ~40% of E_{A1}, ~30% of E_{A2.0}, and only ~4% of E_{A2.1}. This appears to agree with the revised Extended Data Fig. 1k, in which ~40% of substrates were degraded at 10 min. Both E_{A2.0} and E_{A2.1} are thought to represent the states prior to RPN11-catalyzed deubiquitylation, given clear density of ubiquitin bound to RPN11 in these states as shown in the revised Extended Data Fig. 7a, b. Thus, the remaining ubiquitinated substrates, which potentially recycled back from USP14 deubiquitylation, can be re-directed to RPN11 for deubiquitylation and degradation.

2. What is the affinity of USP14 toward the proteasome? What about its two individual domains? What is the ubiquitin binding-enhanced affinity of USP14 USP domain toward the proteasome? The authors use "USP14 binding" and "USP14 bound" to describe the states in which the USP14 USP domain is visible. However, isn't the proteasome associated with USP14 via its UBL domain in the E_{A1} and E_{A2} states? E_D Figure 4a shows that USP14 UBL domain is associated with the proteasome in both E_{A1} and E_{A2} states, which lack the density of the USP14 USP domain, indicating that the USP domain is flexibly tethered. E_{D5.1} and E_{D5.2}, on the other hand, have the USP14 USP domain (bound to ubiquitin) clearly resolved. But the USP14 UBL domain is absent probably because RPN1, its docking site on the proteasome, is poorly resolved. Quantitative measurements are needed to assess whether the proteasome is expected to be saturated with USP14 under the experimental conditions described

for the cryo-EM studies. The numbers will also help the readers to understand the dynamic interactions between the full length USP14 enzyme and the proteasome and validate the claim that “USP14 assembly on the proteasome is facilitated by ubiquitin binding on USP14”.

Response: We thank the referee for these questions. To address these issues, we have conducted additional experiments using MicroScale Thermophoresis (MST) technique to measure the affinity of USP14 toward the proteasome. As shown in the revised Extended Data Fig. 1l-o, the dissociation constant (K_D) of USP14 toward the proteasome was measured to be ~95 nM without polyubiquitylated substrate, and ~44 nM with polyubiquitylated Sic1^{PY}. The UBL and USP domains of USP14 were also recombinantly expressed and purified separately (as shown in the revised Extended Data Fig. 1f) to conduct the MST experiments. Their K_D toward the proteasome are measured to be ~137 nM and ~135 nM, respectively. Based on these measured K_D values, we can estimate that under our experimental condition, ~99% of the proteasome would be bound to USP14 in the absence of substrate, because the molar concentration (10 μ M) of USP14 was about 10 times that of the 26S proteasome. In the presence of the substrate, the K_D value predicts that >99% of the proteasome is bound to USP14. These results have been briefly discussed in the revised text on page 7, as quoted below:

“This ternary architecture of ubiquitin-USP-OB sandwich indicates that ubiquitin binding stabilizes the BL1 β -hairpin conformation and USP interaction with the OB ring. Indeed, microscale thermophoresis (MST) measurements show that the dissociation constant (~44 nM) of USP14 against the proteasome in the presence of Ub_n-Sic1^{PY} is approximately half of that (~95 nM) without the ubiquitin conjugates and one third of that (~137 or ~135 nM) of either UBL or USP domain alone (Extended Data Fig. 1l-o). Thus, USP14 affinity toward the proteasome is equally contributed by its UBL and USP domains and substantially enhanced by Ub_n-Sic1^{PY}, consistent with early studies³³.”

3. Figure 2e shows that, C114A, the catalytically inactive mutant of USP14, fails to block substrate degradation. This is in stark contrast to a previously study reporting the yeast USP14 ortholog, UBP6, can inhibit proteasome degradation independently from its catalytic activity. If the difference stems from the substrate, can it be explained by the structures?

Response: We thank the referee for this good question. The non-catalytic effect previously seen for Ubp6, in light of our structures, is dependent on the USP14 interactions with both ubiquitin and RPT1-RPT2. Our results on C114A mutant of USP14 are consistent with the previously published work on USP14 C114A (ref. 6, Fig. 1), which also failed to block degradation of other substrates (Ub_n-CCNB). Thus, the noncatalytic effect is probably a function of the species of origin of the proteasome, namely, the noncatalytic effect of human USP14 on the proteasome is not as distinct as that of yeast Ubp6-proteasome.

Based on our structures, C114A may reduce ubiquitin engagement with USP14, which in turn disrupts USP14 interaction with RPT1-RPT2 OB domains. Similarly, the single mutant E202A of USP that disrupts its electrostatic interaction, in the form of a salt bridge, with R42 of ubiquitin (revised Fig. 3c), also disrupted USP14 activity and failed in suppressing the proteasomal degradation of the substrate. At least for human USP14, its binding to the OB ring of the proteasome appears to be highly dependent on its appropriate interaction with ubiquitin conjugates.

4. *How do the structures explain the previous observation that USP14 fails to deubiquitinate and inhibit the degradation of Sic1-PY with a single ubiquitin chain?*

Response: Our structural model suggests that ubiquitin recognition by USP14 in the proteasome likely requires at least another helper ubiquitin chain that is already anchored on a nearby ubiquitin receptor. This ubiquitin chain may not be available for USP14 binding but can be subsequently trimmed by RPN11. We have mentioned this point in the legend of Fig. 4, as quoted below:

“Although our data do not intuitively explain why USP14 trims ubiquitin until the last one on a substrate remains, the structures provide geometric constraints for polyubiquitin chain binding to both ubiquitin receptors and USP14 and implicate that ubiquitin recognition by USP14 in the proteasome likely requires at least another helper ubiquitin chain that is already anchored on a nearby ubiquitin receptor. This helper ubiquitin chain may not be available for USP14 binding but can be readily trimmed by RPN11.”

5. *The rationale for making the E90K/D91A double point-mutations in the USP14 linker region is not clear. Are these two residues interacting with any part of the proteasome?*

Response: We have added a few sentences to explain the rationale for making the E90K/D91A double mutant and its interpretation on page 8 as quoted below:

“In the linker region, the negatively charged residues Glu90 and Asp91 are potentially involved in transient interactions with the RPN1 toroid (Extended Data Figs. 7g, 9a)”.

6. *Figure 2ef, what is the E202K mutant for?*

Response: The E202K mutant disrupts the interaction between USP domain and ubiquitin. We have added a short sentence in the figure legend on page 8 to clarify this, as quoted below:

“...as well as single mutant E202K disrupting the salt bridge between Glu202 of USP and Arg42 of ubiquitin (Fig. 3c), ...”. We also revised Fig. 3c to directly illustrate this.

Referee #2:

Zhang and colleagues report the time resolved cryoEM analysis of USP14 in complex with the 26S proteasome. USP14 is one of three deubiquitinating enzymes associated with the proteasome and while it has been suggested that ubiquitin binding to USP14 induces ATPase activity of the proteasome, there are no molecular details concerning this activation. Here Zhang and colleagues report nine conformational states of the proteasome with USP14 bound and a model ubiquitinated substrate (Sic1) included to capture the conformational states adopted by proteasome bound to USP14. Of significance were the observations that no ubiquitin binding to RPN11 was observed in all these states, demonstrating that USP14 engagement of ubiquitin from a ubiquitinated substrate decouples the activity of RPN11 and favours new USP14-bound proteasomal state(s) to efficiently process the ubiquitinated substrates while preventing a second ubiquitinated substrate from binding. The functional importance of USP14 are validated through point mutations to probe the interfaces between USP14 and the core particle.

Overall, this is a significant study that finally reveals atomic details of USP14 activation by ubiquitinated substrates and how this is translated into proteasomal activation. This reviewer would recommend publication in Nature.

Response: We thank the referee very much for constructive comments and recommending publication of this work in Nature. We have revised the manuscript to address all concerns raised, as described below.

Some minor points that would be worth clarifying.

USP14 is not found constitutively bound to the proteasome as is evident due to the reconstitution of the proteasome with USP14 and RPN13. It would be helpful for a more general audience for the authors to explain the rationale for addition of RPN13 to purified proteasomes.

Response: We have added a few sentences to explain the rationale for addition of RPN13 to the purified proteasomes on page 4, as quoted below:

“Although the ubiquitin receptor RPN13 appears to be present in the purified USP14-free proteasome (Extended Data Fig. 1g), it was missing and likely sub-stoichiometric in previous cryo-EM reconstructions of human proteasomes^{8-10,25,27-29}. In an attempt to saturate the proteasome for RPN13, thus enhancing substrate recruitment by the USP14-bound proteasome, both purified USP14 and RPN13 were mixed in stoichiometric excess with the USP14-free proteasome prior to substrate addition.” We have also revised Extended Data Fig. 1g to support this description.

In the Methods section “Atomic model building and refinement” (page 25), we clarify: “Despite the presence of RPN13 in our purified proteasome (Extended Data Fig. 1g) and the addition of excessive RPN13 to saturate the proteasome, no reliable density was observed for RPN13 in all cryo-EM maps, thus precluding the atomic modeling of RPN13 and likely reflecting its highly dynamic association with the proteasome.”

In addition, how do the authors know that the purified 26S proteasomes are in fact USP14 free? Given that these samples are not cross-linked prior to cryo plunging suggests that USP14 should be stably bound to the proteasome? Is the interaction between USP14 and the proteasome regulated through other mechanisms within cells?

Response: To address these issues, we have provided Western blot results as direct evidence for the lack of USP14 in the purified 26S proteasome in Extended Data Fig. 1h. In the revised Method section, we also clarified that high-salt washing step (with ~90 mM Na⁺ in the PBS buffer) has been previously found to remove USP14 specifically in Methods section (refs. 7, 10). It remains possible that USP14 interaction with the proteasome is regulated through other substrate-shuttling proteins, such RAD23A/B and Ubiquilins. This is an ongoing study in our laboratory which is rather difficult and challenging.

The time resolved study shows that most movies were collected at 1 min (13,862) whereas at other time points an average 2k movies were collected. Could the lack of particles prevent adequate sub-classification of all the different states? Given the high percentage of non-USP14 bound particles (greater than 50%) would there be a significant number of particles to independently subclassify the different states at these time points? Or are these sub-classifications performed using prior knowledge of states from the larger datasets?

Response: We thank the referee for these questions that help with the clarity of our

method description. We agree with the referee that the lack of particles could potentially prevent adequate subclassification. During manuscript revision, we have conducted additional 3D classification using more data (~3.4 million particles from 45193 micrographs, which is ~41% increase from ~2.5 million particles in 32003 micrographs of the original submission), which included a new dataset collected at 0.5 min after substrate addition to the USP14-loaded proteasome. Thus, we have combined the datasets from all time points for focused 3D classification. This data-combined classification strategy has been previously used in the study of time-resolved ribosome dynamics by cryo-EM (refs. 51, 52). To clarify this, we have revised the Extended Data Fig. 2b,c to show how this was done in the data processing workflow. The time labels were only used to sort out the time-dependent sub-classes after the combined, focused 3D classification of all states was exhaustively done. Our analysis of considerably more data did not change our major conclusion but did allow minor improvement or corrections to the sparsely populated intermediate states.

Finally, have any kinetic analyses been performed that reflects the speed of proteasomal processing in the presence of USP14? It would be good to put this into context with reported biochemical observations, if known.

Response: We thank the referee for the question and kind suggestion. We have provided in the revised Extended Data Fig. 1k a more complete time-dependent titration of the substrate degradation using Western blot corresponding to the same degradation condition used in cryo-EM analysis. The results show that the half time ($t_{1/2}$) of degradation reaction was increased approximately two-fold in the presence of USP14, reflecting that the overall degradation rate is reduced twice by USP14. Very similar kinetics of USP14-bound proteasomal degradation were also reported in ref. 7 (Lee et al. Nature 2016) albeit with another substrate, cyclin B1 (NCB1). We have added a sentence on page 5, as quoted below:

“Time-dependent quantification of the degraded Ub_n-Sic1^{PY} substrate indicates that USP14 reduced the degradation rate by approximately twice as compared to that by the USP14-free proteasome⁷ (Extended Data Fig. 1k).”

From Extended data Figure 2 five different datasets were collected to obtain over 2.4M particles. However, it is not clear from the methods, could the authors articulate this a little more clearly in the methods section.

Response: We thank the referee for pointing this out. To improve clarity on this issue, we have revised the corresponding Method section on page 23 of the revised manuscript, as quoted below:

“For time-resolved sample conditions, 1,781, 2,298, 15,841, 13,868, 2,073 and 2,071 movies were collected for the reaction time of 0, 0.5, 1, 5, and 10 min, respectively. For the condition of exchanging ATP to ATPyS at 1 min after substrate addition, 21,129 movies were collected.”

We also revised Extended Data Fig. 2 to include more details during the intermediate steps of analysis of each of the six datasets. Note that during revision we added a dataset corresponding to the time point of 0.5 min, thus the revision now included six datasets.

On page 5 the authors state that, “USP14 binding appears to induce a greater degree of proteasome dynamics and expand its conformation space to sample both substrate-engaged and substrate-inhibited states”. Can this be inferred from the numbers of states that have been subclassified for USP14 bound particles versus the non-bound

and more populous E_{A1} and E_{A2} states? Surely could these states not be sub-classified further given the large number of particles present. The authors may want to either clarify this or remove this assertive statement.

Response: We thank the referee for pointing this out. The referee has made a good point that the conformational space cannot be simply inferred from the numbers of states that have been classified, because it remains possible that certain sub-states are insufficiently observed due to the limitation of the classification methods. We therefore have removed this assertion in the revised manuscript on page 5.

Extended data 3, please clearly label which regions of the RP maps correspond to USP14.

Response: To address this suggestion, we have added labels for USP14 in the revised Extended Data Fig. 3.

Referee #4:

Mammalian USP14 (Ubp6 in yeast) is activated catalytically upon specific association with the 26S proteasome, and the molecular mechanism underlying the proteasome-mediated USP14 activation and its reciprocal regulation of the proteasome function remains not thoroughly understand. The manuscript by S. Zhang et al. resolve a series of cryo-EM structures of human 26S proteasome associated USP14, some of which engaged with the model substrate Sic1^{py}, representing conformational states in the substrate-engaged pathway. They reveal USP14 activation allosterically reprograms the conformation of the AAA-ATPase motor and stimulates the gate opening of the CP. The authors further propose several regulatory checkpoints on the proteasome, and the mechanism on USP14-mediated regulation of proteasome function. Overall, the structural work gives new insight into allosteric coordination between USP14 and 26S proteasome. However, the cryo-EM data processing is incomplete and many of the conclusions lack supportive data. Also, with several key conformational states not captured in the current study and lack evidence for the proper ordering of different states especially for the substrate-inhibited pathway, the proposed mechanism remains questionable. The following are comments and questions the authors need to address.

Response: We thank the referee very much for constructive suggestions on the improvement of the manuscript and for confirming the novelty of this work. We have revised the manuscript to address all comments and questions raised, as described below.

1) Due to lacking introduction and clear description on known structures of related proteasome system and their key features, including the previously published proteasome structures E_{A1} , E_{A2} , E_B , E_{C1} and E_{C2} , etc, frequently used to compare with their current structures, this manuscript is hard to read and confusing. The authors are also suggested to provide a structural comparison between their structures and the known proteasome structures with engaged Usp14/Ubp6.

Response: We thank the referee for kind suggestions. We have added a few sentences to introduce the previously published proteasome structures on page 4, as quoted below:

“Previous cryogenic electron microscopy (cryo-EM) studies have determined the atomic structures of the USP14-free, substrate-engaged proteasomes at key functional steps, including ubiquitin recognition (states E_{A1} and E_{A2}), deubiquitylation

(state E_B), translocation initiation (states E_{C1} and E_{C2}) and processive degradation (states E_{D1} and E_{D2})¹⁰.”

We have also provided comparison between our structures and the previous proteasome structures with USP14/Ubp6 in the revised Extended Data Fig. 5c. Given the low-resolution nature of these previous structures, the comparison is limited to the overall consistency of the approximate location of USP14/Ubp6 on the proteasome. Accordingly, we revised the main text on page 5 to briefly mention this, as quoted below:

“The overall USP14 structure bridges the RPN1 and RPT1 subunits in all the above-described S_D -like and E_D -like states, agreeing with previous low-resolution studies²³⁻²⁶ (Extended Data Fig. 5c).”

2) Please provide more introduction on the design of their reaction system: why adding more PRN13, why using Sic1^{PY} as the model substrate, what kind of polyUb chains produced in the system and can they represent the physiological polyUb chains?

Response: We thank the referee for kind suggestions. We have added a few sentences on page 4 to provide more detailed introduction on the design of our reaction system on page 4, as quoted below:

“Although the ubiquitin receptor RPN13 appears to be present in the purified USP14-free proteasome (Extended Data Fig. 1f), it was missing and likely sub-stoichiometric in previous cryo-EM reconstructions of human proteasomes^{8-10,25,27-29}. In an attempt to saturate the proteasome for RPN13, thus enhancing substrate recruitment by the USP14-bound proteasome, both purified USP14 and RPN13 were mixed in stoichiometric excess with the USP14-free proteasome prior to substrate addition. We used Sic1^{PY} conjugated with Lys63-linked polyubiquitin chains (Ub_n-Sic1^{PY}) as a model substrate⁷. Lys63-linked chains are the second most abundant ubiquitin linkages in mammals and regulate essential intracellular functions such as cell cycle and immune responses¹.”

Besides, previous studies have postulated that USP14 or Ubp6 may preferentially cleave Lys63-linked chains (ref. 7).

3) The authors describe that “Our cryo-EM analysis suggests that ATP-to-ATP_γS exchange enriches states S_C^{USP14} , E_{D2}^{USP14} and E_{D4}^{USP14} but reduces states S_B^{USP14} , S_D^{USP14} and E_{D0}^{USP14} with an overall agreement with the ATP-only condition on the characterization of the conformational continuum (Extended Data Fig. 1j).” However, they didn’t provide the original cryo-EM classification and refinement results to delineate how they deduced this conclusion. Please show these data to support their conclusion.

Response: We have revised the Extended Data Figs. 2 and 5a to provide more detail on the cryo-EM classification and refinement that differentiate the results of ATP-only samples and those with ATP-to-ATP_γS exchange. Because of the length limitation of the main text, the revised statement is now moved to the legend of Extended Data Fig. 1r, because this is not our major conclusion, as quoted below:

“Our cryo-EM analysis suggests that ATP-to-ATP_γS exchange enriches states $E_{A2.1}^{UBL}$, S_C^{USP14} , S_{D5}^{USP14} , E_{D1}^{USP14} , $E_{D2.1}^{USP14}$ and E_{D4}^{USP14} , but reduces states E_{A1}^{UBL} , $E_{A2.0}^{UBL}$, S_B^{USP14} , E_{D0}^{USP14} and $E_{D2.0}^{USP14}$ ”.

4) The authors mention that the nominal resolutions of their UBP14 located nine cryo-EM structures are at 3.0-3.6 Å range. However, it is not clear which part of the map the authors used to define the resolutions. They can either use the complete 26S

proteasome map (including both RP and CP) based on the FSC 0.143 standard to define their nominal resolution, or just use the resolution of the RP portion which are intensively analyzed in the manuscript. However, based on Extended Data Fig. 3, the RP masked resolution ranges at 6.2-3.6 Å, instead of 3.0-3.6 Å. The CP masked resolution is about 3.0-3.6 Å, but this portion is not the key component in their analysis. Thus, the authors are suggested to clarify this and use the proper resolution to describe their maps. In addition, how they performed the CP (focused) refinement should be described.

Response: We thank the referee for kind suggestions. We have revised the Extended Data Fig. 3b-e to show both gold-standard FSCs of the complete 26S proteasome maps (including both RP and CP) and of the RP-masked maps, with and without applying mask during resolution measurement. In the revision, we have increased substantially the total amount of data analyzed (~3.4 million particles from 45193 micrographs, which is ~41% increase from ~2.5 million particles in 32003 micrographs of the original submission). This mostly improved the cryo-EM reconstructions of the sparsely populated states, with little improvement for the most populated states. Thus, we were able to improve the RP-masked resolution to the range of 3.3-4.7 Å, while obtained two more sparsely populated states. In addition, we have revised the Extended Data Fig. 2 to clarify the CP-focused refinement.

5) According to Fig. 1f, E_{A1} and E_{A2} are the major conformations in the sample at 1 min time point, but there are few descriptions on them. The authors are suggested to include how they obtained the E_{A1} and E_{A2} state maps in data processing workflow and provide data statics, and they deserve to be better analyzed. Moreover, please provide the original data to support their conclusion that “3D classification of cryo-EM data indicated that intermediate states are approximately maximized in 1 min after substrate addition (Fig. 1e)” in addition to the population plot in Fig. 1e.

Response: We thank the referee for kind suggestions. To address this question, we have revised Extended Data Fig. 2 to include the workflow of data processing related to E_{A1} and E_{A2} and also presented estimates of their resolutions in Extended Data Fig. 3. Additional analysis and structural comparison of these states are provided in the revised Extended Data Figs. 5-7. The original data for the plot in Fig. 1e, f are now provided in the revised Extended Data Fig. 2c.

6) According to their previous study (Fig. 1f in refence 10), ubiquitin was resolved in E_{A1} ; however, in the current study, there appears no density for ubiquitin in the E_{A1} map (Extended Fig. 4a). This is somewhat confusing. Is there bound ubiquitin in E_{A1} in this study? If yes, please illustrate where it is. This affects how to define this state. As shown in Fig. 4a, the authors propose E_{A2} is the previous step of E_B , and E_{A1} is the previous step of E_{D4}^{USP14} . Please explain the reason for this ordering, why not E_{A1} is previous of E_B ? They also put forward a competition mechanism between Rpn11 and USP14 “given the absence of the E_B - and E_C -like states in the presence of USP14”. Although the majority of the particles (~1.4 million) are the “ S_A -like” state without observed USP domain, they did not present any further processing results on this major dataset. They could perform focused processing on the Rsp11 region on this dataset as well as other particle set without observable USP14 to carefully sort out whether there are E_B and E_C states in their current experimental condition. Otherwise, it is hard to draw a competition mechanism between Rpn11 and USP14 as proposed by the author.

Response: We thank the referee for these questions. We apologize that in the methods we did not sufficiently describe our in-depth sub-classification analysis of the E_A-like states of the dataset. To further our search of these states, we have analyzed considerably more data including ~3.4 million particles of 26S (from 45193 micrographs, which is ~41% increase from the original submission using 32003 micrographs). We have taken the referee's suggestion to conduct additional focused classification by using a mask around RPN11. The results appeared to reveal three E_A-like states, namely E_{A1}^{UBL}, E_{A2.0}^{UBL} and E_{A2.1}^{UBL}, without observing any sub-class that resembles state E_B and E_C. This implies the absence or very low population of E_B and E_C below the sensitivity of current 3D classification method (Extended Data Fig. 2). This result is consistent with the previous studies indicating that RPN11-catalyzed deubiquitylation is largely suppressed or inhibited in USP14-loaded proteasomes in refs. 7, 20 of this manuscript (i.e., Lee et al. *Nature* 2016; Kim and Goldberg, *J. Biol. Chem.* 2017). The competition mechanism between RPN11 and USP14 is well supported by biochemical data in refs. 7 and 20.

7) According to Extended Table 2, the major difference between S_{B/C}^{USP14} and S_D^{USP14} are the RPT C-tails inserted into alpha pockets of CP. The S_{B/C}^{USP14} could also be defined as states after S_D^{USP14}. This simple model can as well include and explain all the structures resolved in this study. There is no compelling structural evidence to support that there is a branched pathway. The authors should give more convincing evidence to support this conclusion.

Response: We agree with the referee that S_{B/C}^{USP14} can be reversibly converted to and from S_D^{USP14} and revised Figures 3 and 4 accordingly. In the revised Fig. 4, we postulate that S_{B/C}^{USP14} appears after S_D^{USP14} in the pathways.

To provide additional structural evidence to support that the substrate-inhibited intermediates (S_{B/C/D} states) represent a pathway in parallel to those substrate-engaged intermediates (E_D-like states), we collected a new dataset at time point of 0.5 min after the substrate was added to the USP14-proteasome complex. As shown in the revised Fig. 1f and Extended Data Figs. 2b, c, 5b, the population of these substrate-inhibited states was retained at ~7-10% level from the 0 to 10 min after substrate addition, suggesting that these states coexist in the buffer and were stabilized by USP14.

8) The drawing of Fig. 4b is conflicted to some statements in the text. It's described on page 11 that "In support of this model, all substrate-engaged USP14-proteasome states showed no ubiquitin binding to RPN11." But in Fig. 4b, E_D^{USP14} has ubiquitin attached to Rpn11. Moreover, in Fig. 4b legend, it's written that "The substrate-inhibited pathway,.....has RPN11 blocking the substrate entrance of OB ring before any substrate insertion takes place." and "The substrate-engaged pathway, has RPN11 closed on the OB entrance after a substrate has already inserted into the ATPase ring". Based on these descriptions, Rpn11 orientation appears to be a key element to distinguish these states/pathways. However, the structural difference of Rpn11 in these states has never been described and illustrated in this manuscript. Also, in Fig. 4b, the Rpn11 are drawn in the same position for substrate-inhibited and substrate-engaged states. What's the difference between "blocking" and "closed on"?

Response: We thank the referee for pointing out the confusion in the cartoon drawing. The pink-cartooned subunit in Fig. 4 where ubiquitin is bound is meant to be RPN10, but not RPN11. RPN11 is colored in teal blue. To avoid any confusion, we have revised Fig. 4 accordingly by adding labels to RPN10 and RPN11 and redrawing their cartoon

representations. We have also added a few figure panels in Extended Data Fig. 7f to illustrate the RPN11 orientation relative to the OB ring. RPN11's positions with respect to the OB ring are comparable between the substrate-inhibited and substrate-engaged states. But RPN11 blocks the OB entrance tighter in the substrate-inhibited states than in the substrate-engaged states, as shown in the revised Extended Data Fig. 7f. RPN11 appears to move ~5 Å outwards in the substrate-engaged state as compared to the substrate-inhibited states.

9) *There lack a summary and definition on the overall influence of USP14 on UPS, and explanations on its physiological significance.*

Response: We appreciate the referee for this constructive suggestion of improvement. We have added a few sentences to summarize the overall influence of USP14 on UPS and its physiological significance on page 3, as quoted below:

“Human ubiquitin-specific protease 14 (USP14) is one of the three proteasome-associated deubiquitinating enzymes (DUBs)² and is crucially involved in the regulation of proteostasis, inflammation, neurodegeneration, tumorigenesis and viral infection^{2,14,15}. USP14 is a potential therapeutic target for treating cancer, inflammatory and neurodegenerative diseases^{6,14,16,17}. USP14 and its yeast orthologue Ubp6, which reversibly associate with the proteasome, have major roles in proteasome regulation⁴⁻⁷. USP14 is prominently activated upon association with the proteasome⁴⁻⁷. It stabilizes many cellular proteins against proteasomal degradation by ubiquitin chain disassembly as well as in a noncatalytic fashion¹⁸⁻²⁰.”

10) *On P. 6, the statement that “These kinetic features indicate ... that S_C^{USP14} and S_D^{USP14} changed into S_B^{USP14} in the late stage of substrate processing” is not supported. Dose the ‘late stage’ refer to the timespan between 5min and 10min? S_B^{USP14} didn't increase after 5min according to Fig. 1f, and the only increased state is E_{1A} if this refer to the time span between 1min and 5min.*

Response: We apologize for the insufficient clarity of this sentence. We have deleted this sentence to avoid confusion on page 6.

11) *In Extended Data Fig 5c, please also show the model-map fitting for the substrate and RPT subunits in the contacting regions for all the related states.*

Response: We have added a number of panels in the revised Extended Data Fig. 6a-f to show the model-map fitting for the substrate and RPT subunits in the contacting regions for all the six substrate-bound states.

12) *The authors describe that “While the overall proteasome conformations of states E_{D0}^{USP14} , E_{D1}^{USP14} and E_{D2}^{USP14} closely resemble those of USP14-free, substrate-engaged states $E_{D0,2}$, $E_{D1,2}$ and E_{D2} , only the AAA-ATPase components in states E_{D4}^{USP14} , $E_{D5,1}^{USP14}$ and $E_{D5,2}^{USP14}$ are comparable to those of USP14-free states E_B , E_{C1} and E_{C2} with defined differences, respectively^{10,25} (Extended Data Figs. 5, 6b)”. There is no conformational comparison data to support the similarity between E_{D0}^{USP14} and $E_{D0,2}$, and between $E_{D5,2}^{USP14}$ and E_{C2} in Extended Data Fig. 6b. Also, in Extended Data Fig. 5, we cannot see how “the pore-1 loop of RPT6 in state E_{D4}^{USP14} has considerably moved up toward the substrate as compared to that in state E_B (Extended Data Fig. 5c)”. Please provide data to support these conclusions.*

Response: We thank the referee for pointing this out. We have now provided in the revised Extended Data Figs. 5d-f a systematic comparison between all the USP14-bound states and those previously published USP14-free states.

The comparison between E_{D4}^{USP14} and E_B , as well as how the pore-1 loop of RPT 6 is moved up, are shown in the revised Extended Data Fig. 5d, e.

After we analyzed considerably more cryo-EM data, we were able to correct the misclassification of $E_{D5.2}^{USP14}$ which appears to be mixed from E_{D5}^{USP14} (previously $E_{D5.1}^{USP14}$) and a new substrate-free state S_{D5}^{USP14} , which is now refined to a resolution higher than that of the misclassified $E_{D5.2}^{USP14}$.

13) Please provide the data for the statement that “RPN11 blocks the substrate entrance at the OB ring of the AAA-ATPase motor in states S_B^{USP14} , S_C^{USP14} and S_D^{USP14} , to support their conclusion that “these states sterically inhibit substrate insertion into the AAA-ATPase channel”. Also show data to support their statement that “The AAA-ATPase motors of these substrate-inhibited states all adopt a S_A -like conformation”.

Response: We have revised Extended Data Fig. 7f to show how RPN11 blocks the substrate entrance at the OB ring in the substrate-inhibited states as compared to those E_A -like states.

In the revision, we have removed the original statement “The AAA-ATPase motors of these substrate-inhibited states all adopt a S_A -like conformation”, and revised Extended Data Figs. 5f, 7h, 7i to provide a more comprehensive pair-wise comparison to show the structural similarity and differences of the substrate-inhibited states as compared to other states.

14) The authors show the conformational comparison between E_{D4}^{USP14} and S_D^{USP14} and between E_{D4}^{USP14} and S_C^{USP14} in Extended Data Fig. 6b, and describe that “structural comparison between S_D and E_{D4} , which both have an open CP gate, indicates a highly comparable AAA-ATPase motor conformation, suggesting that S_D^{USP14} is perhaps an intermediate state connecting E_{D4}^{USP14} with other inhibitory states”. However, the aligned structures appear not very similar to each other in the AAA-ATPase motor region. Please provide the RMSD value between the AAA-ATPase motor of the two structures to validate their conclusion.

Response: To address this issue, we have provided detailed structural comparisons in the revised Extended Data Fig. 7d, where the RMSD values between the compared pair of AAA-ATPase motor structures are measured and labeled in the figure panels. A complete RMSD comparison between the ATPase motor structures of all coexisting states is provided in the revised Extended Data Fig. 7i.

15) The authors describe that “the pore-1 loop of RPT6 in state E_{D4}^{USP14} has considerably moved up toward the substrate as compared to that in state E_B (Extended Data Fig. 5c)”. But the referred figure can’t tell such difference, since it doesn’t include the compared E_B structure. Please generate a figure that can address this point.

Response: To address this issue, we have generated a figure panel in the revised Extended Data Fig. 5e to show the comparison of the pore-1 loop of RPT6 in state E_{D4}^{USP14} as compared to that in state E_B .

16) It’s written on Page 7 that “structural comparisons exhibit differential rotations of RPN1 and the lid relative to the base induced by USP14 (Extended Data Fig. 5b)”. But

Extended Data Fig. 5b shows the models for AAA-ATPase, doesn't include Rpn1 and the lid. The authors should generate a figure that can support this point.

Response: We have generated a figure panel, as shown in the revised Extended Data Fig. 7g to show this structural comparison. However, the original sentence was moved from the main text to the legend of Extended Data Fig. 7g, due to the length limitation of the main text.

17) Also on Page 7, the authors describe that "The BL1 region is an open loop in the crystal structures but is refolded into a β -hairpin sandwiched between the OB ring and ubiquitin in the proteasome". Refolding is a dramatic change. There are only panels rendering the BL1 model. Density details for the "refolded β -hairpin" should also be exhibited to convey their point.

Response: To address this question, we have added a new panel to Fig. 2d to show the β -hairpin density fitted with the atomic model of BL1.

18) In the last panel of Fig. 2e, the In vitro degradation experiment, the band intensities for substrates degraded by WT proteasome at the timepoint of 0.5/1.0/2.0 min appear stronger than that at 0 min. Why the band intensity is increased in the later time points if the sample loading is in equivalent amount in a parallel experiment? In addition, the W472A mutant appears promoting proteasome substrate degradation at the timepoint of 2 min, with the substrate seemingly being degraded at 2 min, which doesn't support their conclusion that "the single mutants F332A in BL1 and W472A in BL3 ... inhibited proteasome function (Fig. 2e, f)." There are similar concerns on their statement that "we made three mutants with deletion of residues 93-96, insertion of TEEQ after residue 92, and double point-mutations of E90K/D91A. While all three USP14 mutants exhibited inhibition of proteasomal degradation of Ub_n-Sic1^{PY}". However, E90K/D91A appears promoting proteasome mediated degradation if compare the band intensities at the timepoint of 2 min as shown in Extended Data Fig. 8c, d.

Response: We thank the referee for pointing these out. We have repeated the experiments and replaced the figure panel in the revised Fig. 2h and Extended Data Fig. 9 with the reproduced experimental results that did not have the issue. We suspect that the slight difference in the band intensities in previous Fig. 2e was likely due to the error in the pipetting procedure (such as tip volume variation) or in the electrophoresis step (such as local variation of the gel densities). Most of our repeated experiments did not show the same issue.

We agree with the referee on the two mutants and have revised our text accordingly. With respect to W472A mutant, the result has been reproduced for another three times during revision and is consistent with the original results. We have revised the interpretation that it only suppressed proteasome degradation to a lesser extent, as quoted below:

"... the single mutants F332A in BL1 and W472A in BL3 still retained a reduced DUB activity and suppressed proteasome function to a lesser extent"

We also agree with the referee on E90K/D91A, which appears to retain the activated DUB function of USP14, and it does show a reduced effect of degradation inhibition. We have revised the main text on page 8 accordingly to rectify the inaccurate description, as quoted below:

"In the linker region, the negatively charged residues Glu90 and Asp91 are potentially involved in transient interactions with the RPN1 toroid (Extended Data Figs. 7g, 9a). Deletion of residues 93-96 and insertion of TEEQ after residue 92 in the linker suppressed proteasomal degradation of Ub_n-Sic1^{PY} more potently than the double

point-mutation of E90K/D91A (Extended Data Fig. 8c, d). All three USP14 mutants showed 20-30% reduction in the DUB activity relative to the wildtype USP14.”

19) In Fig. 2f, there are Ubiquitin–AMC hydrolysis results for all the mutants in Fig. 2e, except for C114A, why? There are in vitro degradation results for C114A and E202K in Fig. 2e, but there isn't any comment in the text. So is K375Q in Fig. 3e.

Response: We thank the referee for pointing this out. We have revised the Fig. 2i to include the ubiquitin-AMC hydrolysis results for C114A. C114 catalyzes the deubiquitylation reaction in USP14. C114A is expected to completely disrupt the DUB activity of USP14. We have revised the main text on page 8 to address this question, as quoted below:

“..., the USP mutations, including single mutant K334Q and double mutant V343A/L344A at BL1-OB interface (Fig. 2f), F331A/Y333A at the ubiquitin-BL1 interface, as well as single mutant E202K disrupting the salt bridge between Glu202 of USP and Arg42 of ubiquitin (Fig. 2c), all abrogated the DUB activity of USP14, similar to the effect by C114A mutation that removes the active site of USP14 completely, and showed no obvious inhibition of proteasomal degradation (Fig. 2h, i, Extended Data Figs. 9e, f, 10).”

We also explained K375Q in the revised main text on page 9, as quoted below:

“K375Q showed no significant effect on the ATPase activity in that only the main chain of K375 interacts with RPT1 His197 (Fig. 3g).”

20) On page 8, the authors describe that “States E_{D4}^{USP14} , $E_{D5.1}^{USP14}$ and $E_{D5.2}^{USP14}$ are specifically induced by USP14, since no comparable conformations were reconstructed in the absence of USP14 previously despite exhaustive 3D classification”. They should make it clear why the other USP14 bound states like E_{D1}^{USP14} are not included in this statement, what's the difference between these three and the others, and how they compare with the previous ones?

Response: We thank the referee for this suggestion. To avoid confusion, we have removed this sentence in the revision and added a complete set of panels to show the structural comparison of all USP14-bound conformations with USP14-free conformations in Extended Data Fig. 5f.

21) There should be a more in-depth analysis on the mechanism of how and why the USP mutations influence the ATPase activity differently.

Response: We have revised Fig. 3 to include more structural analysis and illustrations, as shown in the revised Fig. 3e-h, to explain how and why USP mutations influence the ATPase activity differently. We also revised the main text on page 9 to provide more explanation, as quoted below:

“In state E_{D4}^{USP14} , ADP is bound to the Walker A motif between the large and small AAA subdomain of RPT1, the dihedral angle of which is directly stabilized by the USP-AAA interaction (Fig. 3f). In other substrate-engaged states, in which the USP domain is detached from the AAA domain, the nucleotide-binding pocket of RPT1 is either empty or bound to ATP. Thus, the USP-AAA interaction energetically stabilizes the ADP-bound conformation of RPT1 and USP-AAA dissociation promotes nucleotide exchange in RPT1. We postulate that the USP-AAA interaction may allosterically promote the ATPase rate during substrate degradation. Indeed, wildtype USP14 enhanced the ATPase activity during proteasomal degradation of Ub_n -Sic1^{PY} by approximately 10% relative to that of the USP14-free proteasome (Fig. 3i).

... Both Y285A and N383A mutations potentially reduce the overall USP association with the large AAA subdomain of RPT1 (Fig. 3g). In contrast, the single mutant R371A enhanced the ATPase activity by approximately 20% relative to the wildtype USP14 likely via tightening the USP-AAA interactions, whereas K375Q showed no significant effect on the ATPase activity probably because only the main chain of K375 interacts with RPT1 His197 (Fig. 3g). Consistently, the double mutant Y285A/R371A restored the ATPase rate to the approximate level of the USP14-free proteasome, presumably due to the cancellation of two counteracting allosteric effects between the two mutated residues (Fig. 3j)."

23) *The authors describe that "Catalytically, USP14 trims ubiquitin chains from the substrate quite fast⁷. For this reason, we have not captured the USP14 conformation right at the moment of deubiquitylation." Conformation right at the moment of deubiquitylation at Rpn11 was captured for 26S proteasome in their previous study. So why deubiquitylation mediated by USP14 can't be captured? Does it trim ubiquitin chains much faster than Rpn11? Is there any supporting evidence on this?*

Response: The reason that we had previously captured the state of RPN11 deubiquitylation is perhaps that this state is allosterically coupled to the substrate engagement with the ATPase motor that can be stalled by the ATP-to-ATP_γS exchange. In the USP14-bound proteasome, perhaps because USP14 deubiquitylation does not require substrate engagement with the ATPase, the same method of ATP-to-ATP_γS exchange did not allow the capture of the state right at the moment of USP14-catalyzed deubiquitylation, which has been previously suggested to be on the time scale of millisecond (ref. 7). Nonetheless, we do not expect that the overall conformation of activated USP14 changes much before and after substrate deubiquitylation.

In the USP14-loaded proteasome, the DUB activity of RPN11 is allosterically suppressed and USP14 removes ubiquitin chains much faster than RPN11 as shown in the previous studies (ref. 7). In the absence of USP14, RPN11 appears to be quite active in its DUB activity and is expected to be as fast as USP14 to not limit the rate of degradation (ref. 7).

24) *In Fig. 3e, please also provide the P values for the ATPase activity assay.*

Response: We have provided the *P* values for the ATPase activity assay in the revised Fig. 3j. The *P* values were calculated between the mutants and the wildtype USP14.

Minor points:

1) *Please make it clear what the purpose is by adding ATP_γS in the system and how it can quench the ATPase activity of the ATPase?*

Response: We have added a sentence on page 4 to explain the purpose of adding ATP_γS, as quoted below:

"To facilitate structural determination at high resolution, we collected another large dataset by exchanging ATP with the slowly hydrolyzed ATP_γS 1 minute after substrate addition at 10 °C, which is expected to stall all coexisting intermediate conformations¹⁰."

2) *The meaning of "W/O" should be defined in the caption of Fig. 2.*

Response: We have added a short description of "W/O" in the Fig. 2 legend, as quoted below:

"W/O, the proteasome without USP14."

3) It's written on page 9 "The primary USP-AAA interface is mediated by a negatively charged surface on a helix (residues 372-383)". The negatively charged surface should be illustrated. Interacting residues in USP-AAA interfaces are analyzed in this paragraph. What's the criteria to count for interaction, distance between atoms? The authors are also suggested to show how well the model fits in map in the interaction interface. Additionally, the color style in Fig. 3c is confusing. Readers intuitively think the red residues belong to the orange part, and the blue residues belong to the blue chain.

Response: We thank the referee for kind suggestions. In accordance, we have added a panel in Fig. 3e to illustrate the negatively charged surface on USP14. The interactions were counted by both measuring the distance between atoms, as shown in Fig. 3i, and by calculating the buried surface using the PISA server (ref. 59). We have added this information in the revised Methods section "Structural analysis and visualization", as quoted below:

"Inter-subunit interactions and interfacial areas were computed and analyzed using the PISA server⁵⁹ (https://www.ebi.ac.uk/pdbe/prot_int/pistart.html)."

We have also added a panel in the revised Extended Data Fig. 4f to illustrate how well the model fits in the density of the USP-AAA interface. We have revised the Figs. 2c and 3f-h (replacing previous Fig. 3c) to avoid the confusing color style.

4) *The difference between black dashed arrow and black solid arrow are not introduced in the figure caption of Fig.4a.*

Response: To address this comment, we have added a sentence in the legend of the figure, which is now moved from Fig. 4a to Extended Data Fig. 7j, to explain the difference between black dashed and black solid arrows, as quoted below:

"Black solid arrows and dashed arrows represent the possible structural transitions connecting the states observed in the current study and pervious USP14-free studies^{8,10}, respectively."

Reviewer Reports on the First Revision:

Referees' comments:

Referee #1 (Remarks to the Author):

The authors have addressed most of the major concerns this reviewer had. The newly added dissociation constants of USP14 and its two functional domains binding to the proteasome are particularly informative for understanding the dynamic interaction between the enzyme and the degradation machinery. While this reviewer appreciates the extra efforts the authors spent in providing this piece of data, the statement that “USP14 affinity toward the proteasome is *substantially* enhanced by Ubn-Sic1PY” is unfortunately inaccurate and exaggerated. As shown in ED Figure 1i & 1o, the KD value of USP14 toward the proteasome without and with the substrate is 94.6 +/- 27.1 and 43.9 +/- 24.7, respectively. Given the large error bar associated with the measurements and the somewhat questionable MST binding curves, one can barely conclude that the substrate enhances the affinity to USP14 toward the proteasome. This reviewer urges the authors to reconsider making such a statement, which does a disservice to this otherwise structurally revealing study.

Referee #2 (Remarks to the Author):

In the revised manuscript from Zhang and colleagues, additional biochemical and structural data have been incorporated that strengthens the conclusions made in the previous submission. All my previous comments have been sufficiently addressed and the improved figures and textual changes enhance the importance of the study.

Referee #4 (Remarks to the Author):

The authors have addressed my questions and comments and the manuscript has been greatly improved. I have no further questions and would recommend publication of the manuscript.

Author Rebuttals to First Revision:

Point-by-Point Responses to Referees' Comments on Manuscript 2021-07-11629C

We appreciate the reviewing work that all referees have done for our revised manuscript and for their support of publication of this study. In the revised manuscript, we have addressed the referees' comments in full.

Referee #1:

The authors have addressed most of the major concerns this reviewer had. The newly added dissociation constants of USP14 and its two functional domains binding to the proteasome are particularly informative for understanding the dynamic interaction between the enzyme and the degradation machinery. While this reviewer appreciates the extra efforts the authors spent in providing this piece of data, the statement that "USP14 affinity toward the proteasome is *substantially* enhanced by Ubn-Sic1PY" is unfortunately inaccurate and exaggerated. As shown in ED Figure 1i & 1o, the KD value of USP14 toward the proteasome without and with the substrate is 94.6 +/- 27.1 and 43.9 +/- 24.7, respectively. Given the large error bar associated with the measurements and the somewhat questionable MST binding curves, one can barely conclude that the substrate enhances the affinity to USP14 toward the proteasome. This reviewer urges the authors to reconsider making such a statement, which does a disservice to this otherwise structurally revealing study.

Response: We thank the referee very much for constructive comments and recommending publication of this work in *Nature*. We totally agree with the referee's concern and thus have revised the quoted sentence on page 8 (line 191) by removing the word "substantially".

Referee #2:

In the revised manuscript from Zhang and colleagues, additional biochemical and structural data have been incorporated that strengthens the conclusions made in the previous submission. All my previous comments have been sufficiently addressed and the improved figures and textual changes enhance the importance of the study.

Response: We thank the referee very much for constructive comments and recommending publication of this work in *Nature*.

Referee #4:

The authors have addressed my questions and comments and the manuscript has been greatly improved. I have no further questions and would recommend publication of the manuscript.

Response: We thank the referee very much for many constructive suggestions on the improvement of the manuscript and for recommending publication of this work in *Nature*.